# An assessment of the global impact of 21st century land use change on soil erosion

Pasquale Borrelli [1,2], David A. Robinson [3], Larissa R. Fleischer[4], Emanuele Lugato [2], Cristiano Ballabio[2], Christine Alewell[1], Katrin Meusburger[1], Sirio Modugno[5], Brigitta Schütt[6], Vito Ferro[7], Vincenzo Bagarello[8], Kristof Van Oost[9], Luca Montanarella[2] & Panos Panagos[2]

Human activity and related land use change are the primary cause of accelerated soil erosion, which has substantial implications for nutrient and carbon cycling, land productivity and in turn, worldwide socio-economic conditions. Here we present an unprecedentedly high resolution (250 × 250 m) global potential soil erosion model, using a combination of remote sensing, GIS modelling and census data. We challenge the previous annual soil erosion reference values as our estimate, of 35.9 Pg yr$^{-1}$ of soil eroded in 2012, is at least two times lower. Moreover, we estimate the spatial and temporal effects of land use change between 2001 and 2012 and the potential offset of the global application of conservation practices. Our findings indicate a potential overall increase in global soil erosion driven by cropland expansion. The greatest increases are predicted to occur in Sub-Saharan Africa, South America and Southeast Asia. The least developed economies have been found to experience the highest estimates of soil erosion rates.

[1] Environmental Geosciences, University of Basel, Basel CH-4056, Switzerland. [2] European Commission, Joint Research Centre, Directorate for Sustainable Resources, Ispra I-21027, Italy. [3] NERC Centre for Ecology and Hydrology, Environment Centre Wales, Bangor LL57 2UW, United Kingdom. [4] Independent Researcher, Baden-Württemberg 70376, Germany. [5] World Food Programme, Roma 00148, Italy. [6] Department of Earth Sciences, Physical Geography, Freie Universität Berlin, Berlin 12249, Germany. [7] Department of Earth and Marine Science, University of Palermo, Palermo 90123, Italy. [8] Department of Agricultural, Food and Forest Sciences, University of Palermo, Palermo 90128, Italy. [9] TECLIM-Georges Lemaître Centre for Earth and Climate Research, Université Catholique de Louvain, Louvain-la-Neuve, BE 1348, Belgium. Correspondence and requests for materials should be addressed to P.B. (email: Pasquale.Borrelli@unibas.ch) or to (email: lino.borrelli@yahoo.it)

Healthy soil is the foundation of agriculture and an essential resource to ensure human needs in the 21st century[1], such as food, feed, fibre, clean water and clean air. It is a vital part of ecosystems and earth system functions that support the delivery of primary ecosystem services[2,3].

The latest reference document of the United Nations (UN) on the status of global soil resources stresses that '…the majority of the world's soil resources are in only fair, poor or very poor condition'[4]. The results of the meta-analysis reported in this document indicate that accelerated soil erosion is a major threat to soil. This is in line with previous assessments[5,6]. The harmful impacts of accelerated soil erosion processes caused by deforestation, overgrazing[7], tillage and unsuitable agricultural practices[8] are well-known and documented, as are its mechanics[9,10]. Impacts can be severe, not only through land degradation and fertility loss, but through a conspicuous number of off-site effects (e.g., sedimentation, siltation and eutrophication of water ways or enhanced flooding[11]). The impact on climate through erosion-induced changes in soil carbon cycling also remains poorly quantified, as erosion can both increase or decrease $CO_2$ emissions through enhanced mineralization and sediment burial[12,13].

The fear of soil erosion, especially the associated removal of the most fertile soil layer as a prelude to mass starvation has been revised lately[14]. The 13% increase in production rates for the most common crops[15] between 2001 and 2012, due to technological improvements, more rigorous land management[16] and an increased use of fertilizer[17], might have masked the ongoing degradation of soils and their ecosystem service delivery capacity. Feeding Earth's growing population with increasing dietary preferences towards livestock products is undoubtedly enhancing the pressure on fertile soils[18] thus exacerbating the erosion problem. Sustainable governance of soil has therefore become a topic of fundamental importance[19].

The FAO led Global Soil Partnership[20] reports that 75 billion tonnes (Pg) of soil are eroded every year from arable lands worldwide, which equates to an estimated financial loss of US $400 billion per year. This soil erosion estimate dates back to 1993, first reported by Myers[21] and cited by several succeeding studies[19,22,23]. A lack of reliable global erosion estimates forces the scientific community to resort to these pioneering studies carried out during the late 1980s and early 1990s such as UNEP's project Global Assessment of Soil Degradation (GLASOD)[5]. GLASOD and its successor GLADIS[24] provided insights into soil erosion based on a static observation approach but did not quantify the effects driven by changes in land use. Accelerated soil erosion is primarily driven by modifications in land use and management. Spatial patterns of land use and land cover change, especially in areas susceptible to accelerated soil erosion, provide further reason to re-evaluate former qualitative approaches, considering the worldwide increase of croplands and pastures by 279 million hectares (ca. 16.7%) between 1985 and 2013[15,25,26].

Following the publication of GLASOD, subsequent research aimed to improve the ability to predict soil erosion under global change[27]. Models placed greater emphasis on representing the physical processes[28]. This, led to a broader understanding of the processes and methodological improvements[29]. Scaling in space and time, however, was a great challenge for the new mechanistic models[30]. Nearing[9] observed that the disadvantage of the process-based models is the model complexity and their substantial data requirements. The extremely high demand for input data[31] generally precluded their application above field or small catchment scales[32].

In a context where process-based physical models and the availability of input data are not yet mature enough for global scale applications[31], simple, physically plausible empirical methods for predicting soil erosion such as RUSLE[33], can provide reasonably accurate estimates for most practical purposes (Supplementary Note 1). This applies especially to wide spatial scale applications when prediction errors do not exceed a factor of two or three[34]. Notwithstanding the significant scientific contribution of the expert-based GLASOD[5] approach in the 1990s and the need to further advance process-based physical models[28,35], recent studies have shown the potential of RUSLE[33] model-based approaches as a significant step towards a change in global water erosion assessment[36,37].

These scoping studies[36,37], compared to GLASOD, provided a more detailed view of the spatial patterns of accelerated soil erosion at a global scale. They relied, however, on coarse model input data, particularly with respect to land use patterns and erosive power of rainfall. The predictive power of these models is therefore limited to assessments of the land use change effects[38].

In this study, we provide quantitative, thorough estimates of soil erosion at the global scale by means of a high-resolution, spatially distributed, RUSLE-based[33] modelling approach. Unlike previous studies that dealt with soil erosion as a static process, here we shed light on the impacts of 21st century global land use change on soil erosion. Insights into land cover and land use change between 2001 and 2012 are achieved by combining the extent, types and spatial distribution of global croplands and forests measured by satellite imaging with agricultural inventory data. The global rainfall erosivity patterns are quantified with a thorough methodology based on rainfall intensity instead of volume, using a time series of sub-hourly and hourly pluviographic records (3625 stations covering 63 countries) spatialized through a Gaussian Process Regression (GPR) geo-statistical model.

## Results

**Modelling scheme.** The RUSLE-based modelling approach (Supplementary Fig. 1 and Supplementary Methods) provide estimates of the potential rates of soil displacement by water erosion (soil erosion) (Supplementary Note 2) on a $250 \times 250$ m grid cell basis for the land surface of 202 countries (ca. 2.89 billion cells; ~125 million km$^2$); covering ~84.1% of the Earth's land surface. We present results for soil erosion based on data for 2001 and 2012, taking into consideration the individual land cover type, vegetation cover dynamics and farming systems of each cell. For cropland, we ran an additional conservation scenario spatializing the information of the 54 countries that reported the application of conservation tillage practices to the FAO.

**Global perspective on soil erosion.** Our baseline model predicts an annual average potential soil erosion amount of 35 Pg yr$^{-1}$ for 2001, with an area-specific soil erosion average of 2.8 Mg ha$^{-1}$ yr$^{-1}$. In 2012, we estimated an overall increase of 2.5% in soil erosion (35.9 Pg yr$^{-1}$), driven by spatial changes of land use. The area which had undergone a change during the study period totalled about 3.3% of the study area (equal to ~4 million km$^2$), 2.9 million km$^2$, of which ~2.4% showed an increase in the estimates of soil erosion of 1.74 Pg yr$^{-1}$. The remaining 1.1 million km$^2$ (~0.9%) under land use change experienced a decrease in soil erosion of 0.88 Pg yr$^{-1}$. This results in an estimated overall increase of soil erosion for areas with land use change of about 0.86 Pg yr$^{-1}$. The soil conservation practice scenario shows a potential overall offset of the estimated soil erosion increase of about 64% (overall increase 0.31 Pg yr$^{-1}$), primarily driven by the effects of the conservation practices in North America, South America and other advanced and transition economies. This offset is the result of heterogeneous regional dynamics.

The spatial pattern of soil erosion in 2012 is illustrated in Fig. 1. Areas classified as having very low, and low erosion rates (class 1 and class 2), represent about 71.9% and 12.7% of the total, respectively. Moderate (class 3) and high (class 4) soil erosion values are predicted for about 4.2% and 5.1% of the study area, respectively. The remaining land surface (classes 5–7), about 7.5 million $km^2$ in total (6.1% of the land), exceeds the generic tolerable soil erosion threshold ($T$-value) (10 Mg $ha^{-1}$ $yr^{-1}$) in 2012. Descriptive statistics about the severity of soil erosion across the continents in 2012 are provided in Table 1.

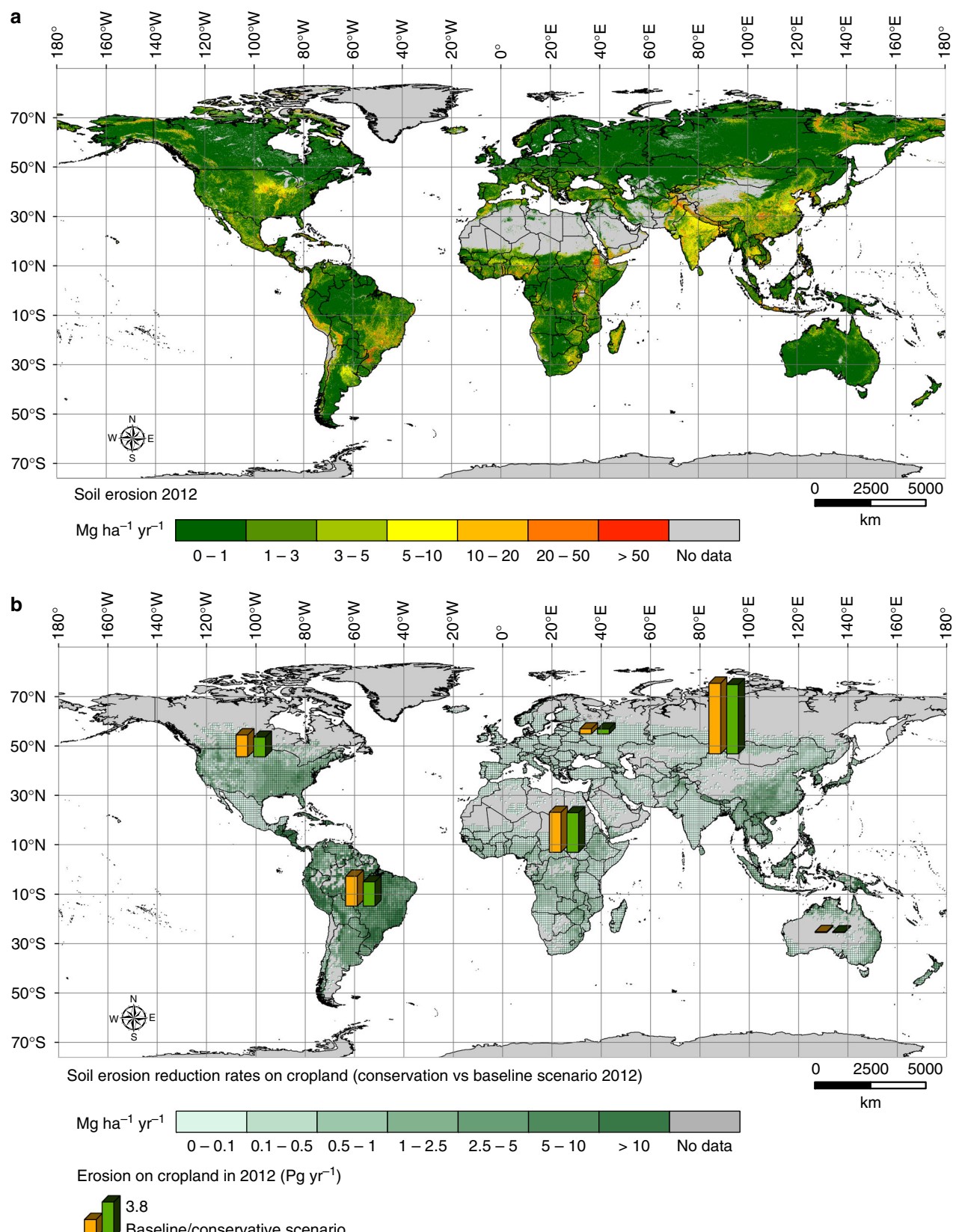

**a** Soil erosion 2012

Mg $ha^{-1}$ $yr^{-1}$

| 0 – 1 | 1 – 3 | 3 – 5 | 5 –10 | 10 – 20 | 20 – 50 | > 50 | No data |

**b** Soil erosion reduction rates on cropland (conservation vs baseline scenario 2012)

Mg $ha^{-1}$ $yr^{-1}$

| 0 – 0.1 | 0.1 – 0.5 | 0.5 – 1 | 1 – 2.5 | 2.5 – 5 | 5 – 10 | > 10 | No data |

Erosion on cropland in 2012 (Pg $yr^{-1}$)

3.8
Baseline/conservative scenario

**Continental and country-specific perspectives on soil erosion.**
Our modelling results suggest that water erosion is a common phenomenon under all climatic conditions encompassing all observed continents. Country-specific results and changes of the estimated annual average soil erosion values between 2001 and 2012 are illustrated in Fig. 2. According to the baseline scenario, at a continental level, South America shows the highest prediction of average soil erosion rate (3.53 Mg ha$^{-1}$ yr$^{-1}$) in 2001, followed by Africa (3.51 Mg ha$^{-1}$ yr$^{-1}$) and Asia (3.47 Mg ha$^{-1}$ yr$^{-1}$). North America, Europe and Oceania show considerably lower predicted values, totalling 2.23, 0.92 and 0.9 Mg ha$^{-1}$ yr$^{-1}$, respectively. In 2012, the latter group of continents indicated an estimated decreasing trend of soil erosion driven by land use change, with the highest decrease predicted in North America (4.8%). For Asia, we predict a slight increase of about one percent, mainly driven by a noticeable increase in soil erosion in the Southeast Asian countries. China (~2%) and India (~0.45%), in contrast, even though densely populated show a decrease in soil erosion estimates, while we observe a noteworthy increase in soil erosion in South America and Africa. Africa surpasses South America with an estimated increase of soil erosion of ~10% in 2012, thus becoming the continent with the highest average soil erosion rate (3.88 Mg ha$^{-1}$ yr$^{-1}$). This seems to be primarily driven by a widespread increase of erosion in the western and central African countries. For the same period, South America showed a predicted ~8% increase in soil erosion as a consequence of deforestation and the large expansion of cropland areas in Argentina (41.6%), Brazil (19.8%), Bolivia (37.8%) and Peru (5.9%).

Classified based according to the International Monetary Fund and United Nations classification, the least developed economies experienced the highest prediction of soil erosion rate in 2001 (4.81 Mg ha$^{-1}$ yr$^{-1}$), equal to 4.8 Pg yr$^{-1}$ and 13.6% of the global soil erosion. The less developed economies have the second highest predicted rate of soil erosion (4.74 Mg ha$^{-1}$ yr$^{-1}$), followed by the advanced economies (1.61 Mg ha$^{-1}$ yr$^{-1}$) and the transition economies (1.02 Mg ha$^{-1}$ yr$^{-1}$). The less developed economies show the highest prediction of annual total soil erosion (20.7 Pg yr$^{-1}$), equal to 59.2% of the global soil erosion. For 2012, we observed a decrease in soil erosion estimates driven by land use change in the advanced (−5.3%) and transition economies (−0.14%), while we estimated substantial increases in the less developed (3%) and least developed economies (11.7%). Notably, the least developed economies (mostly located in Sub-Saharan Africa) show a predicted increase in soil erosion that is three times higher than the less developed economies.

Looking for soil erosion hot-spots higher than 20 Mg ha$^{-1}$ yr$^{-1}$, the largest and most intensively eroded regions (Supplementary Fig. 2) are predicted to be in China (0.47 million km$^2$; 6.3% of the country), Brazil (0.32 million km$^2$; 4.6% of the country), the African territories located along the Equator (0.26 million km$^2$; 3.2% of the region), India (0.20 million km$^2$; 7.5% of the country), South-eastern United States (0.2 million km$^2$; 1.9% of the country) and to a lesser extent in Central Eastern Ethiopia (0.084 million km$^2$; 9.5 of the country), Mexico (0.079 million km$^2$; 4.6% of the country), Indonesia (0.076 million km$^2$; 5% of the country), Peru (0.074 million km$^2$; 7.5% of the country) and Mediterranean Europe (0.06 million km$^2$; 3.2% of the region). Very high estimates of soil erosion rates are often locally exacerbated by high annual average rainfall erosivity values, greater than 4500 MJ mm ha$^{-1}$ h$^{-1}$ yr$^{-1}$ (global average 2000 MJ mm h$^{-1}$ ha$^{-1}$ yr$^{-1}$) which characterize regions such as sub-Equatorial Africa, Southern China, India, South Eastern United States, Northern Oceania, South America and Mexico (Supplementary Fig. 3).

**Land use/cover and soil erosion.** Comparing soil erosion based on land use types, we find a significant decline in the estimates of soil erosion rates from croplands to forests and other forms of semi-natural vegetation. Cropland covers about 11% of the studied land in 2001 and 11.2% in 2012, but is responsible for 49.7% and 50.5% of the total predicted soil erosion, respectively. In either period, the predicted average soil erosion in croplands is more than four times higher than the overall soil erosion rate (12.7 Mg ha$^{-1}$ yr$^{-1}$ compared to an average of 2.8 Mg ha$^{-1}$ yr$^{-1}$). It is estimated to be 77 times higher than in forests (0.16 Mg ha$^{-1}$ yr$^{-1}$) and around seven times higher than the average of the other natural vegetation (1.84 Mg ha$^{-1}$ yr$^{-1}$). Despite the fact that forests cover more than 28% of the observed global land, they have, on average, the lowest soil erosion estimate with about 0.6 Pg yr$^{-1}$, thus being responsible for ca. 1.7% of the total soil erosion estimates.

**Land use changes as drivers of soil erosion.** We assessed the dynamics in land use between 2001 and 2012. The total gross land stock changed about 3.3% (4 million km$^2$ of ~125

| Table 1 Continental perspective on the severity of water erosion in 2012 | | | | |
|---|---|---|---|---|
| **Continent** | **Light (classes 1 and 2)** | **Moderate (class 3)** | **High (class 4)** | **Above T-value (classes 5–7)** |
| *Surface area (%)* | | | | |
| Africa | 80.3 | 6.0 | 6.0 | 7.7 |
| Asia | 80.6 | 4.9 | 6.9 | 7.6 |
| Europe | 94.5 | 2.1 | 1.7 | 1.6 |
| North America | 87.7 | 3.7 | 4.3 | 4.3 |
| South America | 81.9 | 4.6 | 5.2 | 8.3 |
| Oceania | 96.2 | 1.7 | 1.2 | 0.8 |

**Fig. 1** Global rates of soil displacement by water erosion. The estimates are predicted through a RUSLE-based modelling approach integrated in a geographic information system (GIS) environment. The model provides rates on an ~250 × 250 m cell basis for the land surface of 202 countries (ca. 2.89 billion cells; ~125 million km$^2$), covering about 84.1% of the Earth's land. Panel **a** illustrates the soil erosion rates divided into seven classes according to the European Soil Bureau classification. The colour gradation from green to red indicates the intensity of the predicted erosion rates. The grey colour indicates the areas that were excluded from the modelling due to data unavailability (i.e., ice-covered land, terrestrial water bodies, large area of bare rock, deserts and land with bare soil). Panel **b** illustrates the erosion reduction rates on cropland obtained from the comparison between the conservation and the baseline scenario for the year 2012. The green gradient shows the percentage of reduction. The grey colour indicates the areas that were not modelled (no data)

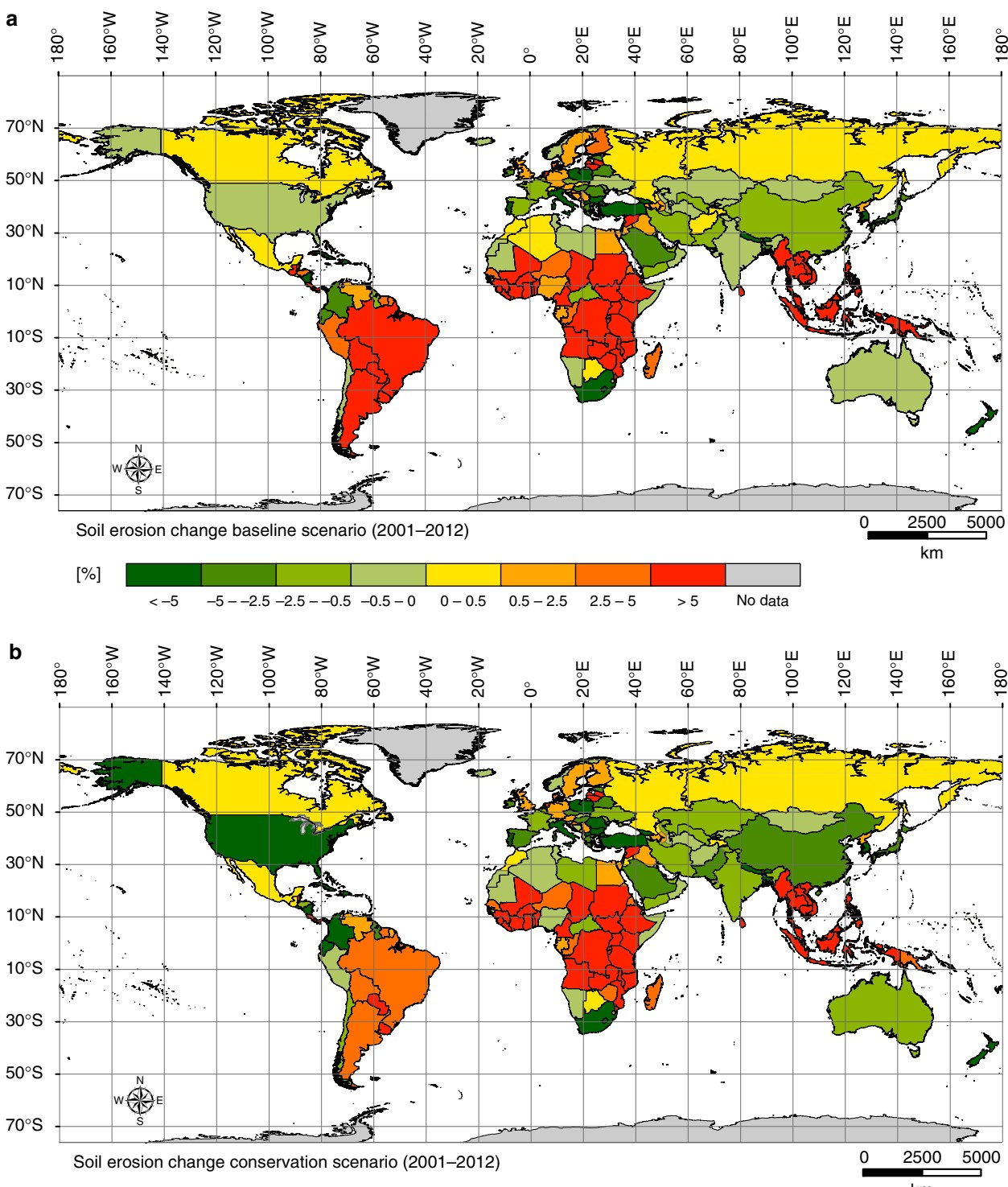

**Fig. 2** Country-specific changes of the annual average soil erosion. Panels **a** and **b** share the same legend. The chromatic scale represents the percentages of increase or decrease of the annual average soil erosion rates obtained by comparing the pixel-based values in each of the 202 countries under observation. **a** Soil erosion change between 2001 and 2012 according to the baseline scenario. The delta between the two observed periods solely depends on the land use and land cover change outlined combining satellite-derived land use land cover information with agricultural inventory data. **b** Soil erosion change between 2001 and 2012 according to the conservation scenario. In this case, the percentage change of erosion results from the combined effect of land use/land cover change and the mitigating effect of soil conservation practices

million km$^2$ of observed land) (Fig. 3). This resulted in a forest decline (~1.65 million km$^2$), an expansion of the semi-natural vegetation (savannah, scrublands, grassland, transition forest) (~1.43 million km$^2$) and an expansion of cropland (~0.22 million km$^2$).

A total of 2.26 million km$^2$ of forests are estimated to be lost during the study period, mostly replaced by semi-natural vegetation (2.17 million km$^2$) and to a lesser extent by cropland (~0.1 million km$^2$). In the baseline scenario, the change from forest to other land uses caused an increase of 0.61 Pg yr$^{-1}$ of soil

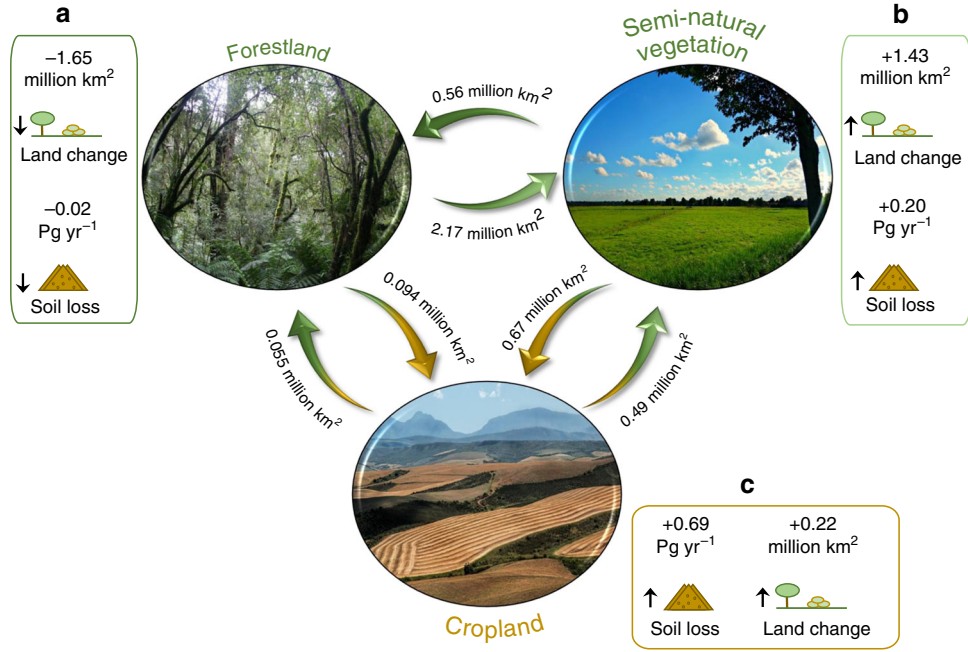

**Fig. 3** Flow diagram of land use changes and their effects on the soil loss estimates. The circular forms refer to the three major land use and land cover groups obtained from the combination of the MODIS MCD12Q1 Land Cover Type (International Geosphere Biosphere Programme (IGBP) system), Global Forest Change maps, and cropland extend and harvested area statistics extracted from the Food and Agriculture Organization's (FAO) FAOSTAT database. The arrows indicate the amount (million $km^2$) of land use and land cover change between 2001 and 2012. Insets **a–c** demonstrate the net change of the land surface (million $km^2$) and soil loss (Pg $yr^{-1}$)

erosion (~ 2200%). The substitution of forests for cropland (4.1% of the forest lost) is responsible for about 52% of this increase in soil loss. At the same time, a forest area gain of about 0.61 million $km^2$ occurred during the study period, resulting in a net loss of ~1.65 million $km^2$. With regard to the soil erosion balance, this forest change accounts for an estimated overall decrease of soil erosion in forestland equal to 0.38 Pg $yr^{-1}$.

With regard to cropland, about 90.5% of the land classified as cropland did not change during the observed period. The change in cropland (loss or gain) is equal to 1.31 million $km^2$ (32% of the total global land use change). The cropland abandonment amounts to 0.55 million $km^2$ (with ca. 88% and ca. 12% of cropland recolonized by forest and semi-natural vegetation, respectively), while 0.76 million $km^2$ of new cropland was established at the expense of semi-natural vegetation (ca. 90%) and forest (ca. 10%). This translates to an increase in soil erosion of 0.69 billion Mg $ha^{-1}$ $yr^{-1}$. Considering the overall increase in soil erosion of about 0.86 Pg $yr^{-1}$, the land changes related to cropland are responsible for about 80% of this increase.

**Conservation agriculture and soil erosion**. In Fig. 4, we illustrate the variation of soil erosion modelled for a selection of the 54 countries, which reported the proportion of their cropland area under conservation agriculture to the FAO. The conservation agriculture covers about 15.3% of the observed cropland (1.6 of 10.3 million $km^2$), resulting in an estimated overall soil erosion reduction of about 7% compared to the baseline scenario in 2012 (from 10.93 to 10.15 Pg $yr^{-1}$). At a continental level, the highest soil erosion reductions are estimated in South America (16%), Oceania (15.4%), North America (12.5%), and to a lesser extent in Europe (1.5%), Asia (1.2%) and Africa (1.1%). From a country-specific perspective, the effects of conservation agriculture are remarkable in South American countries like Argentina, Paraguay

and Brazil, where soil erosion is potentially reduced by 33%, 27% and 20%, respectively.

Of these countries, 28 regularly reported the proportion of their cropland under conservation agriculture during the last decade for different time periods. From an analysis of the data, different dynamics of the continental conservation pattern can be inferred. Oceania (+21.1%), North (17.3%) and South America (+14.2%) show the most substantial increases in areas under conservation, while Asia (5.8%), Europe (2.4%) and Africa (1.8%) have considerably lower values.

The benefits of reducing soil erosion due to the adoption of soil conservation practices are notable. We estimate that if the countries with no information about conservation agriculture would follow the continental patterns obtained from the analysis of the 28 countries, ca. 0.49 Pg $yr^{-1}$ (total 34.5 Pg $yr^{-1}$) and 1.05 Pg $yr^{-1}$ (total 34.8 Pg $yr^{-1}$) of eroded soil would be avoided in 2001 and 2012, respectively. This would redesign the global patterns described in the baseline scenario as follows. At continental level, Africa (3.86 Mg $ha^{-1}$ $yr^{-1}$) and Asia (3.47 Mg $ha^{-1}$ $yr^{-1}$) would show the highest soil erosion rate in 2012. The highest impact of conservation agriculture would be observed in South America (3.38 Mg $ha^{-1}$ $yr^{-1}$), North America (2.03 Mg $ha^{-1}$ $yr^{-1}$) and Oceania (0.74 Mg $ha^{-1}$ $yr^{-1}$), while the impact in Europe would be limited (0.89 Mg $ha^{-1}$ $yr^{-1}$). With regard to the socio-economic prospective, the situation of the conservation scenario does not differ from the baseline scenario. However, while the less developed countries show the highest soil erosion reduction in 2012 (−3.9% vs. −2.9% of the advanced economies), the least developed countries experience a lower increase (−1.1%).

**Discussion**

Land management and the related land use changes have an effect on the spatial patterns and magnitude of accelerated soil erosion

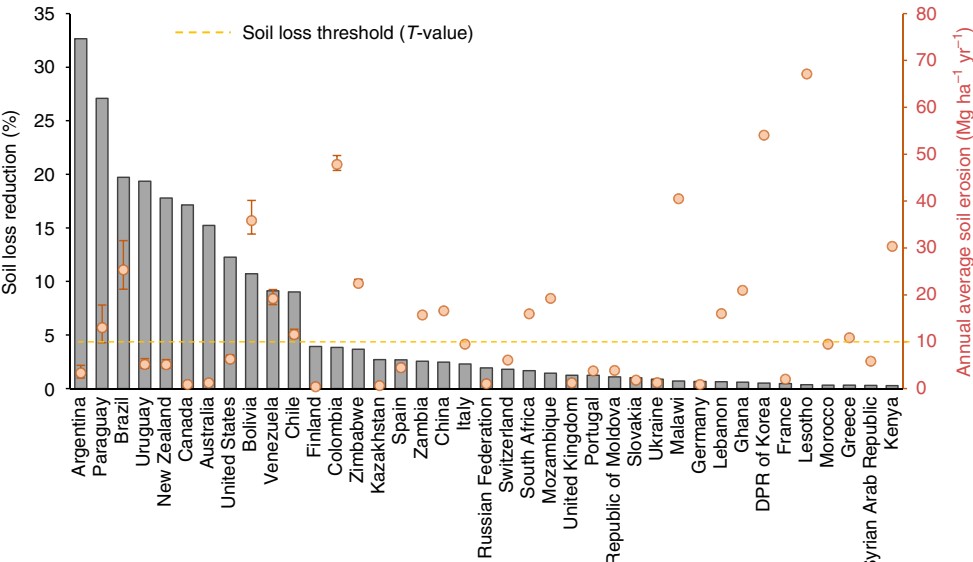

**Fig. 4** Estimated soil loss reduced by conservation agriculture. The grey bars illustrate the estimated soil loss reduction (in percent) derived from the implementation of conservation agriculture (40 countries show the highest reduction values). The values refer to the model application for the year 2012 adjusted for the potential effect of conservation agriculture practices. Red dots indicate the national average of annual soil loss estimated in cropland (Mg ha$^{-1}$ yr$^{-1}$). The red error bars around the dots indicate the variation between the mean values of the conservation scenario, the baseline scenario (positive bar) and the maximum mitigation effect of the practices (negative bar). The dotted orange line shows the soil loss tolerance threshold (T value—10 Mg ha$^{-1}$ yr$^{-1}$)

which can affect land productivity and food security[12,18], biological diversity[39] and carbon cycling[13,38,40]. Global soil erosion dynamics have been previously quantified based on scientific soil expert judgments[4,5], through the extrapolation of plot and river sediment data[41,42] and RUSLE-based modelling[36,37]. While these approaches range in their degree of complexity, their lumped or coarse resolution modelling (ca. 10–60 km cell size) with a static observation approach limits their predictive power to assess the effects of frequent land use changes and to identify soil erosion hotspots. Our study investigates the global soil erosion dynamics by means of high-resolution spatially distributed modelling (ca. $250 \times 250$ m cell size). The proposed geo-statistical approach, allows for the first time, the thoroughly incorporation of land uses and their changes, the extent, types, spatial distribution of global croplands, and the effects of the different regional cropping systems into a global soil erosion model. This, coupled with an improved global assessment of the global rainfall erosivity dynamics and the latest globally consistent dataset paved the path towards a state-of-the-art global RUSLE-based model.

The results of this study shed light on the impacts of the 21st century global land use change on soil erosion, providing insights into the potential mitigating effects attributable to conservation agriculture. The strong bond between remote sensing and inventory statistics formed the basis for globally consistent characterizations of soil erosion with local importance and utility (Fig. 5). The knowledge derived from this global assessment can thus improve our understanding in both global and regional land degradation dynamics and forms an important starting point to develop concepts for a better management of the land and an effective mitigation of land degradation.

The limited availability of globally consistent data on the amount of cropland under conservation agriculture constrained the ability of our study to comprehensively model the mitigating effects for all the 202 countries under observation. To date, 54 counties have provided statistics about their conservation agriculture practices to the FAO, which cover about 73% of the global cropland surface. In the conservation scenario, these countries experience 7% less soil erosion than in the baseline scenario, where no conservation practices are considered. Assuming the

average conservation practices at continental level as representative for the remaining 27% of the global cropland, for 2012 a total annual average soil erosion of $17^{+1}_{-0.7}$ Pg yr$^{-1}$ is predicted. The confidence intervals refer to the variation between the conservation and baseline scenarios (superscript) and the conservation scenario assuming the maximum technical efficiency of the employed conservation practices (subscript).

Previous global estimates of soil erosion on agricultural land span across two orders of magnitude (23.7 and 120 Pg yr$^{-1}$). The most cited estimate of global soil erosion in agricultural land by Pimentel et al.[22] is equal to 73.5 Pg yr$^{-1}$. Citations on global soil erosion estimates during the last three decades estimated around 75 Pg yr$^{-1}$ of soil eroded in cropland[19,23]. Early RUSLE-based spatially distributed modelling approaches confirmed this value[36,43]. However, recent studies using methods that more closely link models to measured erosion values report smaller global erosion rates. By means of a combined plot data and RUSLE modelling extrapolation approach, Van Oost et al.[13] estimated a soil erosion amount of $33.4 \pm 4.4$ Pg yr$^{-1}$ for global agricultural land (pasture and cropland). More recently, Doetterl et al.[37] constrained a simulation of a coarse resolution global application of the RUSLE model (ca. $10 \times 10$ km cell size) with the soil erosion data reported for the US cropland. This resulted in an estimated soil erosion by water in global cropland of $13.1 \pm 6.6$ Pg yr$^{-1}$. Since RUSLE models do not include a description of gully and tillage erosion processes, and also do not represent other geomorphic processes such as landslides and river bank erosion, it is reasonable to assume that their estimates fall into the lower end of the 23.7 and 120 Pg yr$^{-1}$ range of cited soil erosion estimates. The new cropland soil erosion estimate of $17^{+1}_{-0.7}$ Pg yr$^{-1}$ for the year 2012 that we present in this study is consistent with the recent estimate of Doetterl et al.[37]. The good correspondence of our results (without using constraining factors) with regional estimates (US and Europe) and Doetterl et al.'s[37], supports the hypothesis that soil erosion due to sheet and rill processes is smaller than previously assumed in literature.

The estimates reported in this study rest on RUSLE, a deterministic and empirical-based model which was developed based on a statistical analysis of more than 10,000 plot-years of basic

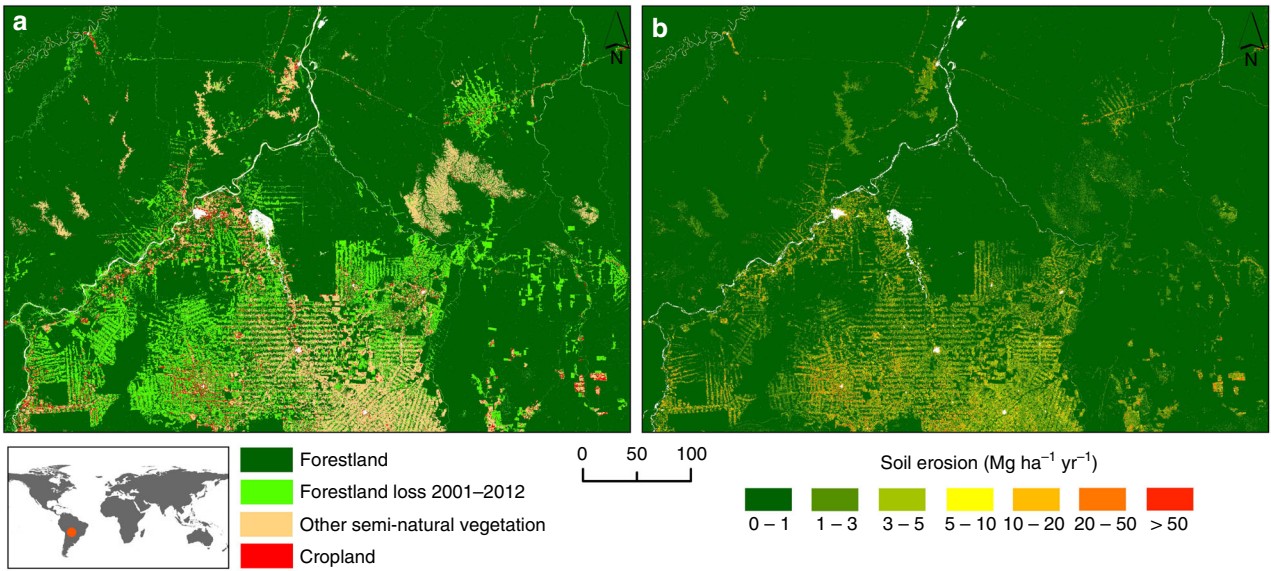

**Fig. 5** Examples of the local relevance and utility of the global soil erosion estimates. **a** Representation of the land use/land cover data employed in this study. The area reported in the image is a region of Mato Grosso in Brasil. The light green indicates forest loss between 2001 and 2012. **b** Representation of the high spatial detail of the soil erosion model predictions, capable to represent the effect of the forest loss in the Mato Grosso

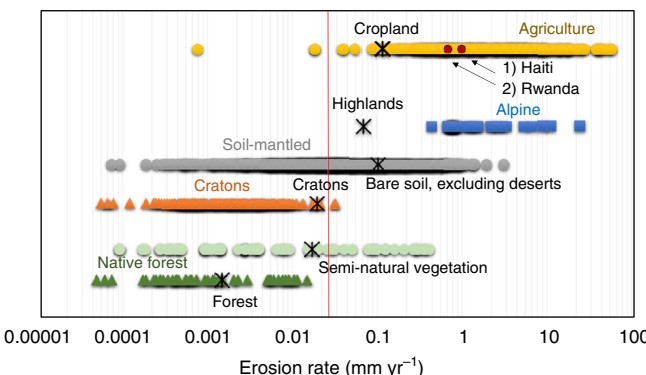

**Fig. 6** Comparison of measured and modelled erosion rates. Representation of soil erosion rates measured on agricultural fields under conventional agriculture (n = 779), geologic erosion rates measured on alpine terrain (n = 44), soil-mantled landscapes (n = 1456), low gradient continental cratons (n = 218), grassland and scrublands (n = 63), native forests (n = 46) and averages of our predictions (indicated by an asterisk). Large parts of the measured data come from the study of Montgomery[8] integrated with data from other meta-analysis studies. The vertical red line indicates average value of soil erosion. The red dots refer to averages soil erosion rates modelled for two country highly susceptible to water erosion (Haiti and Rwanda)

runoff and soil loss data[44] in 49 US locations covering a large variety of landscape conditions. Although RUSLE-based models are derived from the most comprehensive set of measurements available[45] including universally recognized factors that affect soil erosion by water[29,33], they are predominantly built upon parameters that result from experiments conducted in the United States[45]. The application to a non-plot-level and in areas outside the range of the original estimates (e.g., tropics, subarctic and tundra) may substantially reduce the accuracy of the model[45]. The authors recognize that using an empirical-based prediction tool outside the original range of environmental variables could represent a legitimate concern[46]. Considering the proven capacity of RUSLE-based models to overcome their empirical origin[47], the current lack of better performing models[9,31], and the need for

predicting the possible impacts of global change upon soil erosion[27], the authors argue that at this stage the presented global RUSLE application represents a legitimate approach to narrow the current gap of knowledge and support the targeted soil conservation efforts aiming to mitigate soil erosion.

Given the quantitative and harmonized nature of the data set, there seemed to be no reasons to doubt the consistency between the estimates for the two time periods as well as the reliability of the resulting national trends. The difference obtained from the comparison of the estimates for the two time periods was driven by land use change and was unaffected by the predictive limits of the empirical soil erosion model. Validity, i.e., if the model accurately measures the amount of displaced soil, could have been an issue as the predictive capacity of both empirical and process-based physical models is still limited[9,31,46], and because the model was run on a global scale based on a number of data-driven assumptions with soil erosion quantities estimated for each of the 2.9 billion cells of the global raster.

As observed by Auerswald et al.[48], a validation sensu strictu of USLE-based modelling at regional or larger scales is not feasible due to the lack of long-term field-scale measurements. Therefore, a cross-comparison of the modelling results to gain insights on the validity of the modelling predictions was performed. This operation shows that the modelling results are consistent with both empirical observations and other regional soil erosion assessments. The analysis at meta-data level confirms that our estimates fall in the range of measured data collected by Montgomery[8], as well that the global model can describe the magnitude of soil erosion incurring between the different land cover types. Adapting the figure he created (Fig. 6), we superimposed the results from our global analysis for different land covers, to which we added data on native forests and data from other meta-analysis studies (Supplementary Note 5). In addition, an exiguous deviation was observed from the comparison between our estimates and the ones provided by independent studies of the US Department of Agriculture (USDA) for cropland in the United States[30] (4.5%) and by the Joint Research Centre (JRC) of the European Commission for cropland of the 28 EU countries (1.1%). The good agreement between our estimates and the ones provided by independent studies give confidence that the

quantitative estimates achieved through the global model are reliable and valid to a level close to these higher resolution regional assessments (Supplementary Note 5). Further insights supporting the validity of the global model were achieved comparing the spatial patterns of the estimates with the ones reported by previous UN funded global assessments based on expert judgement (GLASOD)[5] and remote sensing data (GLADIS) (Supplementary Fig. 4 and Supplementary Note 5). On the basis of a sensitivity analysis (Supplementary Note 3), the authors observed that the soil erosion predictions of the global RUSLE-based model were most sensitive to the cover-management factor (C-factor) (Supplementary Figs. 5 and 6). This supports the hypothesis that a thorough definition in the C-factor of the land uses and their changes, the extent, types and the spatial distribution of the global croplands and cropping systems are the key to improve RUSLE-based global assessments. In addition, the sensitivity analysis allows us to define the influence of the input parameters on the global RUSLE predictions and spatially map their effects on the model output (Supplementary Fig. 7).

The uncertainty of the spatial predictions was estimated using a Markov Chain Monte Carlo (MCMC) approach (Supplementary Note 4). The map of uncertainty is presented in Supplementary Fig. 8 as the standard deviation of the MCMC simulated values. The map gives an outline of the geographical distribution of the prediction variance, and it can be used to compare the potential error in different areas of the world. The error of the model estimates associated with the input data assessed with a MCMC approach is about 8 Pg yr$^{-1}$ for the whole world. Accounting for uncertainties in the soil erosion rates, we estimated an annual average potential soil erosion amount of $35^{+5.6}_{-2.4}$ and $35.9^{+5.6}_{-2.4}$ Pg yr$^{-1}$ for the 2001 and 2012 baseline scenarios, respectively.

It should be noted that the positive results highlighted by the cross-check analysis do not imply that the global model captures reality by 100%. The authors recognise that the modelling based on data-driven assumptions has its limitations, and there is a need for field monitoring and local scale process-based modelling. However, in light of the useful insights gained from the operations of the model performance evaluation, we argue that the presented RUSLE-based global approach constitutes a powerful assessment tool for identifying hotspots and areas of concern at the global scale. It provides the basis for a more strategic approach in directing new monitoring/modelling efforts and informing decision making for the development of policy. Moreover, in the proposed new form, the global scale RUSLE-based assessment is brought to a new level, as for the first time it links to the key parameters required to assess the effects of global change and support conservation planning and land management. Hereinafter, we therefore discuss the implications of our global modelling from a multidisciplinary perspective linking the findings of our map to GDP measures to identify potential pressures on food production systems, risks of increased food and feed prices due to phosphorous shortages, the global soil organic carbon (SOC) pool that forms the basis for emission levels and climate change analyses, the economic costs of soil erosion and the overall implications for policy decision-making and sustainable development goals.

The increasing population places greater pressure on global food production systems. The global spatial coverage of modelled soil erosion enables us to explore the relationship between the average soil erosion in croplands and the GDP of each country based on World Bank figures. The results presented in Fig. 7 also show latitude according to size. Clearly, the wealthy countries in temperate latitudes have the least erosion with poorest tropical countries being the most susceptible to high levels of soil erosion. The countries that can least afford soil protection measures are the most vulnerable. This emphasises the importance of soil

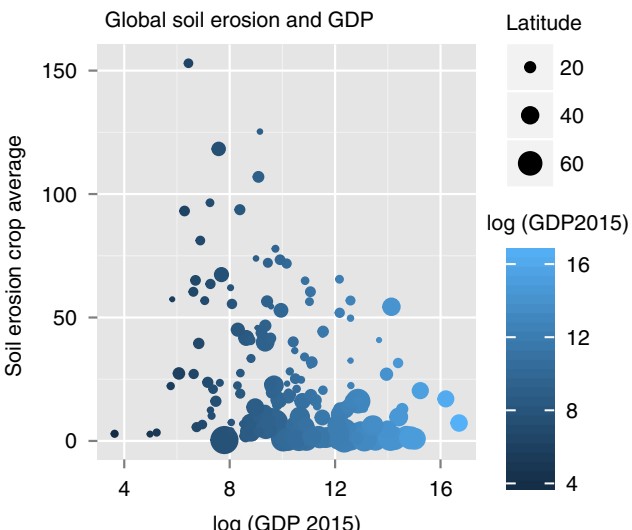

**Fig. 7** Soil erosion in cropland areas on a country basis vs. the log of the respective countries GDP according to the World Bank. The size of the circle represents the latitude indicating the higher latitude, wealthier countries are the least impacted by soil erosion, either through more favourable climatic conditions, or soil erosion prevention measures

protection in the sustainable development goals if there is any hope of intensifying agriculture in these countries to meet the food needs of the populations.

Along with the loss of fertile soil through erosion as quantified in this study goes the imminent threat of limited nutrient resources. In 2009, clean phosphorous reserves were predicted to run out in only 20–50 years[49,50]. Even though this prediction was revised only 2 years later with previously overlooked phosphorous reserves found in Morocco and Western Sarah[51], the finite nature of resources could become a source of political tension especially in developing-world countries where farmers cannot afford phosphate fertilizers even at today's non-monopoly prices. With continuously rising demand due to the increasing world population and thus higher demand of food in general and livestock products in particular, fertilizer prices are likely to increase. This may encourage companies to explore new reserves from lower grade rock which is subject to a higher cadmium pollution and exacerbate the conflict with the less and least developed countries due to food and feed shortages. Both developments could impede sustainable soil use and the application of soil conservation management practices even further. The former development may increase the cadmium pollution of soils and potentially restrict soil usage. The latter, in turn, could lead to an even more intensive land use with negative effects on soil erosion rates. No substitutes exist for phosphates (with the exception of organic farming using manure which is limited by livestock availability in many countries). In this regard, one promising but not yet widely discussed approach could be to protect phosphorous by reducing soil erosion rates. Cordell et al.[50] estimated that around 36% of the total phosphorus fertilizer applied to arable land was directly lost due to erosion. Our global soil erosion assessment highlights the areas where agricultural management based on sustainable farming practices with low soil erosion and high phosphorous recycling rates could be most effectively applied to help keep global food and feed prices at reasonable levels.

The prediction of the global soil organic carbon (SOC) pool by Earth-system models is still one of the main sources of uncertainty, undermining the confidence in the carbon (C) budget and

its future projections[52,53]. Poor representation of different mechanisms driving SOC turnover and low accuracy of soil data inputs are among the primary causes of this uncertainty. Land-C-atmosphere feedbacks may not be properly disentangled as long as relevant missing processes are not implemented. Among those, soil erosion is certainly a key process as it displaces consistent amounts of C as lateral fluxes, then subjects it to different environmental conditions that control its stabilisation and release. The consequences of neglecting this component by large-scale modelling and inventories are still uncertain, so that erosion is estimated to induce a carbon sink or source up to 1 and −1 Pg C yr$^{-1}$ globally[54], respectively. In this respect, the new global soil erosion assessment presented in this paper has the potential to become a reference input for integrating later C fluxes into large-scale model frameworks of different complexity[40]. Combining the global soil erosion with a recent SOC map (Supplementary Methods), we estimated a gross SOC displacement by soil water erosion on the order of 2.5 Pg C yr$^{-1}$. Thirty-six percent came from agricultural land, although this covers only 11% of the total area investigated. Geographically (Supplementary Fig. 9), the regions across the Tibetan plateau and China were extensively exposed to high rates of SOC displacement, as also the north-eastern part of Siberia.

Several on-site effects of soil erosion which occur directly at the site where the soil is removed have been cited in the literature[12]. Increasingly, scientists also mention offsite effects of soil erosion in the surrounding areas[55,56]. Given these on- and offsite effects, soil erosion assessments are highly relevant from an economic point of view because erosion is associated with an unequal distribution of economic costs. It degrades soil and thereby reduces soil productivity and yields of the land due to the loss of fertility and water storage capacity[22,57]. Land users in economies which can afford it, therefore have an incentive to use fertilizers or water management practices which unfold immediate effects to compensate the negative effects of soil erosion on their short-term yields. In economies that cannot afford these measures, the price of erosion is paid for by reduced food or forest production[22]. The on-site costs of soil erosion are thus internalized by pricing them directly into the economic decision making of the land user. As long as cash flows remain positive, land users, by themselves, thus hardly have an incentive to adopt land conservation practices to contain soil erosion because the costs of adopting these technologies are high and occur immediately (reducing early cash flows), while the benefits spread out over a longer time horizon[55] and are discounted more heavily. From an economic perspective, the high resolution global soil erosion assessment could help to estimate the global costs of these on-site effects especially from a long-term land value perspective. More precisely, the monetary costs of fertilizers, additional water management, etc. or productivity losses in the poorer countries could be contrasted with the cost of soil conservation practices to reduce soil loss in the first place in order to derive appropriate political regulations for sustainable on-site land management. Soil erosion, however, also affects the surrounding areas with off-site costs that are external to the future cash flow calculus of the land user. Here the price mechanism breaks down in that the offsite costs caused by sedimentation, siltation, eutrophication, flooding, etc. are not internalized into the land user's investment calculus but are borne by society, i.e., the tax payer. While in the short-term, the exploitation of soil resources may be economically sound for land users, this may not be the case from a socio-economic perspective because the society bears the offsite costs (e.g., cleaning of public waterway infrastructure, prevention of dam bursts) while consumers may be confronted with higher prices. In the case of off-site effects, the global assessment of soil erosion is especially beneficial as it provides the long-awaited basis for an economic assessment of the off-site costs of soil erosion on a global scale. An economic assessment, similar to Pimentel et al.[22] but based on a global assessment with high resolution soil erosion information could assign a value to on- and off-site costs of soil erosion, locally as well as globally, in order to calculate the net costs of soil erosion. Placing a value on the on- and off-site effects may help decision-makers to internalize the off-site costs into the investment calculus of the land user, e.g., in the form of obligatory soil conservation practices in areas with high off-site costs otherwise borne by society. This is important because fertile soils on the planet are limited and essentially non-renewable, at least on human time scales. It is therefore of crucial importance to protect available soil resources from further degradation if society wants to maintain this precious natural resource for future generations. In this sense preserving soil quality and achieving a land degradation neutral world have been explicitly recognized in the recently approved sustainable development goals (SDG). Goal 2 explicitly mentions the relevance of maintaining soil quality for achieving food security while Goal 15 calls for a land degradation neutral world by 2030. These goals can only be achieved if we are able to limit current soil erosion rates by applying sustainable soil management practices especially in the areas mostly affected by erosion processes. Moreover, the results of this study can also be relevant for Goal 13 aiming at taking action to combat climate change and its impacts[12,13] and Goal 6 to ensure availability and sustainable management of water. The recently endorsed FAO Voluntary Guidelines for Sustainable Soil Management[58] provide the necessary guidance to National governments on the way forward in order to achieve such ambitious goals by 2030. The insights of our high resolution global modelling approach can provide a solid starting point to support decision-making in both ex-ante and ex-post policy evaluation, while scientifically, it can enable better estimates of the global SOC pool including the effects of land use change and conservation agriculture. Our findings can also provide the basis to test the possible effects of the four per mil initiative proposed by the French authorities during the COP21 in Paris.

## Methods

**Study area.** The study area includes the area of the 202 countries for which FAOSTAT[14] provides crops statistics. The total modelled area is about 125 million km$^2$, providing living space for a population of about 7.5 billion people.

**Soil erosion modelling.** The years 2001 and 2012 form the reference periods to assess the 21st century human-induced soil erosion by water erosion at a global scale. For 21st century human-induced soil erosion we refer to the effects caused by land use/land cover changes. Permanent loss and gain of global croplands, forests and semi-natural grass vegetation are considered in the modelling scheme while the effects of grazing and the establishment of new pasturelands are implicitly reflected. Short-term effects of land use/land cover change (i.e., forest/rangeland fires and wood harvesting) and overgrazing are not modelled. Climate change and human-induced effects on climate are also not considered.

The long-term annual soil erosion rates for the two different land covers are estimated using an improved large-scale Geographic Information System (GIS) version of the Revised Universal Soil Loss Equation (RUSLE) model[33] (Supplementary Fig. 1). RUSLE belongs to the so called detachment-limited model types where the soil erosion (expressed as a mass of soil lost per unit area and time, Mg ha$^{-1}$ yr$^{-1}$) due to inter-rill and rill erosion processes is given by the multiplication of six contributing factors. Consistently with the predictive capacity of the model, soil displacement due to processes such as gullying and tillage erosion is not estimated.

RUSLE-type models have demonstrated to be able to reduce a very complex system to a quite simple one for the purposes of erosion prediction[9] while maintaining a thorough representation of the main environmental and anthropogenic factors that influence the process[33]. Conceptually, these models follow the same principle of complex process-based models, with a driving force (erosivity of the climate), a resistance term (erodibility of the soil) and the other factors representing the farming choice, i.e. topographical conformation of the field (LS), cropping system (C) and soil conservation practices (P). Field- and catchment-scale experiments that compared the prediction capacity of empirical RUSLE-type models with process-based models did not reveal a substantial

difference between the predictive performances of the two modelling approaches[9,31]. This, together with the moderate data demand of RUSLE-type models has facilitated the process of upscaling. Today, an extensive amount of the literature recommends the use of RUSLE-based models to provide spatially explicit estimations of soil erosion in GIS environments.

In the original version of RUSLE[33], the region of the model simulation is a specific field plot slope with given size, rainfall pattern, soil conditions, topography, crop system and management practices. Before the introduction of GIS-based computational techniques, the input data employed to run the model were generally directly measured in the field and afterwards imported in a specific software. In this study, a simplified application of the RUSLE model at global scale is proposed. For this purpose, the approach paved by previous GIS-based studies dealing with upscaling procedures to extent the applicability of the model as well as regional studies in Europe[59], Australia[60] and China[61] was followed. The rates of soil displacement by water erosion are estimated through the GIS raster scheme. Using a GIS raster scheme applied to the USLE model means hypothesizing that each cell is independent from the others with respect to soil erosion. Soil erosion (synonym to RUSLE soil loss, Supplementary Note 2) refers to the amount of sediment that reaches the end of a specified area (cell) on a hillslope that experiences loss of soil by water erosion. The modelled area does not, in any way, include areas of the slope that experience net deposition over the long term. This is because the displaced soil amount is not routed downslope across each cell from hillslopes to the sink area or the riverine systems through a transport/deposition capacity module.

Conceptually, the global modelling presented in this study is based on the assumption that if a catchment- or a regional-scale application of RUSLE can predict meaningful soil loss estimates, the application of the model at larger-scale will provide meaningful estimates as long as the scale of the input data employed is congruent with the scale used for the estimation of the modelling factors and local applications. If the scale of the input data meets these requisites, the global scope of the model application represents a consistent repetition in space of the calculation for all cells in the modelled area.

Although the input factors show a lower spatial detail in this global-scale study compared to the recent study in Europe[59], the data and the scale used are adequate to ensure a spatial description of the modelling factors and are therefore within the boundaries of the model's applicability. The characterization of land use/land cover change is based on NASA's Moderate Resolution Imaging Spectroradiometer (MODIS), which are the most consistent data currently available for soil erosion modelling at global scale[62]. With a cell size of $250 \times 250$ m spatial resolution, the dimension of each individual land unit (equal to 6.25 ha) fairly describe the general dynamics of landscape fragmentation and can adequately represent the land unit originally targeted by the model, i.e., the US arable fields which show an average size of about 400 acres (162 ha). For the analysis of the effect of the topography a SRTM 3 arc-seconds (ca. 90 m) spatial resolution was used. This ca. 90 m DEM ensured the computation of the combined topographical factor maintaining a scale congruent with the one used during the USLE's experimental measurements with plots having a length ≤122 m. Rainfall and soil characteristics were obtained using the best database available to adequately describe their pattern and dynamics during the elaboration of this study. Both have a spatial resolution of ca. 1 km.

For processing the main model components, the spatiotemporal variations of the land cover and management (C-factor) as well as the rainfall erosivity (R-factor) factors were assessed through new methodologies exclusively designed for this study while the K, LS and P-factor were derived from methods reported in literature (Supplementary Methods).

The C-factor measures the combined effect of the interrelated cover and management variables on the soil erosion process[63]. For this global assessment (Supplementary Fig. 10), we followed the procedure as paved by previous national[64] and pan-European studies[65] to assess the C-factor in a simplified but still meaningful way. We used steps for computing the C-factor in agricultural and non-agricultural land. Within a first step, the global land cover dynamics were outlined using the MODIS Land Cover Type product MCD12Q1 classified according to the International Geosphere Biosphere Programme (IGBP) system[66]. The data were downloaded via NASA EarthData facility (reverb.echo.nasa.gov) and pre-processed using the MODIS Reprojection Tool to obtain two rough global land cover maps (ca. $250 \times 250$ m spatial resolution after applying the nearest resampling method in ArgGIS) for the reference years 2001 and 2012. The IGBP scheme reports seventeen land cover classes (Supplementary Table 1), including three non-vegetated land classes, three developed and mosaicked land classes and eleven natural vegetation classes. In the following step, we spatially defined the agricultural land according to the IGBP-12 class (croplands) and IGBP-14 class (croplands/natural vegetation mosaic). Eleven classes were categorized as non-agricultural land and four classes were precluded from the analysis because they were not relevant for our soil erosion context (i.e., water, permanent wetlands, urban and built-up, snow/ice). The MODIS MCD12Q1 Land Cover Type IGBP was developed using a well-defined classification scheme applied consistently at global scale[67]. Friedl (personal communication), however, strongly advised against mapping changes by taking simple difference across years due to the overall accuracy across all land classes of ca. 75%[67] and the consequent presence of mapping spurious changes. Following this advice, the subsequent step of data processing for the MCD12Q1 global land cover maps was modified in order to better represent the forestland area by considering the well-known inability of the

MCD12Q1 product to accurately detect forestland in some locations[67] and align the national cropland surface with the more accurate estimations reported by the Food and Agriculture Organization (FAO). The global forestland area for the years 2001 and 2012 was outlined using the high-resolution global forest change maps derived by Hansen et al.[25] based on Landsat imagery. Original data at 30 m spatial resolution were downloaded from the online database of the University of Maryland and resampled at 250 m. According to Hansen et al.[25], for the forest gain and loss a tree cover >50% was considered. Thus, the forestland class of the MODIS MCD12Q1 for the year 2001 was replaced by Hansen et al.'s[25] data. Discrepancies between Hansen et al.[25] and MODIS MCD12Q1's data were reclassified to an additional code 21 and renamed as transitional woodland-shrub. To delineate the cropland for each of the 202 selected countries, we capitalized on both the spatially explicit information offered by the MODIS MCD12Q1 product and the detailed cropland statistics reported at national level in the Food and Agriculture Organization's (FAO) FAOSTAT database (http://www.fao.org/faostat/en/#data). Considering the FAOSTAT data as the best estimation of the real cropland extent, a targeted removal or addition of the cropland area (IGBP-12 and IGBP-14 class) from the MODIS 2001 (after the forest adjustment) was performed to obtain a cropland area consistent with the FAOSTAT data for each of the 202 considered countries. The FAO-based cropland surface (arable land and permanent crop area) for the years 2001 and 2012 was obtained by calculating the median of the FAOSTAT data for the triennial periods 2000–2002 and 2011–2013. The MODIS cells to be replaced were selected using the auxiliary information provided in the MODIS MCD12Q1 product (i.e., classification confidence and second most likely IGBP class at each cell) which allowed us to perform a targeted replacement based on the classification confidence of each pixel. The modified MODIS global land cover map of 2001 was used as a baseline for the map of 2012. The final map for the year 2012 was obtained by substituting Hansen et al.'s[25] forest data of 2001 with the one of 2012, calculating the cropland areas replaced by forestland and modifying the cropland surface of each country consistently with the two medians obtained from the FAOSTAT data.

For the countries that experienced a reduction of the cropland surface during the observed period, the number of cells equal to the cropland decrease which presented the lowest classification confidence in the IGBP-12 class of the MODIS MCD12Q1 2012 were removed from the land cover map of 2001. The new cropland established between 2001 and 2012 was obtained selecting the cells of the IGBP-12 class of the MODIS MCD12Q1 2012 with the highest classification confidence.

To outline the cropland maps for the two reference periods, twelve-year crop statistical data (harvested area) were used to statistically describe the national crop conditions and to estimate the C-factor ($C_{CROP}$). A set of 170 crops were considered[15] and subsequently grouped in 14 crop classes according to their soil cover effectiveness (Supplementary Data 1 and Supplementary Table 2). The cropland data provided by FAO at a national level were regionalized for each of the 3247 FAO sub-national administrative unites (ADM1 level) (Supplementary Fig. 11) by means of statistical downscaling operations using the global harvested area data of Monfreda et al.[68] to describe the local crop conditions and compute C-factor values ($C_{CROP}$) representative of the local farming systems (Supplementary Methods). The C-factor for the non-agricultural land was defined for each of the fifteen IGBP classes by combining literature values (Supplementary Methods and Supplementary Table 1) with global vegetation metrics (MOD44B).

World patterns of rainfall erosivity (R-factor) (Supplementary Fig. 3) were spatially described by means of a relevant number of point information data and advanced interpolation techniques[69]. We calculated the long-term average R-factor based on high-resolution temporal rainfall data (5, 10, 15, 30 and 60 min) collected from 3625 distributed precipitation stations across the globe[57] (Supplementary Fig. 12). The spatially continuous R-factor map (at 30 arc-seconds, ca. 1 km) was computed using a Gaussian Process Regression (GPR) geo-statistical model.

For the remaining RUSLE factors (K-, LS- and P-) (Supplementary Methods), the highest spatially-resolved input data available at a global scale were used. We used the algebraic approximation reported by Wischmeier and Smith[63] and Renard et al.[33] to assess the soil erodibility to water erosion (K-factor) (Supplementary Fig. 13). The soil properties (i.e., texture, organic matter, coarse fragmentation) were downloaded from the ISRIC SoilGrids database at a 1 km spatial resolution[70]. Further soil properties such as soil structure and permeability were derived according to the methodology proposed for the soil erodibility map of Europe[71]. We computed the topographical factor (LS-factor) (Supplementary Fig. 14) following the approach suggested by Desmet and Govers[72], limiting the estimation of LS to a maximum slope angle of 26.6 degrees (50%). Hole-filled SRTM 3 arc-seconds (ca. 90 m) Digital Elevation Model Version 4 were employed to represent the land surface between 60° North and 56° South[73]. Extreme North latitudes were covered using ASTER GDEM v2 data products[74]. In the main modelling run the P-factor was assumed to remain constant at 1 since suitable spatial information for all the 202 considered countries were not available.

**Scenario analysis.** The soil erosion rates of the baseline scenarios were predicted for the reference years 2001 and 2012 without considering conservation cropping and management practices. We rested this decision on the current lack of adequate standardised and harmonised worldwide spatial information. Besides the baseline scenario, a conservative scenario was estimated for the 54 countries which reported information about the implementation of conservation agriculture to the FAO.

These countries regularly submit the proportion of their cropland managed in accordance with the three FAO conservation agriculture standards (i.e., minimum soil disturbance, organic soil cover, crop rotation/ association). As suggested by Wischmeier and Smith[63], practices of improved tillage like no-till and cover crops were considered as conservation cropping and management practices and implemented in the C-factor. For these areas, we assumed a reduction of soil erosion of 45% compared to conventional tillage. A subsequent model run assuming a reduction of soil erosion of 75% was performed[75,76]. This second conservation prediction refers to the maximum technical potential reduction which we use to represent the negative variation in our conservation scenario.

**Model performance evaluation**. In order to evaluate the performance of the global model and lay the groundwork for future studies to better identify areas prone to soil erosion by water and their environmental and socio-economic implications, multiple operations aiming to obtain insights about the validity of the modelling predictions were performed. These consisted of the following: a comparison of the estimates with the ones provided by independent RUSLE-based assessments in the US and Europe, an analysis at meta-data level to observe if the estimates fell in the range of measured data, and a comparison of the spatial patterns of soil erosion predicted by the global model for the years 2001 (baseline scenario) with the ones reported by the expert-based Global Assessment of Human-induced Soil Degradation (GLASOD). A further comparison was made using the land degradation trends reported by the UN project Global Assessment of Land Degradation and Improvement (GLADA), an assessment based on remote sensing time series analyses of the normalised difference vegetation index (NDVI) for the period 1981–2003. In a final comparison, the measured soil erosion rates from 2500 locations across the word were superimposed to the estimates of the global model (Supplementary Fig. 15)

**Data availability**. The authors declare that all other data supporting the findings of this study are available within the article and its Supplementary Information files, or are available from the corresponding author upon reasonable request.

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

## Acknowledgements

Funding for P.B. was provided by the European Commission Joint Research Centre in the framework of the SOIL-NACA project and by the University of Basel. We thank Ronald Vargas (FAO) and the Intergovernmental Technical Panel on Soils (ITPS) for their support and access to data. We also thank Dr. Tomislav Hengl and the World Soil Information Foundation (ISRIC) for providing the global soil information and the National Aeronautics and Space Administration (NASA) and United States Geological Survey (USGS) who have made available the MODIS data.

## Author contributions

P.B. and P.P. developed the research concepts, with L.M., D.A.R., C.A. and C.B. coordinating it. P.B. conceptualized the modelling scheme and carried out its computational steps and analyses. C.B. performed the analysis of model uncertainty and the sensitivity analysis. P.P., P.B., K.M., K.V.O. and C.B. worked on the development of the global rainfall erosivity map. K.M. performed the scripting for the global soil erodibility mapping. P.B. conducted the model cross-comparison. P.B., D.A.R., L.R.F., E.L., C.B., C.A., K.M., S.M., B.S., V.F., V.B., K.V.O., L.M. and P.P. discussed the results and contributed to writing the discussions. P.B., D.A.R. and L.R.F. wrote the paper.

## Additional information

**Competing interests:** The authors declare no competing financial interests.

