## [Peer Review File · Nature Communications]

Reviewers' comments:

Reviewer #1 (Remarks to the Author):

Dear authors,

I found your paper of interest but lacks in my opinion relevant issues that finally results in the rejection of the paper.

We must consider here that you submitted this paper to a journal that request the best science executed and presented with the highest quality and accuracy

Following are my reasons:

1. There is a lack of a proper literature review. There are key scientist not cited in this paper. Your paper is based on reports from environmental and governmental agencies and there is little references to scientific research
2. I found your paper more related to a report than a scientific paper. USLE, we know, that was developed for a certain terrain, and when applied to other regions produce some numbers, yes, but they are not accurated enough
3. I suggest to develop a deep publication review instead to re-use the data of Montgomery
4. The is a lack of discussion in your paper, and the introduction is poor, which repetitions to what is knowNature needs a alive and fresh research and in your paper there is many many information that is already know.

Please, check your conclusions or read your abstract. Those findings are not new, we know.... developing countries are the ones that suffer the problem... it is a socioeconomic issue that finally affect the biophysical world

I suggest to reject the paper because a lack of novelty. A old method under doubt if it is correct or not to apply to other regions than the Central North America rolling landscapes, a poor introduction with many flaws and lack of key previous research, and finally the lack of a proper discussion to move further the knowledge of soil erosion....

Reviewer #2 (Remarks to the Author):

Review of Borelli et al. 'Quo vadis soil erosion?...' for Nature Communications

26 April 2017

I have asked two colleagues to assist me with this review, as their very broad experience of soil erosion in the real world complements my own expertise as a soil erosion modeller. They are:

Prof. John Boardman, Environmental Change Institute, University of Oxford

Dr Bob Evans, Global Sustainability Institute, Anglia Ruskin University

Their comments follow my own.

Dave Favis-Mortlock

Dave Favis-Mortlock:

I recommend rejection of this paper.

A castle built on sand, however elegant, is still a structure built on unstable foundations. Similarly, this study's use of admirably high-resolution input data, and a well-chosen variety of land-use change scenarios, cannot hide the poor erosion modelling foundation upon which it rests. "Garbage in, garbage out" is one of the oldest catchphrases in computing, but it applies admirably here.

The USLE model, on which RUSLE is based, is a statistical model which was devised to estimate rates of erosion on small plots in the USA (most of which were situated in the mid-western states: e.g. Nicks, 1998). As with any statistical model: using it with data other than that which was used to derive the model, amounts to extrapolation. Whether this is trivial or serious depends on the degree to which the two sets of data differ. I suggest that the data used in this study, and the data used to define the USLE and its RUSLE derivative, differ tremendously. A few differences are:

1. USLE/RUSLE data refers to small plots which are hydrologically disconnected from their surroundings. This study (and other similar studies) use RUSLE on landscape tiles which are not, in reality, hydrologically disconnected from their surroundings. What about flow onto/off of each tile? Completely ignoring the connectivity of runoff and sediment, as does the present study, is a vast oversimplification of reality. This alone renders tile-based RUSLE studies, such as the present study, meaningless.

2. The USLE and RUSLE rely on an R-factor which is based on rainfall intensity (EI30). Thus low rainfall intensities imply low erosion rates. However in many parts of the world (e.g. northern Europe), erosion occurs under low rainfall intensities provided that the soil is saturated (in other words, runoff here results from SOF rather than HOF). For these areas, the R-factor is an inappropriate measure of erosivity: variants of the USLE such as MUSLE or the Onstad-Foster are better suited. This study (and other similar studies) uses only the EI30-based R-factor for all locations considered, irrespective of whether erosivity at that location is driven by rainfall intensity or amount (or a mix of the two).

3. The USLE and RUSLE do not attempt to describe any form of water erosion other than sheet or rill erosion. Gullying, tillage erosion, bank erosion, and loss of soil with crops are all ignored. While the authors of the present study do admit this on lines 302-303, this major limitation should be much more prominently stated.

4. Within the USLE database, there is a Coefficient of Variation of up to 15% for soil loss measurements from replicate plots, as pointed out by Nearing et al. (1999). Random variability will of course be higher for non-replicate plots. Thus there is a considerable error range for any USLE or RUSLE prediction. However, I see little in the present study regarding error. There should, at the very least, be an estimate of error associated with all RUSLE predictions.

5. Soil erosion by water is highly patchy, spatially. To present erosion rates as an average over the whole of each 250 x 250 m tile is potentially misleading, since erosion rates within that tile may well vary greatly. Even in locations suffering severe erosion, gullying on one part of the tile may co-exist with very little erosion on other portions of the tile. In this study, I see little consideration of how average rates for whole tiles would compare with rates observed at that tile's location.

Detailed responses

Line 72. Why choose these two previous assessments?

Line 73. "Unsuitable agricultural practices" should be added to this list.

Line 74. I dispute that the mechanics of water erosion are "well-understood". If they are, why do we not have a robust and useful model for e.g. gully erosion?

Line 98. This implies that qualitative approaches are somehow less valid than quantitative approaches. This is not the case if the quantitative approaches are based on unsuitable models.

Line 102. Why do the authors omit to mention earlier studies which show the limitations of erosion models? The GCTE Soil Erosion Network studies (e.g. Jetten et al., 1999) should certainly be discussed here.

Line 105. "the erosive power of rainfall": see my comments re. R-factor above.

Line 113. Ditto

Line 126. Which empirical observations?

Line 126. Which regional assessments?

Line 262. This implies that finer resolution input data improves the quality of results. It does not, if the modelling approach is itself flawed: GIGO (see above).

Lines 298-310: While the authors at last admit to a major limitation of their study (only sheet and rill erosion considered), they then attempt to use this limitation to "provide evidence of the comprehensiveness of their modelling approach". This is deeply flawed logic.

References

Jetten et al. 1999. Evaluation of field-scale and catchment-scale soil erosion models. *Catena*

Nearing et al. 1999. Variability in soil erosion from replicated plots. *Soil Science Society of America Journal*.

Nicks, A.D. 1998. The use of USLE components in models. In *Modelling Soil Erosion by Water*. 1998. Boardman, J. and Favis-Mortlock, D.T. (eds), Springer-Verlag NATO-ASI Series I-55, Berlin.

John Boardman:

Just finished reading this. Difficult to take it seriously based as it is on RUSLE figures.

line 90. I am willing to bet this is re-cycled from Pimental or GLASOD.

line 91. Therefore I don't think these are 'succeeding studies'

line 100. The references are to forests

line 114. Note the emphasis is on rainfall intensity again!

Line 119. I think the basic cell size in 5 ha...rather large to be averaging across!

Line 126. Where are these 'empirical observations' they are using?

line 147. T values have been discredited as a political tool with no scientific basis

line 171. What does it mean 'Classified...'?

line 199. Average cropland erosion rate (12) is ridiculously high

line 228. Conservation agriculture covers great variety of measures...very dodgy data

line 293. 'Most cited estimate is by Pimental et al.' This was conclusively shown to be very bad science and probably an order of magnitude too high (Boardman 1998, Journal of Soil & Water Conservation).

line 296. What are these measured erosion values?

line 297. Note that 'plot data' does not give use rates for the landscape as a whole

lines 299- 301. This need unpicking. the 10 x 10 km cell size is worrying; the link to the US cropland data (I think this is EPIC data at pre-determined points?) is unclear and worrying.

line 310. What is this 'previously assumed in the literature'? cf Boardman 1998

line 318. What are these 'cratons'?

Lines 331-333. Lets the cat out of the bag...problems with their approach BUT it is still a 'powerful assessment tool'. I beg to differ.

They claim somewhere that their approach enables us to identify hotspots. But Boardman (2006 Catena) already identified these, as best one can, based on empirical data.

At this point I stopped commenting though I did read the whole thing. The rest is based on the false premise that their modelling with RUSLE gives useful results.

One final comment: they group all cropland together. Rates of erosion range greatly on cropland e.g from very low rates on Oil Seed Rape (<0.5t/ha/yr) to higher rates on maize (>3t/ha/yr).

John Boardman

Environmental Change Institute

University of Oxford

Bob Evans:

Comments for Dave on Borelli et al paper submitted to Nature Communications

I would reject this paper on the grounds that a model is being used that has not been validated. You are back to the old adage "Rubbish in, rubbish out". It's all based on a non-proven model.

General comments:

Throughout the paper rates of erosion are given as if they are actual rates, not modelled estimates. Thus, the information is never qualified, it is always 'are' or 'have' instead of 'estimated to be'.

The GIS manipulation and data crunching are impressive but they do not override the fact that RUSLE is not a satisfactory way of assessing erosion. Indeed, it does not model gully or ephemeral gully erosion. There is a need to acknowledge that there are problems with RUSLE, and these problems have been described in the literature. One of them is that models are based on soil passing across a line, usually a 'waterfall' (driver) at the base of the plot. It could well be that large amounts are splashed across a line, but that is happening everywhere on the slope so the input = output except at the top of the slope.

RUSLE's validation/cross checking is not with field assessments of erosion but with other modelled assessments, the model often being USLE based.

That figures (rates, mass) given and repeated in papers does not mean that they are correct, eg acceptance of a soil loss tolerance value (what about very low rates – wash – carrying pollutants) or Pimental et al's data.

Are averages meaningful? The average rates given here are such that they imply rills and deposits would be clearly seen in the field.

Would it be better to use relative values rather than what they consider actual rates? Thus, x times more than a base rate?

Land use and land use change is the main discriminator of erosion rate. And its varying over time (increase or decrease in erosion is associated with land use change). The information on land use and land use change is interesting in its own right.

Much of what is written, especially in the Discussion, could be (and has been) written without trying to tart the statements up with spurious figures. Thus, generalisations made in the Discussion are probably sensible. Because of the dubiousness of the estimates of erosion, much of the Discussion is of little value.

Recent work funded by NERC and Defra, as yet unpublished, shows that models may not be adequate substitutes for monitored data. And Philippe Bavaye in his 2017 Ecosystem Services paper also casts doubt on the use of models and simulations.

Detailed comments:

Line 87. This paper is only of "fundamental importance" because Montanarella says so (see below for my response to that paper). And he was basing his paper on RUSLE findings (Panagos et al).

Line 107. How are estimates robust when there is no sound basis for the inputs into RUSLE? The inputs are based on plot experiments not reality.

Line 296. Modelled rates coming down as models are linked to measured erosion values, but these values are from plots and still don't represent reality. And are probably still out by at least on order of magnitude.

Line 321. Predicting rates for alpine areas, RUSLE not set up for such landscapes.

Line 326/7. But that is a 17 times difference!

Line 331/2. Acceptance of need for field monitoring – good. But they do not try to relate monitored data to their model, and there was certainly information on that by the time of submission, mine and John's papers in Environmental Science and Policy for starters. But the comprehensive approach taken here does not make up for poor quality of input and output data, and much of that is based on modelled data.

Line 581. Interesting that quote John and Jean's book but do not draw attention to the chapter on models.

Line 765. What does 'delta' mean?

I tried to check Reference 19 but was told 'page not found'.

Notes that spatial resolution is 250m, in fact for soils is 1km.

In Supplementary results it notes "no reason to doubt the reliability of our model'. There is. And "Validity assessments can be challenging" and hence it would seem can be ignored. Thus their comparisons of rates of erosion, as noted above, are with rates from models with a similar basis. That is not validation, but a form of calibration.

[Redacted]

Robert Evans

Global Sustainability Institute

Anglia Ruskin University

Reviewer #3 (Remarks to the Author):

Quo Vadis soil erosion ?Global Impacts of 21st Century land use change

Authors : Pasquale Borelli et al

Reviewer: Prof Gamini Herath, Monash University, Malaysia

Report

This paper is an excellent paper on a very difficult but timely topic of soil erosion. Considerable amount of work has gone in to the paper and these examine very critical issues on soils at a time when natural resources are being devastated by unlimited use.

The issue of conservation farming and damages due to traditional farming techniques is highlighted but some focus on how poverty affects soil erosion may be needed (may be one para).

The discussions is excellent and comprehensive. It is very significant because the authors try to refer to the SDGS which is the most recent institutional agreement for a sustainable planet. The authors refer to goal 2 (zero hunger) and 15 (land 0 and goal 15(life on land). I believe that other goals are related too eg. Goal 6 –clean water and goal 13-climate change and some reference to these may be valuable.

If we take specific countries, there may be some anomalies. The paper does not refer to very poor countries in Asia except India and China. Can the paper add some specific countries and soil erosion. This may be difficult due to the macro nature of the study but may add to the paper if possible.

I suggest publication of this paper with minor revisions.

“*Quo vadis* soil erosion? Global impacts of 21st century land use change”

- Response to reviewer comments -

We would like to thank all three referees for reviewing the manuscript ‘*Quo vadis* soil erosion? Global impacts of 21st century land use change’ (NCOMMS-17-05657) and for providing constructive suggestions and criticism. We have carefully reviewed their comments and have revised the manuscript accordingly. The comments on the accuracy of the global model results were especially useful and have helped us to improve the presentation of the model results.

Based on the reviewers’ comments, we have conducted additional model simulations, comparisons with other global assessments, a sensitivity analysis and adjustments in the presentation of the RUSLE model and its evaluation to enhance the scientific deepness of our study:

1. We modified the introduction to acknowledge the research activity of the GCTS Soil Erosion Network for the development of process-based physical models (Reviewer #2). We also substantiated our decision about the selected soil erosion model. With regard to the request for a more extensive literature review (Reviewer #1), we consulted two additional experts with more than 20 years of experience each in field measurements and multi-scale RUSLE-based modelling (Prof. Vito Ferro and Prof. Vincenzo Bagarello) to improve and broaden the literature review and improve the consistency of the manuscript. The introduction was enhanced by about 1,400 words and 40 additional citations of well-recognized papers. Given that our manuscript is not a review paper per se, this additional review material is provided as Supplementary Note 1.
2. We also modified the discussion chapter to openly deal with the limits of the model. This addresses the concerns of Reviewer #2 that the model application to non-plot-level and in areas outside the range of original estimates (Reviewer #1) could substantially reduce the accuracy of the model prediction. We also added quality to the discussion of our model outcomes.
3. The cross-comparison of the modelling results was combined with a sensitivity analysis to evaluate the influence of parameter estimation on uncertainty and the effects of the individual parameters on the model output (Reviewer #2). In addition, we integrated a comparison of the spatial patterns of our global model to the ones reported by previous UN funded global assessments based on expert judgement (GLASOD) and remote sensing data (GLADIS). Extensive descriptions of all cross-comparison activities of the modelling results are provided in the new Supplementary Discussions. A new paragraph in the methods sections was also added to address the requests of the Reviewers.
4. For the avoidance of doubt, in the revised version of the manuscript sentences such as ‘*The less developed economies show the highest annual soil erosion...*’ are not modified in ‘*The less developed economies show the highest predictions of annual soil erosion...*’ (Request of Reviewer #2 Colleague No. 2). The terms such as ‘prediction’ and ‘estimation’ have been consistently included in all similar sentences. More examples are provided in the reply to Reviewer #2 Colleague No. 2.
5. In the revised version of the manuscript and in the Supplementary Note we argue that although the deterministic and empirical-based model used in our study has certain limitations in its accuracy (Reviewer #1), comparisons between USLE-based models with process-based models indicate at least equal prediction capacities (details in the point-by-point reply). In light of this and the currently unavailable input data required to run process-based models at global level, we argue that the model presented in this study represents a valid tool for identifying hotspots and areas of concern. We perceive that it is better than GLASOD (based on heterogeneous expert judgments of the 1980’s) and that it provides the basis for a more strategic approach in directing new

monitoring/ modelling efforts and preparing policy decision-making. In its new form, the global scale RUSLE-based assessment is brought to a new level linking it for the first time to the key parameters needed to understand the effects of global change and to support conservation planning and land management.

6. Further reference to the UN Development Goals were added (Reviewer #3). Supplementary Tables including descriptive statistics of soil erosion prediction at country level have been prepared (Reviewer #3).
7. Data availability and transparency. All input data, intermediate database and results as well as final results will be made freely available for download on the European Soil Data Centre (ESDAC) web platform and/or on an infrastructure of Nature (e.g., Scientific Data – Nature).

The comments and critical questions of the three reviewers helped us to significantly improve the quality of the manuscript. We trust that our revisions lead to the acceptance of our paper for publication. Please be assured that we are grateful for any additional suggestions or further comments.

Next, you will find our point-by-point response to each of Reviewer's comments (*italic, underline text*).

Sincere regards on behalf of all the authors,

Pasquale Borrelli

Reviewer #1

Comment 1. *There is a lack of a proper literature review. There are key scientist not cited in this paper. Your paper is based on reports from environmental and governmental agencies and there is little references to scientific research.*

Reply 1. In the first version of our manuscript 64 studies that are highly relevant for the topic were cited. Please note that thereof 50 citations referring to articles published in internationally peer reviewed scientific journals. This equals to about 78% of the total citations. More than 20% of our citations refer to studies published in high impact factor journals such as *Nature* and *Science*.

With that said, we recognized that additional literature information offering a more comprehensive overview on existing studies could further improve the overall quality of the manuscript. Please note that in the revised version of the manuscript i) the introduction was modified, ii) the research activity of the GCTS Soil Erosion Network was acknowledged, and iii) a further extensive literature review was provided as Supplementary Note. We consulted two well-recognized experts with many years of experience in field measurement and multi-scale RUSLE-based modelling (Prof. Vito Ferro and Prof. Vincenzo Bagarello). In the Supplementary Note, we dealt with the concept of soil erosion modelling, the categories of existing models and studies referring to the use of RUSLE-based models outside the United States (US). We added ca. 1,400 words and cited around 40 additional studies to enrich the quality. The decision to include this new text in the Supplementary Information rest on the fact that the word limit was reached in our revised version the manuscript.

Comment 2. *I found your paper more related to a report than a scientific paper. USLE, we know, that was developed for a certain terrain, and when applied to other regions produce some numbers, yes, but they are not accurated enough.*

Reply 2. While we well respect the freedom of option we do not agree with your first statement. The scientists who contributed to this study cumulatively published more than 900 scientific articles in international scientific journals. This is an indication for the well-founded experience and scientific reputation of the research team. According to our understanding, a scientific report documents monitoring data, reports on the application of well-established methods or affirms a previously reported conclusion. In contrast, however, our paper presents new scientific advances, originality and novelty on the reported research in that we evaluate the state-of-the-art on global soil erosion assessments, identify a research gap and proposed an innovative method to narrow this gap providing harmonized and spatially consistent information.

With regard to the question of accuracy, we would warmly welcome a debate how much uncertainty is generally acceptable in science and in soil science specifically. Apart from thermodynamically closed systems in physics almost no scientific measurement can be made without error (Mulligan and Wainwright, 2013). This is especially true for extremely heterogenous systems like soils. In a study conducted in Kingdom City, Missouri, Wendt *et al.* (1986) compared the results of 40 erosion plots all treated identically. They found a great variation in soil erosion rates (between 18% and 91%). These scientists concluded '*The relatively huge amount of unexplained variability shows that several replications of treatments are needed to confidently estimate mean runoff or soil loss for comparison purposes.*' Rüttimann *et al.* (1995) conducted a statistical analysis of data from four different sites, each with five to six reported treatments (each treatment had three replications). They noted that the coefficients of variation of soil loss ranged

from 3.4% to 173.2% (average 71%) and suggested '*as many replications as possible*' for erosion assessment. Analyzing replicated plot pairs for 2,061 storms, Nearing *et al.* (1999) observed a coefficient of variation ranging from 14% to nearly 150%.

Model predictions are intrinsically more prone to errors than measurements (Wainwright and Mulligan, 2013). If one accepts the conclusion of studies dealing with replicate analysis that replication measurements enhance the quality the same should also hold for choosing an empirical model. Scientists should prefer models resting on extensive field measurements. USLE/RUSLE models rest on empirical observations based on more than 10,000 plot-years of basic run-off and soil loss (Lafren and Moldenhauer, 2003) data in 49 US locations covering a large variety of landscape conditions. This is the largest data collection ever made. Nearing *et al.* (2000) observed that '*Despite the limitations of the USLE and its successor, the revised USLE (RUSLE), there exists today no other environmental technology that is based on a larger and more comprehensive set of measurements.*' While the authors referred to a specific criticism made for the US, we can extend this concept worldwide. In absence of other empirical models based on large sets of landscape conditions, RUSLE, globally represents the best empirically based model currently available. The US Department of Agriculture (USDA, 2017a) estimates that RUSLE has been used in more than 100 countries to guide conservation planning (USDA, 2017b).

Excluding the feasibility of performing global-scale field monitoring campaigns, the only alternative to RUSLE are process-based models. However, as observed by Jetten *et al.* (2003) when analysing the results of the model tested by the GCTS Soil Erosion Network, Boardman and Favis-Mortlock (1998) noted that '*more complex, physically based models do not necessarily perform better than lumped, regression-based models, mainly because input errors increase with increasing model complexity.*' Favis-Mortlock (1998), comparing result process-based model (WEPP, Lafren *et al.* (1997)) with USLE-based models (Wischmeier *et al.*, 1978) such as EPIC (Williams *et al.*, 1983) and GLEAMS (Leonard *et al.*, 1987) for a field site in the US, observed no significant difference in the model performances. As reported by Quinton (2013), Morgan and Nearing (2000) compared the predictive performance of USLE, RUSLE and WEPP and observed that the process-based model (WEPP) did not outperform the empirical models. For this analysis, 1,700 plot years of data from 208 natural plots were considered. Stolpe (2005) applied RUSLE, EPIC and WEPP at the Mediterranean climate of Chile. His results showed that RUSLE and WEPP models provide the same accuracy in soil loss estimation.

Nearing (2013) notes that '*the disadvantage of the process-based model is also the complexity of the model*' and that '*data requirements are huge.*' The unattainable data requirements and the current lack of research on upscaling such mechanistic models, usually preclude the application at anything above field or catchment scale (De Vente and Poesen, 2005). In the book 'Environmental Modelling Finding Simplicity in Complexity', Wainwright and Mulligan (2013) state that '*simple models that can illustrate emergent behaviour are useful exploratory tools, and can illustrate the dominant controls on different parts of the environmental system.*'

The large amount of plot measurements, together with the simple model structure and the low data requirements, compared with mechanistic models such as WEPP and EUROSEM, has made USLE/RUSLE the by far most applied soil erosion prediction model around the globe. By an ISI query (<http://apps.isiknowledge.com/>) for the period 2003-2014, Auerswald *et al.* (2014) stated that 844 hits correspond to the keywords 'Universal Soil Loss Equation', 'USLE', 'Revised Universal Soil Loss Equation' and 'RUSLE'. Our own ISI query for 2003-2016 resulted in 1,118 hits, which indicates a steep increase over the last years (2013: 97, 2016: 129 with 8 papers highly cited and 3 'hot papers'). The most well-known models based on USLE/RUSLE technology, such as SWAT (Arnold *et al.*, 1998), AGNPS (Cronshey and Theurer, 1998), Watem/Sedem (Van Rompaey *et al.*, 2001), EPIC (Williams *et al.*, 1983) led to 243 hits in the same period (2003-2014). Approaches independent of the USLE technology, such as WEPP (Lafren *et al.*,

1997), LISEM (DE ROO *et al.*, 1996), EUROSEM (Morgan *et al.*, 1998) and PESERA (Kirkby *et al.*, 2008) achieved 254 hits. As observed by Sonneveld and Nearing (2003), USLE/RUSLE applications reach from small-scale up to national studies (P. Borrelli *et al.*, 2016; Panagos *et al.*, 2015; Schaub and Prasuhn, 1998; Teng *et al.*, 2016; UNEP/RIVM/ISRIC, 1996; Yue *et al.*, 2016).

Highly recognized studies already used RUSLE-based model applications to assess the magnitude of soil erosion process at global scale. Van Oost *et al.* (2007) used a previous version of the RUSLE model in 2007 to estimate soil erosion from agricultural land globally which was published in *Science*. Therein, they made a thorough attempt to validate the model against cesium-137 inventories. This RUSLE-based global estimate is currently the most recognized estimation of soil erosion due to rill and sheet processes. Recent work in *Nature Geoscience* by Quinton *et al.* (2010) also adopts this approach as it is the only way to enable large-scale soil erosion estimates. On *Nature Climate Change*, Wang *et al.* (2017) published a study using RUSLE-based predictions to simulate global anthropogenic sediment flux and assess their potential impact on the global carbon emissions.

As we note in the revised version of the manuscript: '*Notwithstanding the significant scientific contribution of the expert based GLASOD (Oldeman, 1994) approach in the 1990s and the need to further advance process-based physical models (Boardman and Favis-Mortlock, 1998; Jetten and Favis-Mortlock, 2006).*' and '*In a context where process-based physical models and the availability of input data are not yet mature enough for global scale applications (Jetten *et al.*, 2003), simple and physically plausible empirical methods for predicting soil erosion such as RUSLE (Renard *et al.*, 1997) can provide reasonably accurate estimates for most practical purposes (Supplementary Note 1). This especially applies to wide spatial scale applications when prediction errors do not exceed a factor of two or three (Bagarello *et al.*, 2012).'*'

In this regard, we believe that a RUSLE-based global model as presented in this study, can represent a powerful tool for identifying hotspots of soil erosion and areas of concern. We also believe that it provides the basis for a more strategic approach in directing new monitoring/modelling efforts and preparing policy decision-making. Moreover, it can stimulate to a scientific discussion about global soil degradation processes due to water erosion.

Comment 3. *I suggest to develop a deep publication review instead to re-use the data of Montgomery.*

Reply 3. Noted with thanks. As described in the reply to comment 1, the introduction was modified and a broader literature review is presented as Supplementary Note 1. Further discussions are also provided dealing with this aspect.

With regard to the data reported in Montgomery's (2007) meta-analysis, we believe that the study of Montgomery (2007) is the most comprehensive meta-analysis on measured soil erosion rates. Based on these well-appreciated insights, we substantially extended his comparison of soil erosion rates by adding various studies obtained from an in-depth literature review. Accordingly, we did not just re-use Montgomery's data but integrated them to achieve a broader dataset.

Please, note that the analysis at meta-data level is only one of the four approaches that we undertook to evaluate the consistency of the global soil loss predictions. Further analyses are provided in the revised Discussions and the Supplementary Discussions. In addition, a sensitivity analysis to evaluate the influence of parameter estimation on uncertainty and the effect of the parameters on the model output was performed as requested by the Reviewers.

We also suggest to compare the approach of Montgomery with the one proposed in our study to appreciate the difference (e.g., Figure below).

Our Figure 6 (right): Representation of soil erosion rates measured on agricultural fields under conventional agriculture (n = 779), geologic erosion rates measured on alpine terrain (n = 44), soil-mantled landscapes (n = 1456), low gradient continental cratons (n = 218), grassland and scrublands (n = 63), native forests (n = 46) and averages of our predictions (indicated by an asterisk). Large parts of the measured data come from the study of Montgomery⁶ integrated with data from other meta-analysis studies. The vertical red line indicates average value of soil erosion.

Comment 4. The is a lack of discussion in your paper, and the introduction is poor, which repetitions to what is knowNature needs a alive and fresh research and in your paper there is many information that is already know.

Reply 4. Naturally, an introduction of a scientific paper deals with the state-of-the-art on the topic and this refers to the already existing knowledge. As you rightly pointed out you would prefer to see more literature cited and discussed. We recognize that additional literature information could offer a more comprehensive overview on existing studies and could also improve the overall quality of the manuscript (see reply to Comment 1). Accordingly, we extended the introduction and added a further and larger literature review as Supplementary Note 1. For this, we contacted two well-recognized experts with many years of experience in field measurement and multi-scale RUSLE-based modelling (Prof. Vito Ferro and Prof. Vincenzo Bagarello) to help us improving the quality of the literature review and the consistency of the manuscript.

With regard to your comment ‘Nature needs a alive and fresh research and in your paper there is many information that is already know’, in the introduction and throughout the paper we highlighted current research gaps and expressed that scientists, to date, cannot make statements about i) the extent to which different countries are prone to erosion based on a harmonized methodology, ii) the effects of land use change that occurred during the last ten years and iii) the mitigation effects derived from conservation agriculture programs. The effects of the different land management and conservation policy applications observed in South America and Africa on soil erosion are indisputably a novelty. The proposed geo-statistical approach allowed, for the first time, to thoroughly define the land uses and their changes, the extent, types, the spatial distribution of the global croplands and the effects of the different regional cropping systems in a global soil erosion model. The country-specific changes of the annual average soil erosion derived from this innovative global modelling approach (Figure 2) offer unprecedented information on country-based soil erosion trend.

Of course, we do not re-invent the patterns of global soil erosion risk. We brought this assessment, however, to a new level linking it for the first time to the key parameters required to design conservation

planning. For the first time research is able to estimate the impact of mapped land cover change on soil erosion - something unprecedented at regional scale, not to mention global scale.

We feel that the lack of discussion criticized by Reviewer #1 remains unsubstantiated but without any further observations or arguments provided we have difficulties to respond to it in more detail. Nevertheless, we would like to remark that in the first part of the discussions chapter, we dealt with the **a)** novelty of the proposed model application and its results, **b)** the potential role of conservation agriculture, **c)** we provided a comparison between our global estimates and previous ones, and **d)** evaluated the consistency of the modelling results. In the second part, we dealt with the implications of our global modelling from a multidisciplinary perspective linking the findings of our map to **i)** other RUSLE based models **ii)** GDP measures to identify potential pressures on food production systems, **iii)** risks of increased food and feed prices due to phosphorous shortages, **iv)** the global soil organic carbon (SOC) pool that form the basis for emission levels and climate change analyses, **v)** the economic costs of soil erosion and **vi)** the overall implications for policy decision-making and sustainable development. We consider this multidisciplinary discussion as an alive and fresh contribution to research.

We believe that our paper reaches far beyond the already known insights on soil erosion and its spatial dynamics and thus fulfils the criteria of scientific advance, originality and novelty.

Final remark of Reviewer #1: *I suggest to reject the paper because a lack of novelty. A old method under doubt if it is correct or not to apply to other regions than the Central North America rolling landscapes, a poor introduction with many flaws and lack of key previous research, and finally the lack of a proper discussion to move further the knowledge of soil erosion....*

Reply to final remark: We ask Reviewer #1 to kindly refer to our reply to Comment 2 regarding the applicability of the model to regions outside Central North America and Comment 4 for our response to the supposed lack of novelty and discussion. With regard to the comment of Reviewer #1 regarding the 'lack of previous research' we refer to our Comment 1.

Based on the little details that Reviewer #1 provides to substantiate his or her opinion, we find it difficult to accept a mere rejection of the paper without giving us the opportunity to engage in a scientific discussion. In this response as well as in the manuscript itself, we have presented our arguments and supported them with citations and illustrations why our paper rests on the latest soil erosion modelling techniques, even advances them and provides critical new insights that have not been available before.

Previous studies did not (and actually could not) make statements about:

- i) the extent to which different countries are prone to erosion based on a harmonized methodology (which not only prevents comparison but also a classification of different regions/countries)
- ii) the effects of land use change that occurred during the last ten years and
- iii) the potential mitigation effects derived from conservation agriculture programs

A recent study of Doetterl, van Oost and Six (2012) highlights the need to perform more accurate and representative land use and land use change RUSLE-based soil erosion assessments. A justified request that we aimed to accomplished with our manuscript by ensuring the following novel aspects:

1. Our modelling approach presents a step change, not only allowing an unprecedented high resolution global prediction of soil erosion (250m vs. 10-60 km of previous studies), but also allowing us to predict how it is changing in space and time.

2. Unlike previous studies which dealt with soil erosion as a static process, we shed light on the impacts of the 21st century global land use change on soil erosion. Providing trends on national base.
3. For the first time we provide insights into the mitigating effects attributable to conservation agriculture (CA). As pointed out by Reviewer #3, in addition to land use change our model also integrates the offsetting effect of the application of conservational farming practices - an element that cannot be found in any other global study on soil erosion.
4. The proposed geo-statistical approach allows, for the first time, to thoroughly define the land uses and their changes, the extent, types, the spatial distribution of the global croplands and the effects of the different regional cropping systems in a global soil erosion model. With a level of accuracy considerably higher than previous studies.
5. The global rainfall erosivity patterns are quantified based on a novel, thoroughly unbiased methodology based on rainfall intensity rather than annual total rainfall volume. We used a time series of sub-hourly and hourly pluviographic records (3,625 stations covering 63 countries) spatialized through a Gaussian Process Regression (GPR) geo-statistical model. This is the result of the collaboration of 30 researchers worldwide. (Recently published on Nature Scientific Reports, Panagos, Borrelli *et al.*, 2017).
6. The full global assessment enables us to link erosion to the economy through GDP so that policy makers will get a full global picture, rather than insights into cherry picked areas.
7. Beyond the geographic appraisal, the discussion consists of an assessment from multidisciplinary perspectives.

We therefore kindly ask Reviewer 1 to reconsider his or her decision of rejecting our paper.

Reviewer #2

General Intro 1. I have asked two colleagues to assist me with this review, as their very broad experience of soil erosion in the real world complements my own expertise as a soil erosion modeller. They are:

Prof. John Boardman, Environmental Change Institute, University of Oxford

Dr Bob Evans, Global Sustainability Institute, Anglia Ruskin University

Their comments follow my own.

Dave Favis-Mortlock

Reply to General Intro 1. Thankfully acknowledged. A reply to all comments is provided in the following part of the letter.

General Intro 2. A castle built on sand, however elegant, is still a structure built on unstable foundations. Similarly, this study's use of admirably high-resolution input data, and a well-chosen variety of land-use change scenarios, cannot hide the poor erosion modelling foundation upon which it rests. "Garbage in, garbage out" is one of the oldest catchphrases in computing, but it applies admirably here.

The USLE model, on which RUSLE is based, is a statistical model which was devised to estimate rates of erosion on small plots in the USA (most of which were situated in the mid-western states: e.g. Nicks, 1998). As with any statistical model: using it with data other than that which was used to derive the model, amounts to extrapolation. Whether this is trivial or serious depends on the degree to which the two sets of data differ. I suggest that the data used in this study, and the data used to define the USLE and its RUSLE derivative, differ tremendously.

Reply to General Intro 2. While we appreciate that Reviewer 2 acknowledged the study's 'admirably high-resolution input data, and a well-chosen variety of land-use change scenarios' we do not agree with the statements regarding the 'poor erosion modelling foundation upon which it rests' and the link to the old adage 'Garbage in, garbage out'.

We are aware that the scientific community has been divided about the role of empirical- and-process-based soil erosion prediction models during the last two decades. This scientific debate has deep roots. The scientific exchange of opinions published in Science 17 years ago (Nearing et al., 2000) on the application of process-based modelling techniques from plot to field-, regional- or national scale is a good example. It seems to us that we face a similar exchange of scientific views right now. We are aware of the contribution provided by Dave Favis-Mortlock (Reviewer 2) and his colleagues on the development of process-based models. Without a doubt, we recognize the value of process-based models. Especially because we use process-based models, together with other state-of-the-art techniques such as Caesium-137 and biomarkers, to assess soil erosion in small-scale studies. At present, the research community is working on both in parallel: i) improving the applicability of complex process-oriented models (Boardman, 2006; Favis-Mortlock, 2013; Morgan and Nearing, 2000) and ii) updating the existing empirical approaches such as the Universal Soil Loss Equation (USLE) (Bagarello et al., 2015, 2008; Bargarello and Ferro, 2004; Di Stefano et al., 2017; Kinnell, 2014) which remains attractive from a practical point of view (Cao et al., 2015; Gessesse et al., 2015) and large scale assessments (Ito, 2007; Van Oost et al., 2007).

As we note in the revised version of the manuscript: 'Notwithstanding the significant scientific contribution of the expert based GLASOD (Oldeman, 1994) approach in the 1990s and the need to further advance process-based physical models (Boardman and Favis-Mortlock, 1998; Jetten and Favis-Mortlock, 2006)

recent studies have shown the potential of RUSLE (Renard et al., 1997) model-based approaches as a significant step towards a change in global water erosion assessment (Doetterl et al., 2012; Ito, 2007).'

In our view, the choice of a model should be a matter of scale and the purpose of its application. We aimed at a global assessment of soil erosion focusing our attention on the role of global land use changes and agricultural conservation. We believe that for our purpose, the use of RUSLE is an appropriate choice. Our decision is supported by the absence of thorough large-scale model alternatives. As we demonstrate on the following pages and state in the revised version of the manuscript: *'In a context where process-based physical models and the availability of input data are not yet mature enough for global scale applications (Jetten et al., 2003), simple and physically plausible empirical methods for predicting soil erosion such as RUSLE (Renard et al., 1997) can provide reasonably accurate estimates for most practical purposes (Supplementary Note 1). This especially applies to wide spatial scale applications when prediction errors do not exceed a factor of two or three (Bagarello et al., 2012).'*

As a matter of fact, today global information on soil erosion are still largely from the Global Assessment of Human-induced Soil Degradation (GLASOD) (Oldeman, 1994), a qualitative product resting on observations made about 30 years ago. GLASOD provides insights about soil erosion based on a static observation approach but does not quantify the effects driven by changes in land use. Accelerated soil erosion, however, is primarily driven by modifications in land use and management.

In this study, we present a powerful exploratory tool to quantify soil erosion and illustrate its potential patterns and trends in a context of global change. It is a tool working in a GIS environment that allows the spatial link to environmental factors, satellite imagery and land use map to enable model scenario analysis.

In the view of the current lack of understanding of the effects of globe change on soil erosion, *a priori* rejection of our model that provides advanced information mainly because it is based on a deterministic and empirical-based model (i.e., RUSLE) and not on a process-based model appears to be unjustified. We therefore kindly ask the group of Reviewer 2 to reconsider their decision of rejecting our paper.

Comment 1. *USLE/RUSLE data refers to small plots which are hydrologically disconnected from their surroundings. This study (and other similar studies) use RUSLE on landscape tiles which are not, in reality, hydrologically disconnected from their surroundings. What about flow onto/off of each tile? Completely ignoring the connectivity of runoff and sediment, as does the present study, is a vast oversimplification of reality. This alone renders tile-based RUSLE studies, such as the present study, meaningless.*

Reply 1. Thank you for bringing forward your arguments. USLE/RUSLE models rest on empirical observations based on more than 10,000 plot-years of basic run-off and soil loss data (Lafren and Moldenhauer, 2003) in 49 US locations covering a large variety of landscape conditions. As every statistical approach, one needs to select a representative set of conditions to model a complex system. USLE/RUSLE plots had a length less than or equal to 122 m and a slope ranging from 3 to 18%. Defining the mathematical structure of the USLE a reference condition, named as *unit plot*, was also used. The unit plot was defined as a plot 22.1 m long, with a 9% slope, maintained in a continuous regularly tilled fallow condition with up-and-down hill tillage. The unit plot was used to compare soil loss data collected on plots that had different slopes, lengths, cropping and management and conservation practices.

In our opinion, the unit *'plot'* (Wischmeier and Smith, 1965) constitutes a valid approach to develop an empirical equation to predict soil erosion. Plot measurements have been, and still are, essential to improve the ability to predict soil erosion and evaluate the mitigation effect of soil conservation practices (Anache et al., 2017; Cerdan et al., 2010; García-Ruiz et al., 2015; Guo et al., 2015).

In the RUSLE model, soil erosion (soil loss) refers to the amount of sediment that reaches the end of a specified area on a hillslope that experiences net loss of soil by water erosion (we added Supplementary Note 2 for more details). It is expressed as a mass of soil lost per unit area and time ($\text{Mg ha}^{-1} \text{ yr}^{-1}$). As explained by Nearing et al. (2017) '*soil loss does not equate to the sediment yield from a hillslope that exhibits toe-slope deposition, which are most cases. It is, rather, the sediment delivered to the bottom of the slope area that feeds onto the toe slope. Slope lengths of soil loss areas end where deposition begins.*'

At a large spatial scale, such as in the present study, the study area has to be discretized using a square grid subdivision (raster scheme), choosing a mesh size consistent with the scale of the original model deduction. Using a raster scheme applied to the RUSLE model corresponds to hypothesize that each cell is independent of the others with respect to soil loss. In other words, at large spatial scale such as a region or a global perspective, a simple index-based model able to calculate an average soil loss, at annual or mean annual scale, allows to compute the involved factors using spatially distributed input values.

By contrast, the cells cannot be assumed to be independent from each other when the sediment delivery processes have to be modeled such as at basin scale. Even in this case, the RUSLE scheme can be applied by coupling it with a mathematical operator that expresses the hillslope transport efficiency (Ferro, 1997; Ferro and Porto, 2000; Van Oost et al., 2007). Please note, however, that this is not the case in our current study.

The difference between the two approaches would be that RUSLE provides the erosion risk for each pixel due to the combination of the six factors, thus acknowledging the on-site soil degradation potential. Models considering the catchment hydrology or connectivity of landscape units, in contrast, focus on the off-site sediment yield which will not provide information on the on-site soil degradational status.

We interpret the prediction of our global RUSLE-based model in line with previous large-scale applications (Van Oost et al., 2007; Ito, 2007; Borrelli et al., 2016; Panagos et al., 2015), hypothesizing that each cell is independent from the others with respect to soil loss.

Comment 2. *The USLE and RUSLE rely on an R-factor which is based on rainfall intensity (EI30). Thus low rainfall intensities imply low erosion rates. However in many parts of the world (e.g. northern Europe), erosion occurs under low rainfall intensities provided that the soil is saturated (in other words, runoff here results from SOF rather than HOF). For these areas, the R-factor is an inappropriate measure of erosivity: variants of the USLE such as MUSLE or the Onstad-Foster are better suited. This study (and other similar studies) uses only the EI30-based R-factor for all locations considered, irrespective of whether erosivity at that location is driven by rainfall intensity or amount (or a mix of the two).*

Reply 2. Erosion from saturated overland flow is not explicitly considered in the RUSLE model. This has been an extensive scientific debate by the USDA scientists and the scientific community which has become more silent by the agreement that slight underestimations must be accepted given the lack of alternative 'omnivorous' empirical or process-based physical approaches.

From a global scale prospective, high magnitude erosion occurs under Hortonian conditions (rainfall intensity > infiltration capacity). While in local studies, for instance in a catchment of the UK, MUSLE or Onstad-Foster can outperform RUSLE, we believe that this will not be true for a global scale study. In support of this, we would like to cite some previous large-scale studies in the European Union (Panagos et al., 2015), Australia (Teng et al., 2016), China (Yue et al., 2016) or at global scale (Doetterl et al., 2012; Ito, 2007; Quinton et al., 2010; Van Oost et al., 2007; Wang et al., 2017). These studies are all based on RUSLE. By contrast, MUSLE remains mostly employed for catchment-scale applications.

Please find the results of a physically-based model like PESERA (Kirkby et al., 2004) in the image below showing that even process-based modelled soil erosion rates of northern Europe (where saturated overland flow is dominant) are of less concern compared to soil erosion prone areas such as the Mediterranean region (rainfall intensity dominated processes).

Comment 3. The USLE and RUSLE do not attempt to describe any form of water erosion other than sheet or rill erosion. Gullying, tillage erosion, bank erosion, and loss of soil with crops are all ignored. While the authors of the present study do admit this on lines 302-303, this major limitation should be much more prominently stated.

Reply 3. RUSLE, just like the vast majority of the other water erosion models, only incorporates rill and sheet erosion impacts. Gully, bank erosion and landslide are not considered as we clearly defined in the manuscript. We kindly ask Reviewer 2 to provide us with detailed references to passages of our manuscript that could lead to misunderstandings that gullying, tillage and bank erosion are part of the presented model. We are happy to include further clarifications into these passages in order to avoid misunderstandings. Modelling gully, bank or landslide erosion on a global scale would be a very different study which is not the aim of this manuscript and should be done by the respective experts within these fields.

Comment 4. Within the USLE database, there is a Coefficient of Variation of up to 15% for soil loss measurements from replicate plots, as pointed out by Nearing et al. (1999). Random variability will of course be higher for non-replicate plots. Thus there is a considerable error range for any USLE or RUSLE prediction. However, I see little in the present study regarding error. There should, at the very least, be an estimate of error associated with all RUSLE predictions.

Reply 4. This is an intrinsic limit that all empirical models have in common independent from the location where they are applied. We hope that you agree with us on the fact that errors are present in all models. Process-based models applied at field or small catchment scale also have to deal with these challenges. Jetten et al. (2003) noted that 'more complex, physically based models do not necessarily perform better than lumped, regression-based models, mainly because input errors increase with increasing model complexity.' Favis-Mortlock (1998), comparing the results of process-based models (WEPP, Laflen et al., 1997) with USLE-based models (Wischmeier et al., 1978) such as EPIC (Williams et al., 1983) and GLEAMS

(Leonard et al., 1987) for a field site in the US, observed no significant difference between the model performances. Other studies confirmed that physically-based models do not necessarily perform better than the empirical models (Morgan and Nearing, 2000; Quinton, 2013; Stolpe, 2005).

Our study aims to shed new light on the impacts of 21st century global land use change on soil erosion, providing insights into the mitigating effects that can be attributed to conservation agriculture. Due to the global nature of the study, a RUSLE-based approach reflects the required level of complexity and precision. At the same time, we believe that it constitutes an appropriate method in terms of affordable input data requirements and expenditure of time to ensure that the results are still relevant by the time they are available.

With regard to your suggestion of estimating the error associated to the predictions, we agree with your point. We added a sensitivity analysis (Supplementary Note 3) to the revised version of the manuscript in which we estimated the possible error associated with the individual input factors. Moreover, further approaches to evaluate the performance of the model were also undertaken and described in the Discussions and the Supplementary Discussion Chapters.

Comment 5. *Soil erosion by water is highly patchy, spatially. To present erosion rates as an average over the whole of each 250 x 250 m tile is potentially misleading, since erosion rates within that tile may well vary greatly. Even in locations suffering severe erosion, gullying on one part of the tile may co-exist with very little erosion on other portions of the tile. In this study, I see little consideration of how average rates for whole tiles would compare with rates observed at that tile's location.*

Reply 5. Please note that we do not aim at modelling the erosion risk associated to gully erosion. Our paper focuses on the erosion risk associated with sheet and rill erosion processes. These are two different processes which need different algorithms and input data for the model simulation.

With regard to modelling sheet and rill erosion, compared to the former state-of-the-art modelling which was based on a 10 km to 40 km cell size resolution, we increased the cell size resolution to a 250m grid which constitutes at least a 1,600 times increase in the modelling spatial detail. This is crucial because the accuracy of the model outcomes critically depends on the detail of the input data. A better representation of the actual land cover thus results in a better model prediction. Figure 1 illustrates the significant gain in spatial detail. Global modelling in general, not only soil erosion, depends on the availability of land use and land cover data at a resolution finer than 250m. We would like to highlight that at present, in fact, the unavailability of resolution data <250m constitutes the limiting factor and that this should not be understood as a limitation of the modelling per se.

GLADIS (FAO)

Borrelli et al.

Spatial resolution (previous studies)

GLADIS = 10 km

Spatial resolution (new study)

Borrelli et al. 2017 (250m)

Reply to further comments of Reviewer 2

Line 72. Why choose these two previous assessments?

Reply: The fact that soil erosion is considered to be one of the main threats to soil is well acknowledged in literature. We selected these assessments because they are relevant global scale studies made by the Food and Agriculture Organization of the United Nations in collaboration with the International Soil Reference and Information Centre (ISRIC).

Thank you for bringing forward this improvement potential. Please note that in the revised version of our manuscript we refer to a broader number of studies and global assessments to address your point.

Line 73. "Unsuitable agricultural practices" should be added to this list.

Reply: Noted with thanks. We have included this in the revised version of our manuscript.

Line 74. I dispute that the mechanics of water erosion are “well-understood”. If they are, why do we not have a robust and useful model for e.g. gully erosion?

Reply: Please note that we are not dealing with gulying processes in this paper but we would warmly welcome scientific contributions on gulying processes so that future studies can take these processes into consideration. Please note that in line 74 we refer to mechanics of deforestation, overgrazing, tillage and unsuitable agricultural practices (*the later as suggested by your former Comment*) that are well-known and documented. Note that line 74 does not refer to the process of accelerated soil erosion. Please also refer to our reply to Comment 3.

Line 98. This implies that qualitative approaches are somehow less valid than quantitative approaches. This is not the case if the quantitative approaches are based on unsuitable models.

Reply: Please note that we do not state that qualitative approaches are less valid than quantitative approaches. For the avoidance of doubt, in line 95-99 (old version of the manuscript) we state the following:

‘Accelerated soil erosion is primarily driven by modifications in land use and management. Spatial pattern of land use and land use cover change, especially in areas susceptible to accelerated soil erosion, provide further reason to re-evaluate former qualitative approaches, considering the worldwide increase of croplands and pastures by 279 million hectares (ca. 16.7%) between 1985 and 2013.’

We stressed that the qualitative GLASOD (Oldeman, 1994) assessment was made about 30 years ago. Ever since, there have been substantial land use changes and these effects have not yet been incorporated into the GLASOD assessment.

Line 102. Why do the authors omit to mention earlier studies which show the limitations of erosion models? The GCTE Soil Erosion Network studies (e.g. Jetten et al., 1999) should certainly be discussed here.

Reply: Please note that in the revised version of our manuscript we acknowledge the work done by GCTE Soil Erosion Network and their studies. We also discuss the limits of the RUSLE model in greater detail and clearly state why we decided to use it for a global assessment. We also added a Supplementary Note 1 where we deal with the concept of soil erosion modelling, the categories of existing models and studies referring to the use of RUSLE-based models outside the United States (US). We added ca. 1,400 words and cited around 40 additional studies to enrich the quality. The decision to include this new text in the Supplementary Information rests on the fact that the word limit was reached in our revised version the manuscript.

Line 105. “the erosive power of rainfall”: see my comments re. R-factor above.

Reply: Please refer to our response to Comment 2.

Line 126. Which empirical observations?

Reply: The empirical observation of Montgomery (2007) were further enhanced with other measurements (Figure below). For further information please refer to our response to Comment 3.

Our Figure 6 (right): Representation of soil erosion rates measured on agricultural fields under conventional agriculture (n = 779), geologic erosion rates measured on alpine terrain (n = 44), soil-mantled landscapes (n = 1456), low gradient continental cratons (n = 218), grassland and scrublands (n = 63), native forests (n = 46) and averages of our predictions (indicated by an asterisk). Large parts of the measured data come from the study of Montgomery (2007) integrated with data from other meta-analysis studies. The vertical red line indicates average value of soil erosion.

Line 126. Which regional assessments?

Reply: Please refer to the Discussion Chapter and the extensive Supplementary Materials that we provide (Supplementary Discussions). Therein, one can find comparisons of our RUSLE model with RUSLE models applied in the US and Europe at regional scale. As a result of the comparisons to assess the performance of the model, we find that our model performs as good as the regional high resolution applications.

Line 262. This implies that finer resolution input data improves the quality of results. It does not, if the modelling approach is itself flawed: GIGO (see above).

Reply:

While large scale modelling applications naturally come along with higher uncertainty than local applications, the use of high resolution land use data enhances the meaningfulness of the model. In this regard, we would like to refer to the figure provided in our reply to Comment 5. In the figure, a 10 km grid resolution study (FAO GLADIS) is compared with our study using a 250m grid. There is a 1,600 times increase in the modelling spatial detail.

With regard to the validity and applicability of the global model we refer to our reply to Comment 2.

Lines 298-310: While the authors at last admit to a major limitation of their study (only sheet and rill erosion considered), they then attempt to use this limitation to “provide evidence of the comprehensiveness of their modelling approach”. This is deeply flawed logic.

Reply: Our paper clearly states that only rill and sheet erosion are dealt with but as mentioned above, we are happy to review passages if you provide us with detailed references to passages of our manuscript that could lead to misunderstandings. Please also refer to our response to Comment 3.

With regard to the quoted sentence, we recognized that the sentence that you refer may lead to misunderstandings. Thus, we decided to remove it from the revised manuscript text. ~~‘This, moreover, provides evidence for the comprehensiveness of the proposed modelling approach’.~~

Side note: Please note that with the term comprehensive we do not mean that we modeled all soil erosion processes (e.g., gully). We state this in the text. Within the RUSLE framework, the comprehensiveness refers to i) the remote sensing data used for representing the land use, ii) the statistical approach to classify the different crop system, iii) the distribution of conservation agriculture, iv) a global emotivity quantified with a new thorough unbiased methodology based on rainfall intensity instead of volume, and v) several other modelling elements described in detail in the methodology. We hope that you agree that including all these elements in a global model constitutes a major advancement compared to previous global assessments.

Reviewer #2 Colleague No. 1

Intro comment: *Just finished reading this. Difficult to take it seriously based as it is on RUSLE figures.*

Reply: Please refer to our responses to Favis-Mortlock concerning the validity of the model.

line 90. I am willing to bet this is re-cycled from Pimental or GLASOD.

Reply: For the avoidance of doubt, in line 88-91 we stated the following and clearly cited the origin:

'The FAO led Global Soil Partnership¹⁹ reports that 75 billion tones (Pg) of soil are eroded every year from arable lands worldwide, which equates to an estimated financial loss of US\$400 billion per year. This soil erosion estimate dates back to 1993, first reported by Myers²⁰ and cited by several succeeding studies^{18,21,22},

line 91. Therefore I don't think these are 'succeeding studies'

Reply: Please also refer to our response to your Comment above. Unfortunately, the Reviewer does not express an opinion to support this doubt with further observations or arguments so that we can respond to it in detail to the reason why he doubt that these studies are succeeding.

18. Montanarella, L. *Agricultural policy: Govern our soils. Nature 528, 32–33 (2015).*

21. Pimentel, D. et al. *Environmental and Economic Costs of Soil Erosion and Conservation Benefits. Science 267, 1117–1123 (1995).*

22. Eswaran, H., Lal, R. & Reich, P. F. in *Response to land degradation (ed Bridges, E. M.) 20–35 (Science Publishers Inc, 2001).*

line 100. The references are to forests

Reply: Thank you very much for bringing this to our attention. We have mistaken between the two FAO references for the year 2016. This is corrected in the revised version of our manuscript.

line 114. Note the emphasis is on rainfall intensity again!

Reply: The team of authors dedicated much effort to achieve this quantification of the global rainfall erosivity dynamics. It is the result of an international collaboration involving 30 scientists across the globe, where the global rainfall erosivity patterns were quantified with a thorough methodology based on rainfall intensity instead of volume, using a time series of sub-hourly and hourly pluviographic records (3,625 stations covering 63 countries) spatialized through a Gaussian Process Regression (GPR) geo-statistical model. Our validation approach showed that it is considerably more accurate than former global studies:

Figure. Comparison of predicted and measured values of the R-factor for the different previous global rainfall erosivity models (Naipal et al., 2015 (a); Nachtergaele et al., 2010 (b); Yang et al., 2003 (c)) with the unbiased estimate used in our study (d) with the study adopted from Panagos, Borrelli et al. (*Nature Scientific Report*).

The paper has been recently published by *Nature Scientific Report*. It can be found here:

<https://www.nature.com/articles/s41598-017-04282-8>

Line 119. I think the basic cell size in 5 ha...rather large to be averaging across!

Reply: We suggest to address your concerns to the authors of previous global studies who presented models with a cell size resolution between 10 and 40 km. We present a model with 250m which has been a major improvement (1,600 times more detailed in terms of land use and topography).

Line 126. Where are these 'empirical observations' they are using?

Reply: Please see Figure below (Figure 6 in the revised manuscript) reporting the study of Montgomery (2007) published in PNAS modified by the Authors.

We believe that the study by Montgomery (2007) is the most comprehensive meta-analysis on measured soil erosion rates. Based on these well-appreciated insights, we substantially extended his comparison of

rates of soil erosion by adding various studies obtained from an in-depth literature review. We invite the Reviewer to compare the approach of Montgomery with the one proposed in our study to appreciate the difference (e.g., Figure below).

Please note that we did not aim to present a review paper. With 64 citations, we are close to the limit suggested for scientific articles by *Nature Communications*. We refer to the guidelines of *Nature Communications* regarding the submission of papers which we adhered to.

line 147. T values have been discredited as a political tool with no scientific basis

Reply: T value means 10 Mg/ha/yr. The concept of T values is widely applied in scientific publications. Please also refer to: McCormack, D. E., and K. K. Young, 1981 – Technical and societal implications of soil loss tolerance. In: Morgan R.P.C. (Ed.), *Soil Loss Conservation: Problems and Prospects*; John Wiley & Sons, Chichester, p. 365-376.

line 171. What does it mean 'Classified...'?

Reply: With the term classified we refer to the activity of arranging data in classes or categories.

line 199. Average cropland erosion rate (12) is ridiculously high

Reply: In contrast to the UK where erosion is rather limited, tropical locations are found to have significantly higher soil erosion rates which can even reach up to 100 Mg/ha within a single event.

line 228. Conservation agriculture covers great variety of measures...very dodgy data

Reply: We use official FAO data. These are to the best of our knowledge the best data available at the moment. If the author do have better data, please be so kind and consider sharing these with us.

line 293. 'Most cited estimate is by Pimental et al.' This was conclusively shown to be very bad science and probably an order of magnitude too high (Boardman 1998, Journal of Soil & Water Conservation).

Reply: The paper of Pimentel et al. (1995) in Science received 2,347 citations. It has some limitations imposed by the limited knowledge and global datasets available at that time.

One order of magnitude too high would imply passing from several billion to hundreds of million tones. The tendency of Pimentel et al.'s (1995) study to overestimate erosion has been acknowledged in recent literature. We refer the reviewer to our abstract where we state, 'we challenge the previous annual soil loss reference values as our estimate, with 35.9 Pg yr⁻¹ of soil eroded in 2012, is at least two times lower.' Our estimation on rill and inter-rill processes is considerably lower than the 75 Pg yr⁻¹ proposed by Pimentel et al. (1995). We believe, however, that the statement about Pimentel et al. (1995) that it 'is bad science' would lead into a disrespectful debate and a lack of scientific foundations. Note that global soil erosion rates cannot be one order of magnitude lower than the Pg.

line 296. What are these measured erosion values?

Reply: Please, see related reference in the text.

line 297. Note that 'plot data' does not give use rates for the landscape as a whole

Reply: We would like to emphasize that we did not state it otherwise. Please, refer to our reply to Comment 1 of Reviewer 2 and to the newly added Supplementary Discussions.

lines 299- 301. This need unpicking. the 10 x 10 km cell size is worrying; the link to the US cropland data (I think this is EPIC data at pre-determined points?) is unclear and worrying.

Reply: We refer to the sentence again. Therein, we refer to the study of Doetterl, Van Oost, and Six, 2012.

line 310. What is this 'previously assumed in the literature'? cf Boardman 1998

Reply: With 'previously assumed in the literature' we refer to Doetterl, Van Oost, and Six, 2012 as cited in the sentence.

line 318. What are these 'cratons'?

Reply: A craton is a term communally used in geomorphology. With regard to the definition and scientific discussion of cratons please refer for example to:

Yuan, H., & Romanowicz, B. (2010). Lithospheric layering in the North American craton. Nature, 466(7310), 1063-1068.

Lines 331-333. Lets the cat out of the bag...problems with their approach BUT it is still a 'powerful assessment tool'. I beg to differ.

Reply: Noted with thanks. We respect that you may have a different opinio. We would appreciate some evidence to support this opinion.

Additional comment: They claim somewhere that their approach enables us to identify hotspots. But Boardman (2006 Catena) already identified these, as best one can, based on empirical data.

Reply: We do not agree Boardman (2006 Catena) reported global hotspots as copied in below. Our model aims to be more updated and sophisticated than a literature review citing studies made between 1923 and 2004 (Figure 1 and a Supplementary Figure S2 are reported below).

Suggested global erosion hotspots	
Hotspot	Evidence and comments
Loess plateau, the Yangtze basin and the southern hilly country	Reviews by Hao et al. (2004), Liu (1988: 192-3); Yellow river from Loess Plateau far higher sediment concentrations than other large rivers of the World (Milliman and Meade, 1983)
Ethiopia	Hurni (1993), Billi and Dramis (2003), Daba et al. (2003), Nyssen et al. (2004)
Swaziland	WMS Associates (1988), Mushala (1988, 1997), Morgan et al. (1997)
Lesotho	Highest erosion hazard of any single country in central or southern Africa (Chalela and Stocking, 1988)
Andes	Central Andes: badlands, stripped bedrock and sand dunes widespread through the area (Coppus et al., 2003)
South and East Asia	Moderate to extreme water erosion reported from India (10% of area), the Philippines (38%), Pakistan (12.5%), Thailand (15%) and Vietnam (10%) (Van Lynden and Oldeman, 1997); also river sediment yields are very high (Milliman and Meade, 1983)
Mediterranean basin	Extensive evidence of historical erosion; current problems including flooding and dam sedimentation e.g. Sole-Benet (in press), Kosmas et al. (in press), Rendell (1997), Faulkner and Hill (1997), Grove and Rackham (2001), Butzer (2005)
Iceland	Overgrazing has led to extensive desertification: Arnalds et al. (2001), Arnalds and Barkarsson (2003)
Madagascar	Little data but see Wells and Andriamihaja (1993) for serious gullyng
Himalayas	Controversial: dispute as to importance of natural/anthropogenically driven erosion
West African Sahel	Desertification due to population pressure and droughts (Glantz, 1994)
Caribbean and Central America	High rates of erosion mainly related to deforestation and subsequent cultivation e.g. Haiti (Paskett and Philoctete, 1990); Mexico (Sanchez-Colon, 2003); Jamaica (McGregor, 1995); Dominican Republic (Lugo et al., 1981); Montserrat and Nevis (Watts, 1988); Barbados (Carson and Tam, 1997)

Our global model:

Figure 1: Global rates of soil displacement by water erosion (soil erosion) ($\text{Mg ha}^{-1} \text{ yr}^{-1}$). The estimates are predicted through a RUSLE-based modelling approach integrated in a geographic information system (GIS) environment. The model provides rates on an approximately 250 x 250m cell basis for the land surface of 202 countries (ca. 2.89 billion cells; ~ 125 million km^2), covering about 84.1% of the Earth's land. The upper image (a) illustrates the soil erosion rates divided into seven classes according to the European Soil Bureau classification. The colour gradation from green to red indicates the intensity of the predicted erosion rates. The grey colour indicates the areas that were excluded from the modelling due to data unavailability (i.e., ice-covered land, terrestrial water bodies, large areas of bare rock, deserts and land with bare soil). The lower image (b) illustrates the erosion reduction rates on cropland obtained from the comparison between the conservation and the baseline scenario for the year 2012. The green gradient shows the percentage of reduction. The grey colour indicates the areas that were not modelled (no data).

Supplementary Figure S2: Global soil erosion hot-spots. Illustration of the soil erosion hotspots (highlighted in red in the map) across the globe defined as the areas with soil erosion estimates above $20 \text{ Mg ha}^{-1} \text{ yr}^{-1}$.

Reviewer #2 Colleague No. 2

Comment 1: *Throughout the paper rates of erosion are given as if they are actual rates, not modelled estimates. Thus, the information is never qualified, it is always 'are' or 'have' instead of 'estimated to be'.*

Reply 1: We presented a global RUSLE-based soil erosion assessment. Otherwise specified modelling results are always to be understood as estimations. Please note that in the revised version of the manuscript we modified the text in order to clearly state that our numbers are estimates. E.g.:

'Our findings indicate ~~an~~ a potential overall global...'

'The greatest increases are estimated to occur in Sub-Saharan Africa...'

'In 2012, we ~~observed~~ estimated an overall increase of 2.5%...'

'The soil conservation practice scenario shows ~~an~~ a potential overall offset of the estimated soil erosion increase...'

Comment 2: *The GIS manipulation and data crunching are impressive but they do not override the fact that RUSLE is not a satisfactory way of assessing erosion. Indeed, it does not model gully or ephemeral gully erosion. There is a need to acknowledge that there are problems with RUSLE, and these problems have been described in the literature. One of them is that models are based on soil passing across a line, usually a 'waterfall' (driver) at the base of the plot. It could well be that large amounts are splashed across a line, but that is happening everywhere on the slope so the input = output except at the top of the slope.*

Reply 2: Thank you for observing that the 'GIS manipulation and data crunching are impressive.' It took nearly two years of work to complete this study. As you will see, the revised version of the manuscript was substantially modified to provide more detail and highlight the limits of soil erosion modeling.

RUSLE, just like any other water erosion model, only incorporates rill and sheet erosion impacts. Gully, bank erosion and landslide are not considered as we clearly defined in the manuscript.

In the revised version of the manuscript, we also acknowledged more clearly that there are certain problems with RUSLE, and these challenges have been described in previous literature. We also added a Supplementary Note (1) about soil erosion modelling and a Supplementary Note (2) about soil loss in RUSLE. Moreover, we refer to the reply to your college for further details on this aspect.

Comment 3: *RUSLE's validation/cross checking is not with field assessments of erosion but with other modelled assessments, the model often being USLE based.*

Reply 3: As observed by Auerswald et al. (2003), a validation *sensu strictu* of USLE-based modelling at regional or larger scales is not feasible due to the lack of long-term field-scale measurements. Therefore, a cross-comparison of the modelling results to gain insights on the validity of the modelling predictions was performed. Two different comparison procedures were proposed in the first version of the manuscript. In addition, in the revised version of our manuscript an additional comparison (Supplementary Discussions) and a sensitivity analysis (Supplementary Note 3) were added to meet the concerns expressed by the reviewers.

As you noted, in one of them comparisons our global model predictions were related to regional RUSLE-based estimates in the US and EU. As we stated in the text: *'The good agreement between our estimates and the ones provided by independent studies give confidence that the quantitative estimates achieved through the global model are reliable and valid to a level close to these higher resolution regional assessment (Supplementary Discussions).'*

Comment 4: *Would it be better to use relative values rather than what they consider actual rates? Thus, x times more than a base rate?*

Reply 4: Some studies do so. However, according to the majority of the regional scale studies we preferred to present the estimates as potential soil erosion rates (understood as the soil loss of the RUSLE-based models).

Comment 5: *'Land use and land use change is the main discriminator of erosion rate. And its varying over time (increase or decrease in erosion is associated with land use change). The information on land use and land use change is interesting in its own right.'*

Reply 5: Thank you for recognizing that *'information on land use and land use change is interesting in its own right.'* This approach is innovative *per se* and it is used for the first time in a global soil erosion prediction model. As you can see from our title we aim to i) initiate a scientific debate on global soil erosion dynamics (*Quo vadis* soil erosion?) and present the potential impact of land use change and current conservation agriculture (Global impacts of 21st century land use change). New studies, roundtables and collaborations are needed to improve our understandings on the impact of global change on soil erosion process and *vice versa*.

Comment 6: *Recent work funded by NERC and Defra, as yet unpublished, shows that models may not be adequate substitutes for monitored data. And Philippe Bavaye in his 2017 Ecosystem Services paper also casts doubt on the use of models and simulations.*

We agree that monitored data are better than modelled. We believe that every modeler would agree with this statement. However, for large-scale studies we have to refer to models given the limited monetary and time resources. Please, note that in the revised version of our manuscript we stated:

'The authors recognise that the modelling based on data-driven assumptions has its limitations, and the need for field monitoring and local scale process-based modelling (Evans and Boardman, 2015). In the light of the encouraging insights gained from the operations of the model performance evaluation, however, the authors argue that the presented RUSLE-based global approach constitutes a powerful global assessment tool for identifying hotspots and areas of concern. It provides the basis for a more strategic approach in directing new monitoring/ modelling efforts and preparing policy decision-making.'

Comment 7: *Line 87. This paper is only of "fundamental importance" because Montanarella says so (see below for my response to that paper). And he was basing his paper on RUSLE findings (Panaqos et al).*

Reply 7: Please note that we do not refer to the fundamental importance of the paper in line 87. Rather, we refer to the fundamental importance of sustainable governance of soil.

86-87: *'Sustainable governance of soil has therefore become a topic of fundamental importance'¹⁸*

Comment 8. *Line 107. How are estimates robust when there is no sound basis for the inputs into RUSLE? The inputs are based on plot experiments not reality.*

Reply 8. Naturally, models can only be an approximation of the reality.

More than 10,000 years of cumulative data observations were used to develop USLE/RUSLE. To the best of our knowledge, there is no other study which rests upon such large data observation.

In addition, we would like to remark that we do not state otherwise in our manuscript.

Comment 9. *Line 296. Modelled rates coming down as models are linked to measured erosion values, but these values are from plots and still don't represent reality. And are probably still out by at least on order of magnitude.*

Reply 9. Please refer to our reply to your Comment 8 and similar comments made by Reviewer #2 and his colleague no 1.

Comment 10. *Line 331/2. Acceptance of need for field monitoring – good. But they do not try to relate monitored data to their model, and there was certainly information on that by the time of submission, mine and John's papers in Environmental Science and Policy for starters. But the comprehensive approach taken here does not make up for poor quality of input and output data, and much of that is based on modelled data.*

Reply 10. Noted with thanks. The reference to your study was added in the revised version: *'The authors recognise that the modelling based on data-driven assumptions has its limitations, and the need for field monitoring and local scale process-based modelling (Evans and Boardman, 2015).'*

Comment 11. *Line 581. Interesting that quote John and Jean's book but do not draw attention to the chapter on models.*

Reply 11. Please note that in the revised version of our manuscript the study was added and cited.

Comment 12. *I tried to check Reference 19 but was told 'page not found'.*

Reply 12. The link seems to work when we use it. Please refer to and kindly let us know if technical problems should occur

<http://www.fao.org/global-soil-partnership/resources/highlights/detail/en/c/416516/>

Comment 13. *Notes that spatial resolution is 250m, in fact for soils is 1km.*

Reply 13. This is correct. We stated it in the text (SI). By the time we made the study as the new global soil map at ca. 250 was not available. The ca. 250m resolution refers to the MODIS satellite imagery. For instance, to derive the topographical factors we used even more spatially detailed data, i.e., a ca. 90m SRTM DEM.

Comment 14. *In Supplementary results it notes “no reason to doubt the reliability of our model’. There is. And “Validity assessments can be challenging” and hence it would seem can be ignored. Thus their comparisons of rates of erosion, as noted above, are with rates from models with a similar basis. That is not validation, but a form of calibration.*

Reply 14. As mentioned before, this aspect has been substantially modified in both the Discussions and the Supplementary Discussions. Now we stated the aspect more clearly and hope that it helps to avoid potential misunderstandings.

Comment 15. *Line 326/7. But that is a 17 times difference!*

Reply 15. Unfortunately, Reviewer #2 does not support his opinion with further observations or arguments that we can respond to in detail. We invite him to substantiate his or her opinion so that we can respond.

Comment 16. *Line 765. What does ‘delta’ mean?*

Reply 16. Please, note that we use delta (Δ) to refer to the difference between the two numbers, i.e., a change of state between 2001 and 2012.

Comment 17. *Finally, I did try to respond to Montanarella’s Nature paper but it wasn’t accepted, see below for my responses:*

Reply 17. We believe that this goes beyond this discussion on our paper presented to Nature Communications. For other studies published on Nature we kindly ask you to discuss directly with the Editors of Nature.

Reviewer #3

We would like to thank Reviewer #3 for stating '*This paper is an excellent paper*' and more importantly, for highlighting that '*it is done on a very difficult but timely topic of soil erosion*'.

We are grateful to see that Reviewer #3 remarks the importance to protect natural resources and the relevance of including the use of conservation farming into the model.

Suggested modification: *The paper does not refer to very poor countries in Asia except India and China. Can the paper add some specific countries and soil erosion. This may be difficult due to the macro nature of the study but may add to the paper if possible.*

Reply to suggested modification: We see the point made by the Reviewer. Due to the high amount of data of this study, we could not describe the predicted erosion dynamics for each of the 202 considered countries. Note, however, that we would like to publish all data in a georeferenced grid format on Nature Scientific Data. Moreover, we would like to include some country-based descriptive statistics, e.g., mean erosion in 2001 and 2012, mobilized SOC as well as the land use data in the final version of the manuscript.

We believe that this would be the best way to comprehensively cover all the 202 countries and share our finding in a transparent way with the scientific community.

References rebuttal letter

- Anache, J.A.A., Wendland, E.C., Oliveira, P.T.S., Flanagan, D.C., Nearing, M.A., 2017. Runoff and soil erosion plot-scale studies under natural rainfall: A meta-analysis of the Brazilian experience. *Catena* 152, 29–39. doi:10.1016/j.catena.2017.01.003
- Arnold, J.G., Srinivasan, R., Muttiah, R.S., Williams, J.R., 1998. Large area hydrologic modeling and assesment Part I: Model development. *JAWRA J. Am. Water Resour. Assoc.* 34, 73–89. doi:10.1111/j.1752-1688.1998.tb05961.x
- Auerswald, K., Fiener, P., Martin, W., Elhaus, D., 2014. Use and misuse of the K factor equation in soil erosion modeling: An alternative equation for determining USLE nomograph soil erodibility values. *Catena* 118, 220–225. doi:10.1016/j.catena.2014.01.008
- Auerswald, K., Kainz, M., Fiener, P., 2003. Soil erosion potential of organic versus conventional farming evaluated by USLE modelling of cropping statistics for agricultural districts in Bavaria. *Soil Use Manag.* 19, 305–311. doi:10.1079/SUM2003212
- Bagarello, V., Di Piazza, G. V., Ferro, V., Giordano, G., 2008. Predicting unit plot soil loss in Sicily, south Italy. *Hydrol. Process.* 22, 586–595. doi:10.1002/hyp.6621
- Bagarello, V., Di Stefano, V., Ferro, V., Giordano, G., Iovino, M., Pampalone, V., 2012. Estimating the USLE soil erodibility factor in Sicily, South Italy. *Appl. Eng. Agric.* 28, 199–206. doi:10.13031/2013.41347
- Bagarello, V., Ferro, V., Pampalone, V., 2015. A new version of the USLE-MM for predicting bare plot soil loss at the Sparacia (South Italy) experimental site. *Hydrol. Process.* 29, 4210–4219. doi:10.1002/hyp.10486
- Bargarello, V., Ferro, V., 2004. Plot-scale measurement of soil erosion at the experimental area of Sparacia (southern Italy). *Hydrol. Process.* 18, 141–157. doi:10.1002/hyp.1318
- Boardman, J., 2006. Soil erosion science: Reflections on the limitations of current approaches. *Catena* 68, 73–86. doi:10.1016/j.catena.2006.03.007
- Boardman, J., Favis-Mortlock, D., 1998. *Modelling soil erosion by water*. Springer Science & Business Media.
- Borrelli, P., Paustian, K., Panagos, P., Jones, A., Schütt, B., Lugato, E., 2016. Effect of Good Agricultural and Environmental Conditions on erosion and soil organic carbon balance: A national case study. *Land use policy* 50, 408–421. doi:10.1016/j.landusepol.2015.09.033
- Borrelli, P., Paustian, K., Panagos, P., Jones, A., Schütt, B., Lugato, E., 2016. Effect of Good Agricultural and Environmental Conditions on erosion and soil organic carbon balance: A national case study. *Land use policy* 50, 408–421. doi:10.1016/j.landusepol.2015.09.033
- Cao, L., Zhang, K., Dai, H., Liang, Y., 2015. Modeling Interrill Erosion on Unpaved Roads in the Loess Plateau of China. *L. Degrad. Dev.* 26, 825–832. doi:10.1002/ldr.2253
- Cerdan, O., Govers, G., Le Bissonnais, Y., Van Oost, K., Poesen, J., Saby, N., Gobin, A., Vacca, A., Quinton, J., Auerswald, K., Klik, A., Kwaad, F.J.P.M., Raclot, D., Ionita, I., Rejman, J., Rousseva, S., Muxart, T., Roxo, M.J., Dostal, T., 2010. Rates and spatial variations of soil erosion in Europe: A study based on erosion plot data. *Geomorphology* 122, 167–177. doi:10.1016/j.geomorph.2010.06.011
- Cronshey, R.G., Theurer, F.D., 1998. AnnAGNPS-non-point pollutant loading model, in: *Proceedings First Federal Interagency Hydrologic Modeling Conference*. Las Vegas.
- DE ROO, A.P.J., WESSELING, C.G., RITSEMA, C.J., 1996. Lisem: a Single-Event Physically Based Hydrological and Soil Erosion Model for Drainage Basins. I: Theory, Input and Output. *Hydrol. Process.* 10, 1107–1117. doi:10.1002/(SICI)1099-1085(199608)10:8<1107::AID-HYP415>3.0.CO;2-4
- De Vente, J., Poesen, J., 2005. Predicting soil erosion and sediment yield at the basin scale : Scale issues and semi - quantitative models. *Earth - Sci. Rev.* 71, 95–125. doi:10.1016/j.earscirev.2005.02.002
- Di Stefano, C., Ferro, V., Pampalone, V., 2017. Testing the USLE-M Family of Models at the Sparacia

- Experimental Site in South Italy. *J. Hydrol. Eng.* 22, 5017012. doi:10.1061/(ASCE)HE.1943-5584.0001535
- Doetterl, S., Van Oost, K., Six, J., 2012. Towards constraining the magnitude of global agricultural sediment and soil organic carbon fluxes. *Earth Surf. Process. Landforms* 37, 642–655. doi:10.1002/esp.3198
- Evans, R., Boardman, J., 2015. The new assessment of soil loss by water erosion in Europe. Panagos P. et al., 2015 *Environmental Science & Policy* 54, 438–447—A response. *Environ. Sci. Policy* 58, 11–15.
- Favis-Mortlock, D., 2013. Non-Linear Dynamics, Self-Organization and Cellular Automata Models, in: *Environmental Modelling: Finding Simplicity in Complexity: Second Edition*. pp. 45–67. doi:10.1002/9781118351475.ch4
- Favis-Mortlock, D., 1998. Validation of field-scale soil erosion models using common datasets, in: Boardman, J., Favis-Mortlock, D. (Eds.), *Modelling Soil Erosion by Water*. Springer, Berlin, Heidelberg, pp. 89–127.
- Ferro, V., 1997. Further remarks on a distributed approach to sediment delivery. *Hydrol. Sci. J.* 42, 633–647. doi:10.1080/02626669709492063
- Ferro, V., Porto, P., 2000. Sediment Delivery Distributed (SEDD) Model. *J. Hydrol. Eng.* 5, 411–422. doi:10.1061/(asce)1084-0699(2000)5:4(411)
- García-Ruiz, J.M., Beguería, S., Nadal-Romero, E., González-Hidalgo, J.C., Lana-Renault, N., Sanjuán, Y., 2015. A meta-analysis of soil erosion rates across the world. *Geomorphology*. doi:10.1016/j.geomorph.2015.03.008
- Gessesse, B., Bewket, W., Bräuning, A., 2015. Model Based Characterization and Monitoring of Runoff and Soil Erosion in Response To Land Use/Land Cover Changes in the L. *Degrad. Dev.* 26, 711–724.
- Guo, Q., Hao, Y., Liu, B., 2015. Rates of soil erosion in China: A study based on runoff plot data. *Catena* 124, 68–76. doi:10.1016/j.catena.2014.08.013
- Ito, A., 2007. Simulated impacts of climate and land-cover change on soil erosion and implication for the carbon cycle, 1901 to 2100. *Geophys. Res. Lett.* 34. doi:10.1029/2007GL029342
- Jetten, V., Favis-Mortlock, D., 2006. Modelling soil erosion in Europe, in: Boardman, J., Poesen, J. (Eds.), *Soil Erosion in Europe*. John Wiley & Sons, pp. 695–716.
- Jetten, V., Govers, G., Hessel, R., 2003. Erosion models: Quality of spatial predictions. *Hydrol. Process.* 17, 887–900. doi:10.1002/hyp.1168
- Kinnell, P.I.A., 2014. Applying the QREI30 index within the USLE modelling environment. *Hydrol. Process.* 28, 591–598. doi:10.1002/hyp.9591
- Kirkby, M.J., Irvine, B.J., Jones, R.J.A., Govers, G., Boer, M., Cerdan, O., Daroussin, J., Gobin, A., Grimm, M., Le Bissonnais, Y., Kosmas, C., Mantel, S., Puigdefabregas, J., Van Lynden, G., 2008. The PESERA coarse scale erosion model for Europe. I. - Model rationale and implementation. *Eur. J. Soil Sci.* 59, 1293–1306. doi:10.1111/j.1365-2389.2008.01072.x
- Kirkby, M.J., Jones, R., Irvine, B., Gobin, a., Govers, G., Cerdan, O., Rompaey, A.J.J. Van, Le Bissonnais, Y., Daroussin, J., King, D., Montanarella, L., Grimm, M., Vieillefont, V., Puigdefabregas, J., Boer, M., Kosmas, C., Yassoglou, N., Tsara, M., Mantel, S., Lynden, G.J. Van, Huting, J., 2004. Pan-European Soil Erosion Risk Assessment: The PESERA Map, Version 1 October 2003. Explanation of Special Publication Ispra 2004 No.73 (S.P.I.04.73). *Eur. Soil Bur. Res. Report. Off. Off. Publ. Eur. Communities*, Luxemb. 16, 18pp. and 1 map in ISO B1 format.
- Lafren, J.M., Elliot, W.J., Flanagan, D.C., Meyer, C.R., Nearing, M. a., 1997. WEPP-predicting water erosion using a process-based model. *J. Soil Water Conserv.* 52, 96.
- Lafren, J.M., Moldenhauer, W.C., 2003. Pioneering soil erosion prediction – The USLE story, World Asso. ed. World Association of Soil and Water Conservation, Beijing, China.

- Leonard, R.A., Knisel, W.G., Still, D.A., 1987. GLEAMS: Groundwater loading effects of agricultural management systems. *Trans. ASAE* 30, 1403–1418.
- Montgomery, D.R., 2007. Soil erosion and agricultural sustainability. *Proc. Natl. Acad. Sci. U. S. A.* 104, 13268–72. doi:10.1073/pnas.0611508104
- Morgan, R.P.C., Nearing, M.A., 2000. Soil erosion models: present and future, in: Rubio J.L., Asins S., Andreu V., de Paz J.M., G.E. (Ed.), *Proceedings, Third International Congress of the European Society for Soil Conservation*. Valencia, pp. 145–164.
- Morgan, R.P.C., Quinton, J.N., Smith, R.E., Govers, G., Poesen, J.W.A., Auerswald, K., Chisci, G., Torri, D., Styczen, M.E., 1998. The European soil erosion model (EUROSEM): a process-based approach for predicting soil loss from fields and small catchments. , 527-544. *Earth Surf. Process. Landforms* 23, 527–544. doi:10.1002/(SICI)1096-9837(199806)23:6<527::AID-ESP868>3.0.CO;2-5
- Mulligan, M., Wainwright, J., 2013. Modelling and Model Building, in: *Environmental Modelling: Finding Simplicity in Complexity: Second Edition*. pp. 7–26. doi:10.1002/9781118351475.ch2
- Nearing, M.A., 2013. Soil Erosion and Conservation, in: *Environmental Modelling: Finding Simplicity in Complexity: Second Edition*. pp. 365–378. doi:10.1002/9781118351475.ch22
- Nearing, M.A., Govers, G., Norton, L.D., 1999. Variability in Soil Erosion Data from Replicated Plots. *Soil Sci. Soc. Am. J.* 63, 1829. doi:10.2136/sssaj1999.6361829x
- Nearing, M.A., Romkens, M.J.M., Norton, L.D., Stott, D.E., Rhoton, F.E., Laflen, J.M., Flanagan, D.C., Alonso, C. V, Binger, R.L., Dabney, S.M., Doering, O.C., Huang, C.H., McGregor, K.C., Simon, A., 2000. Measurements and models of soil loss rates. *Science* (80-.). 290, 1300–1301. doi:10.1126/science.290.5495.1300b
- Nearing, M.A., Yin, S.-Q., Borrelli, P., Polyakov, V.O., 2017. Rainfall erosivity: An historical review. *Catena* 157. doi:10.1016/j.catena.2017.06.004
- Oldeman, L., 1994. The global extent of soil degradation. *Soil Resil. Sustain. L. use*.
- Panagos, P., Borrelli, P., Meusburger, K., Yu, B., Klik, A., Lim, k j, Yang, j e, Ni, J., Miao, C., Chattopadhyay, N., Sadeghi, s h, Hazbavi, Z., Zabihi, M., Larionov, G.A., Krasnov, s f, Garobets, A., Levi, Y., Erpul, G., Birkel, C., Hoyos, N., Naipal, V., Oliveira, P.T.S., Bonilla, c a, Meddi, M., Nel, W., Dashti, h a, Boni, M., Diodato, N., Van Oost, K., Nearing, M.A., Ballabio, C., 2017. Global rainfall erosivity assessment based on high-temporal resolution rainfall records. *Sci. Rep.* In review, 1–12. doi:10.1038/s41598-017-04282-8
- Panagos, P., Borrelli, P., Poesen, J., Ballabio, C., Lugato, E., Meusburger, K., Montanarella, L., Alewell, C., 2015. The new assessment of soil loss by water erosion in Europe. *Environ. Sci. Policy* 54. doi:10.1016/j.envsci.2015.08.012
- Pimentel, D., Harvey, C., Resosudarmo, P., Sinclair, K., Kurz, D., McNair, M., Crist, S., Shpritz, L., Fitton, L., Saffouri, R., Blair, R., 1995. Environmental and economic costs of soil erosion and conservation. *Science* (80-.). 267, 1117–1123. doi:10.1126/science.267.5201.1117
- Quinton, J., 2013. Erosion and Sediment Transport, in: *Environmental Modelling Finding Simplicity in Complexity*. pp. 187–196.
- Quinton, J.N., Govers, G., Van Oost, K., Bardgett, R.D., 2010. The impact of agricultural soil erosion on biogeochemical cycling. *Nat. Geosci.* 3, 311–314. doi:10.1038/ngeo838
- Renard, K., Foster, G., Weesies, G., McCool, D., Yoder, D., 1997. Predicting soil erosion by water: a guide to conservation planning with the Revised Universal Soil Loss Equation (RUSLE). *Agric. Handb. No. 703*. doi:DC0-16-048938-5 65–100.
- Rüttimann, M., Schaub, D., Prasuhn, V., Rüegg, W., 1995. Measurement of runoff and soil erosion on regularly cultivated fields in Switzerland - some critical considerations. *Catena* 25, 127–139. doi:10.1016/0341-8162(95)00005-D

- Schaub, D., Prasuhn, V., 1998. A map on soil erosion on arable land as a planning tool for sustainable land use in Switzerland. *Adv. GeoEcology* 31, 161–168.
- Sonneveld, B.G.J.S., Nearing, M.A., 2003. A nonparametric/parametric analysis of the Universal Soil Loss Equation. *Catena* 52, 9–21. doi:10.1016/S0341-8162(02)00150-9
- Stolpe, N.B., 2005. A comparison of the RUSLE, EPIC and WEPP erosion models as calibrated to climate and soil of south-central Chile. *Acta Agric. Scand. Sect. B - Soil Plant Sci.* 55, 2–8. doi:10.1080/09064710510008568
- Teng, H., Viscarra Rossel, R.A., Shi, Z., Behrens, T., Chappell, A., Bui, E., 2016. Assimilating satellite imagery and visible-near infrared spectroscopy to model and map soil loss by water erosion in Australia. *Environ. Model. Softw.* 77, 156–167. doi:10.1016/j.envsoft.2015.11.024
- UNEP/RIVM/ISRIC, 1996. A Qualitative Assessment of Water Erosion Risk using 1:5 M SOTER Data. An application for Northern Argentina, South-East Brazil and Uruguay, Report 96/. ed. ISRIC, Wageningen.
- USDA, 2017a. About the Universal Soil Loss Equation [WWW Document].
- USDA, 2017b. Universal Soil Loss Equation - 2003 [WWW Document]. URL <http://www.asabe.org/awards-landmarks/asabe-historic-landmarks/universal-soil-loss-equation-41.aspx> (accessed 4.5.17).
- Van Oost, K., Quine, T. a. a, Govers, G., De Gryze, S., Six, J., Harden, J.W., Ritchie, J.C., McCarty, G.W., Heckrath, G., Kosmas, C., Giraldez, J. V, da Silva, J.R.M., Merckx, R., 2007. the Impact of Agricultural Soil Erosion on the Global Carbon Cycle. *Science* (80-.). 318, 626–9. doi:10.1126/science.1145724
- Van Rompaey, A.J.J., Verstraeten, G., Van Oost, K., Govers, G., Poesen, J., 2001. Modelling mean annual sediment yield using a distributed approach. *Earth Surf. Process. Landforms* 26, 1221–1236. doi:10.1002/esp.275
- Wainwright, J., Mulligan, M., 2013. Environmental Modelling: Finding Simplicity in Complexity: Second Edition, *Environmental Modelling: Finding Simplicity in Complexity: Second Edition*. doi:10.1002/9781118351475
- Wang, Z., Hoffmann, T., Six, J., Kaplan, J.O., Govers, G., Doetterl, S., Van Oost, K., 2017. Human-induced erosion has offset one-third of carbon emissions from land cover change. *Nat. Clim. Chang.* 7, 345–350. doi:10.1038/nclimate3263
- Wendt, R.C., Alberts, E.E., Hjelmfelt, A.T., 1986. Variability of runoff and soil loss from fallow experimental plots. *Soil Sci. Soc. Am. J.* 50, 730–736.
- Williams, J.R., Renard, K.G., Dyke, P.T., 1983. EPIC: a new method for assessing erosion's effect on soil productivity. *J. Soil Water Conserv.*
- Wischmeier, W., Smith, D.D., Wischmer, W.H., Smith, D.D., 1978. Predicting rainfall erosion losses: a guide to conservation planning. U.S. Dep. Agric. Handb. No. 537 1–69. doi:10.1029/TR039i002p00285
- Wischmeier, W.H., Smith, D.D., 1965. Predicting rainfall erosion losses in the Eastern U.S. – a guide to conservation planning, *Agricultur.* ed.
- Yue, Y., Ni, J., Ciais, P., Piao, S., Wang, T., Huang, M., Borthwick, A.G.L., Li, T., Wang, Y., Chappell, A., Van Oost, K., 2016. Lateral transport of soil carbon and land-atmosphere CO₂ flux induced by water erosion in China. *Proc. Natl. Acad. Sci.* 113, 6617–6622. doi:10.1073/pnas.1523358113

Reviewers' comments:

Reviewer #1 (Remarks to the Author):

Dear author,

I found your paper of interest

I can only suggest that in the introduction you will inform the reader that the soil erosion concerns are related to the impact of soil losses on soil quality and then the soil services, goods and resources that are definitive for the humankind development

I think this idea will bring to the reader the interest to read your paper

Soil erosion can be seeing as a pure geomorphological process but in your paper I suggest to power the environmental view that support that without soils there will be no life in the Earth, but neither the humanity

You can read the following papers to be inspired

Keesstra, S. D., Bouma, J., Wallinga, J., Tittonell, P., Smith, P., Cerdà, A., Montanarella, L., Quinton, J. N., Pachepsky, Y., van der Putten, W. H., Bardgett, R. D., Moolenaar, S., Mol, G., Jansen, B., and Fresco, L. O. 2016. The significance of soils and soil science towards realization of the United Nations Sustainable Development Goals, *SOIL*, 2, 111-128, doi:10.5194/soil-2-111-2016.

Mol, G., Keesstra, S.D., 2012. Soil science in a changing world. *Current Opinions in Environmental Sustainability* 4: 473–477.

Reviewer #3 (Remarks to the Author):

I have evaluate the authors response to my comments:

1. comment 1: to include other developing countries other than China and India was not done because it is too much to do. I agree .

2. comment 2: include some of the sustainable development goals, but not done. This is of course a minor point and I am happy to ignore it.

decision: PUBLISH WITHOUT AMENDMENTS.

Reviewer #4 (Remarks to the Author):

I have read the manuscript, all reviewers' comments and rebuttals. As the 5th reviewer I think it is not useful to focus on the many details that have been discussed extensively already and the responses of the authors. I think the authors have done a remarkable job, tackling a very difficult subject of soil erosion on a global scale. There are many constraints to such an analysis, but in conclusion I think it is worthwhile to publish. Previous global erosion studies have a much lower resolution and this study adds new insights.

In my opinion, the acceptance of publication hinges on three things: i) can it be made acceptable that the method and data used are acceptable at this scale and resolution, and ii) if the method is acceptable, what exactly is calculated by it, and iii) how uncertain are the estimates spatially.

i) Is this method acceptable at this scale? Basically there are two kinds of erosion models as the authors mention. First there are process models that are event based, that simulate the hydrology and sediment dynamics during and after a rainfall event. Obviously these are not useful for this study. Annual global erosion is not a matter of individual events. This also limits the scope of this study, namely the absence of including extreme events (see remark below in point ii). Second there are annual erosion models that are based on a multiple regression equation of predicted and observed sediment loss from small plots, USLE type models and derivatives. The USLE type models are sediment production models, they do not simulate sediment transport and deposition. One of the exceptions is the Morgan Morgan and Finney model that gives an annual hydrological response and couples this to detachment transport and deposition of sediment. However the parameters of this model are not reproducible at a global scale. The USLE type models have been used and misused so many times, that you can find literally 100s of papers showing the pros and cons and find as many experts. However, since the RUSLE has not been done at this scope, literature references are pretty useless as a support of its validity. The only way to prove the validity of the method is to argue conceptually for each factor why this is appropriate for the 250m resolution used. This is as much an

argument for the method (equations) as for the data used. The data and the intermediate steps to arrive from the basic data to the factors in the USLE deserve more attention.

The global scale is a data problem, not a model problem, as the model is executed at 90m resolution for the DEM related factors, and at 250m resolution for the other factors. The global aspect is just repeating the calculations for all cells on the globe. The only true spatial processes are the erosivity based on rainfall and its interpolation, and the 2D LS factor. The erodibility and plant related factors are cell by cell 1D and depend only on data quality for this resolution.

The argument the authors could make in the main article could be along the line of the following:

Conceptually soil erosion is product of a driving factor (rainfall and runoff energy or shear stress) and a soil strength factor (an empirical factor or a physical factor such as cohesion), whereby vegetation exerts a mitigating effect (strengthening the soil, protection against splash and slowing down runoff), and soil conservation measures exert a mitigating effect because they affect these processes. Both process models and empirical models use this principle. If we accept this, it is a matter of combining the correct data and algorithms at this scale and resolution. This is as much a data question as a list of equations, which are by themselves not very explanatory at this scale/resolution. I would therefore show for each of the factors the global map (erosivity, erodibility, LS, C factor). This would improve the credibility of the method.

Rainfall erosivity: there are two potential weaknesses in the manuscript. First the spatial distribution and interpolation of the rainfall stations, the method is mentioned but the results of the GPR algorithm are not shown and no insight is given in the quality of the predictions. The density of the African continent stations is very low, but substantial erosion is shown here. How good is this prediction? This should be quantified in some way. Supp Fig 7 gives an overall sensitivity analysis, but given the scarcity of points in Africa for instance, the overall sensitivity doesn't say much about the spatial uncertainty.

Second is equation 7 by Brown and Foster 1987. I think the reference is missing! (I assume it is: Brown, L.C. and Foster, G.R. (1987) Storm Erosivity Using Idealized Intensity Distribution. Transactions of the ASAE, 30, 379-386.). The research seems relatively old and I imagine something better can be found for rainfall energy. For instance https://www.researchgate.net/profile/Albert_Van_Dijk/publication/222832607_Rainfall_intensity-kinetic_energy_relationships_A_critical_literature_appraisal/links/09e4150b461f20ace2000000/Rainfall-intensity-kinetic-energy-relationships-A-critical-literature-appraisal.pdf is an excellent review that leads to a different equation. The authors should give some proof that the Foster and Brown equation is an acceptable equation as a global average. Also there are many attempts to calculate the R factor for large parts of continents and it would be good to see how the proposed method compares to those.

The runoff is related to the LS factor. This has the potential problem of being over complicated. In reality water does not flow very long in a landscape and often ends at field boundaries. I cannot judge how the 2D algorithm works out in the various parts of the world but I cannot see how the aspect X of a gridcell in eq 8 conceptually relates to erosion, and equations 9 and 10 with factors m and beta are probably related to the loess belt in Belgium (this is a guess?) and it is questionable whether this is usable for the entire globe. The risk of replacing 1D LS factor to a 2D LS factor is overestimating the effect of the relief. The sensitivity analysis shows that the LS goes to a value to 25 and that it is very sensitive. Of course many flat areas in the world have a low LS so these cells dominate, but the cells with a high value give the highest erosion which is the point of the article, so extra care should be taken with this.

Did the authors impose a maximum value for LS to avoid the accumulation to continue for large catchments? If the upstream area is allowed to become very large so that it conceptually loses all relation to rill erosion, this is not an acceptable method. The authors need to specify the implementation of these equations, in particular the accumulation constraints. Also here a global LS map would be helpful.

I have no comments of the K factor and the C factors, the authors have done a good job using the latest databases, and the careful application of them

ii) All this relates directly to the what the authors are calculating: in my opinion, since there is no transport and deposition, the end-map shows the sediment production by rill erosion from small fields inside a gridcell, where it is assumed that those fields can be characterized by the 250m resolution properties. This sediment doesn't go anywhere, part of flows into a river system, part ends up as deposition in lower parts of the landscape. Possibly the deposition is positive for the accumulation of soil, but this cannot be expressed at a global scale. Annual erosion models are known to be bad in predicting the effects of extreme events in a year. This means that the resulting values are averages and that the many extreme events at a global scale (cyclones, tropical storms, la Nina years) are ignored. Also the effects of droughts (el Nino) which affects the vegetation cover and promotes overgrazing etc, are ignored. The erosion of an average year following an el Nino year is very different from the erosion of an average year following a la Nina year. It may well be that extreme events have much more erosion than the years in between events. I have been engaged in flood studies with tropical storms that had 500mm in two days, with intensities of more than 200mm/h! The amount of sediment displaced in such events is staggering. This type of events are completely outside the scope of this method. As such the authors should be VERY careful in stating anything related to climate change, as climate change is not only an increase in trend but more an increase in climate variability. The method adopted here is not appropriate for this.

I therefore think that the statement "Unlike previous studies which dealt with soil erosion as a static process, here we shed light on the impacts of the 21st century global land use change on soil erosion." Should be made VERY carefully, with a lot of constraints as to what the authors do show and do not show. This is not a complete land degradation map with all soil displacement processes included, only the type of 'accelerated' erosion that is the result of land use/land cover changes by

agriculture. I am not sure whether animal husbandry (overgrazing) is sufficiently included in this analysis.

iii) uncertainty. I would like to draw a parallel to the SREX IPCC report where climate change is quantified but also an indication is given into the certainty of a given change: winter are warmer in region X with great certainty, or higher rainfall is expected in region Y but this conclusion has a low certainty. I find this a very careful way of expression and I think this article merits a similar approach. Specifically the main driver, rainfall does not have the same spatial certainty everywhere. The same goes for the soil data, SOILGRIDS has information on the validity of the estimates. For the other factors: the SRTM has the same uncertainty everywhere (possibly more uncertain in flat areas where erosion is not a problem) and the LULC data is also similar in uncertainty everywhere (?). If a map could be produced with the certainty of the estimate in classes, this would greatly increase the value of the article, and also highlight a different but very real problem: the need for good ground data in certain areas.

Summarizing:

The article is acceptable for publication if: i) proof is given in the supplementary note 1 how the algorithms in combination with the data and data manipulation steps are conceptually correct to be used at this resolution for this process, supported with maps of the individual factors, and ii) if the authors in the main document very clearly state what the end results means, what it shows and what it not shows (both in terms of erosion processes and in terms of circumstances leading to erosion), and if possible iii) an uncertainty estimate could be produced for the soil erosion on a global scale.

Referee #1

Comment. *Dear author, I found your paper of interest.*

I can only suggest that in the introduction you will inform the reader that the soil erosion concerns are related to the impact of soil losses on soil quality and then the soil services, goods and resources that are definitive for the humankind development. I think this idea will bring to the reader the interest to read your paper. Soil erosion can be seeing as a pure geomorphological process but in your paper I suggest to power the environmental view that support that without soils there will be no life in the Earth, but neither the humanity. You can read the following papers to be inspired

*Keesstra, S. D., Bouma, J., Wallinga, J., Tiftonell, P., Smith, P., Cerdà, A., Montanarella, L., Quinton, J. N., Pachepsky, Y., van der Putten, W. H., Bardgett, R. D., Moolenaar, S., Mol, G., Jansen, B., and Fresco, L. O. 2016. The significance of soils and soil science towards realization of the United Nations Sustainable Development Goals, *SOIL*, 2, 111-128, doi:10.5194/soil-2-111-2016.*

*Mol, G., Keesstra, S.D., 2012. Soil science in a changing world. *Current Opinions in Environmental Sustainability* 4: 473–477.*

Reply. We thank Referee #1 for the appreciative comments and suggestions. The point raised is very important and we agree that the Introduction Chapter would benefit from a concise but comprehensive description of the importance of soil and its key functions.

We suggest to start the Introduction Chapter with the following paragraph describing the importance of soil and the services that it provides to the society. The text is provided together with bibliographical references that can help readers who are interested to deepen their knowledge about these aspects.

Modification to the main text:

Introduction

'Healthy soil is the foundation of agriculture and an essential resource to ensure human needs in the 21st century¹, such as food, feed, fibre, clean water and clean air. It is a vital part of ecosystems and earth system functions that support the delivery of primary 'ecosystem services^{2,3}.

The latest reference document of the United Nations (UN) on the status of global soil resources stresses that '...the majority of the world's soil resources are in only fair, poor or very poor condition⁴. The results of the meta-analysis...'

Added references:

1. Amundson, R. *et al.* Soil and human security in the 21st century. *Science* (80-.). **348**, 1261071–1261071 (2015).
2. Robinson, D. A. *et al.* Soil natural capital in Europe; A framework for state and change assessment. *Sci. Rep.* **7**, (2017).
3. Keesstra, S. D. *et al.* The significance of soils and soil science towards realization of the United Nations Sustainable Development Goals. *SOIL* **2**, 111–128 (2016).

Referee #3

Comment.

I have evaluate the authors response to my comments:

1. Comment 1: to include other developing countries other then China and India was not done because it is too much to do. I agree.

2. Comment 2: include some of the sustainable development goals, but not done. This is of course a minor point and i am happy ignore it.

Decision: PUBLISH WITHOUT AMENDMENTS.

Reply. We thank Referee #3 for reviewing the revised version of the manuscript and for his positive feedback.

Comment.

I have read the manuscript, all reviewers comments and rebuttals. As the 5th reviewer I think it is not useful to focus on the many details that have been discussed extensively already and the responses of the authors. I think the authors have done a remarkable job, tackling a very difficult subject of soil erosion on a global scale. There are many constraints to such an analysis, but in conclusion I think it is worthwhile to publish. Previous global erosion studies have a much lower resolution and this study adds new insights.

In my opinion, the acceptance of publication hinges on three things: i) can it be made acceptable that the method and data used are acceptable at this scale and resolution, and ii) if the method is acceptable, what exactly is calculated by it, and iii) how uncertain are the estimates spatially. i) Is this method acceptable at this scale? Basically there are two kinds of erosion models as the authors mention. First there are process models that are event based, that simulate the hydrology and sediment dynamics during and after a rainfall event. Obviously these are not useful for this study. Annual global erosion is not a matter of individual events. This also limits the scope of this study, namely the absence of including extreme events (see remark below in point ii). Second there are annual erosion models that are based on a multiple regression equation of predicted and observed sediment loss from small plots, USLE type models and derivatives. The USLE type models are sediment production models, they do not simulate sediment transport and deposition. One of the exceptions is the Morgan Morgan and Finney model that gives an annual hydrological response and couples this to detachment transport and deposition of sediment. However the parameters of this model are not reproducible at a global scale. The USLE type models have been used and misused so many times, that you can find literally 100s of papers showing the pros and cons and find as many experts. However, since the RUSLE has not been done at this scope, literature references are pretty useless as a support of its validity. The only way to prove the validity of the method is to argue conceptually for each factor why this is appropriate for the 250m resolution used. This is as much an argument for the method (equations) as for the data used. The data and the intermediate steps to arrive from the basic data to the factors in the USLE deserve more attention.

The global scale is a data problem, not a model problem, as the model is executed at 90m resolution for the DEM related factors, and at 250m resolution for the other factors. The global aspect is just repeating the calculations for all cells on the globe. The only true spatial processes are the erosivity based on rainfall and its interpolation, and the 2D LS factor. The erodibility and plant related factors are cell by cell 1D and depend only on data quality for this resolution. The argument the authors could make in the main article could be along the line of the following: Conceptually soil erosion is product of a driving factor (rainfall and runoff energy or shear stress) and a soil strength factor (an empirical factor or a physical factor such as cohesion), whereby vegetation exerts a mitigating effect (strengthening the soil, protection against splash and slowing down runoff), and soil conservation measures exert a mitigating effect because they affect these processes. Both process models and empirical models use this principle. If we accept this, it is a matter of combining the correct data and algorithms at this scale and resolution. This is as much a data question as a list of equations, which are by themselves not very explanatory at this scale/resolution. I would therefore show for each of the factors the global map (erosivity, erodibility, LS, C factor). This would improve the credibility of the method.

Rainfall erosivity: there are two potential weaknesses in the manuscript. First the spatial distribution and interpolation of the rainfall stations, the method is mentioned but the results of the GPR algorithm are not shown and no insight is given in the quality of the predictions. The density of the African continent stations is very low, but substantial erosion is shown here. How good is this prediction? This should be quantified in some way. Supp Fig 7 gives an overall sensitivity analysis, but given the scarcity of points in Africa for instance, the overall sensitivity doesn't say much about the spatial uncertainty.

Second is equation 7 by Brown and Foster 1987. I think the reference is missing! (I assume it is: Brown, L.C. and Foster, G.R. (1987) Storm Erosivity Using Idealized Intensity Distribution. Transactions of the ASAE, 30, 379-386.). The research seems relatively old and I imagine something better can be found for rainfall energy. For instance https://www.researchgate.net/profile/Albert_Van_Dijk/publication/222832607_Rainfall_intensity-kinetic_energy_relationships_A_critical_literature_appraisal/links/09e4150b461f20ace2000000/Rainfall-intensity-kinetic-energy-relationships-A-critical-literature-appraisal.pdf is an excellent review that leads to a different equation. The authors should give some proof that the Foster and Brown equation is an acceptable equation as a global average. Also there are many attempts to calculate the R factor for large parts of continents and it would be good to see how the proposed method compares to those.

The runoff is related to the LS factor. This has the potential problem of being over complicated. In reality water does not flow very long in a landscape and often ends at field boundaries. I cannot judge how the 2D algorithm works out in the various parts of the world but I cannot see how the aspect X of a gridcell in eq 8 conceptually relates to erosion, and equations 9 and 10 with factors m and beta are probably related to the loess belt in Belgium (this is a guess?) and it is questionable whether this is usable for the entire globe. The risk of replacing 1D LS factor to a 2D LS factor is overestimating the effect of the relief. The sensitivity analysis shows that the LS goes to a value to 25 and that it is very sensitive. Of course many flat areas in the world have a low LS so these cells dominate, but the cells with a high value give the highest erosion which is the point of the article, so extra care should be taken with this.

Did the authors impose a maximum value for LS to avoid the accumulation to continue for large catchments? If the upstream area is allowed to become very large so that it conceptually loses all relation to rill erosion, this is not an acceptable method. The authors need to specify the implementation of these equations, in particular the accumulation constraints. Also here a global LS map would be helpful.

I have no comments of the K factor and the C factors, the authors have done a good job using the latest databases, and the careful application of them.

ii) All this relates directly to the what the authors are calculating: in my opinion, since there is no transport and deposition, the end-map shows the sediment production by rill erosion from small fields inside a gridcell, where it is assumed that those fields can be characterized by the 250m resolution properties. This sediment doesn't go anywhere, part of flows into a river system, part ends up as deposition in lower parts of the landscape. Possibly the deposition is positive for the accumulation of soil, but this cannot be expressed at a global scale. Annual erosion models are known to be bad in predicting the effects of extreme events in a year. This means that the resulting values are averages and that the many extreme events at a global scale (cyclones, tropical storms, la Nina years) are ignored. Also the effects of droughts (el Nino) which affects the vegetation cover and promotes overgrazing etc, are ignored. The erosion of an average year following an el Nino year is very different

from the erosion of an average year following a la Nina year. It may well be that extreme events have much more erosion than the years in between events. I have been engaged in flood studies with tropical storms that had 500mm in two days, with intensities of more than 200mm/h! The amount of sediment displaced in such events is staggering. This type of events are completely outside the scope of this method. As such the authors should be VERY careful in stating anything related to climate change, as climate change is not only an increase in trend but more an increase in climate variability. The method adopted here is not appropriate for this.

I therefore think that the statement "Unlike previous studies which dealt with soil erosion as a static process, here we shed light on the impacts of the 21st century global land use change on soil erosion." Should be made VERY carefully, with a lot of constraints as to what the authors do show and do not show. This is not a complete land degradation map with all soil displacement processes included, only the type of 'accelerated' erosion that is the result of land use/land cover changes by agriculture. I am not sure whether animal husbandry (overgrazing) is sufficiently included in this analysis. iii) uncertainty. I would like to draw a parallel to the SREX IPCC report where climate change is quantified but also an indication is given into the certainty of a given change: winter are warmer in region X with great certainty, or higher rainfall is expected in region Y but this conclusion has a low certainty. I find this a very careful way of expression and I think this article merits a similar approach. Specifically the main driver, rainfall does not have the same spatial certainty everywhere. The same goes for the soil data, SOILGRIDS has information on the validity of the estimates. For the other factors: the SRTM has the same uncertainty everywhere (possibly more uncertain in flat areas where erosion is not a problem) and the LULC data is also similar in uncertainty everywhere (?). If a map could be produced with the certainty of the estimate in classes, this would greatly increase the value of the article, and also highlight a different but very real problem: the need for good ground data in certain areas.

Summarizing: The article is acceptable for publication if: i) proof is given in the supplementary note 1 how the algorithms in combination with the data and data manipulation steps are conceptually correct to be used at this resolution for this process, supported with maps of the individual factors, and ii) if the authors in the main document very clearly state what the end results means, what it shows and what it not shows (both in terms of erosion processes and in terms of circumstances leading to erosion), and if possible iii) an uncertainty estimate could be produced for the soil erosion on a global scale.

Reply. We thank Referee #4 for the appreciative comments and suggestions. Referee #4 provided a comprehensive and positive evaluation of the study. Before publication, however, Referee #4 suggests that some methodological clarifications should be provided, together with further descriptions of the model in the main text, some new figures and an uncertainty estimation of the spatial predictions of the global model.

All the requests made by Referee #4 were addressed and both the manuscript and the Supplementary Information were modified following the advice. Please, find below a detailed reply to each individual point or comment.

Comment (i), 'are the method and data used acceptable at this scale (?).'

Referee #4 correctly highlighted that considering the current state-of-the-art of RUSLE applications 'The global scale is a data problem, not a model problem, as the model is executed at 90m resolution

for the DEM related factors, and at 250m resolution for the other factors. The global aspect is just repeating the calculations for all cells on the globe.'

The previous version of our main manuscript did not contain a section about the reproducibility of our GIS-based model RUSLE at global scale. We agree with Referee #4, at this stage of the RUSLE development 'The global scale is a data problem, not a model problem'. Previous regional studies in Europe (Panagos et al., 2015)¹, Australia (Teng et al., 2016)², China (Yue et al., 2016)³ or at global scale (Doetterl et al., 2012⁴; Ito, 2007⁵; Quinton et al., 2010⁶; Van Oost et al., 2007⁷) already proved that the model upscaling is feasible.

However, in order to create meaningful global RUSLE-based estimates, the scale of the input data employed must be congruent with the one of the original experimentations and the physical parameters.

Following the suggestion provided by Referee #4, the following changes were implemented in the **Methods** section of the **main text**:

Modification to the main text:

Methods

Study area. The study area includes the area of the 202 countries for which FAOSTAT¹⁴ provides crops statistics. The total modelled area is about 125 million km², providing living space for a population of about 7.5 billion people.

Soil erosion modelling. The years 2001 and 2012 form the reference periods to assess the 21st century human-induced soil erosion by water erosion at a global scale. **For 21st century human-induced soil erosion we refer to the effects caused by land use / land cover changes. Permanent loss and gain of global croplands, forests and semi-natural grass vegetation are considered in the modelling scheme while the effects of grazing and the establishment of new pasturelands are implicitly reflected. Short-term effects of land use / land cover change (i.e., forest/rangeland fires and wood harvesting) and overgrazing are not modelled. Climate change and human-induced effects on climate are also not considered.**

The long-term annual soil erosion rates for the two different land covers are estimated using an improved large-scale **Geographic Information System (GIS)** version of the **Revised Universal Soil Loss Equation (RUSLE)** model (Renard et al., 1997) (Supplementary Fig. S1). RUSLE belongs to the so called detachment-limited model types where the soil erosion (expressed as a mass of soil lost per unit area and time, Mg ha⁻¹ yr⁻¹) due to inter-rill and rill erosion processes is given by the multiplication of six contributing factors. **Consistently with the predictive capacity of the model,**

¹ Panagos, P. et al. The new assessment of soil loss by water erosion in Europe. *Environ. Sci. Policy* **54**, 438–447 (2015).

² Teng, H. et al. Assimilating satellite imagery and visible-near infrared spectroscopy to model and map soil loss by water erosion in Australia. *Environ. Model. Softw.* **77**, 156–167 (2016).

³ Yue, Y. et al. Lateral transport of soil carbon and land-atmosphere CO₂ flux induced by water erosion in China. *Proc. Natl. Acad. Sci.* **113**, 6617–6622 (2016).

⁴ Doetterl, S., Van Oost, K. & Six, J. Towards constraining the magnitude of global agricultural sediment and soil organic carbon fluxes. *Earth Surf. Process. Landforms* **37**, 642–655 (2012).

⁵ Ito, A. Simulated impacts of climate and land-cover change on soil erosion and implication for the carbon cycle, 1901 to 2100. *Geophys. Res. Lett.* **34**, (2007).

⁶ Quinton, J. N., Govers, G., Van Oost, K. & Bardgett, R. D. The impact of agricultural soil erosion on biogeochemical cycling. *Nat. Geosci.* **3**, 311–314 (2010).

⁷ Van Oost, K. et al. The Impact of Agricultural Soil Erosion on the Global Carbon Cycle. *Science* **318**, 626–9 (2007).

soil displacement due to processes such as gully and tillage erosion is not estimated.

RUSLE-type models have demonstrated to be able to reduce a very complex system to a quite simple one for the purposes of erosion prediction (*Nearing, 2013*) while maintaining a thorough representation of the main environmental and anthropogenic factors that influence the process (*Renard et al., 1997*). Conceptually, these models follow the same principle of complex process-based models, with a driving force (erosivity of the climate), a resistance term (erodibility of the soil) and the other factors representing the 'farming choice', i.e. topographical conformation of the field (LS), cropping system (C) and soil conservation practices (P). Field- and catchment-scale experiments that compared the prediction capacity of empirical RUSLE-type models with process-based models did not reveal a substantial difference between the predictive performances of the two modelling approaches (*Jetten et al., 2003; Nearing, 2013*). This, together with the moderate data demand of RUSLE-type models has facilitated the process of upscaling. Today, an extensive amount of the literature recommends the use of RUSLE-based models to provide spatially explicit estimations of soil erosion in GIS environments.

In the original version of RUSLE (*Renard et al., 1997*), the region of the model simulation is a specific field plot slope with given size, rainfall pattern, soil conditions, topography, crop system and management practices. Before the introduction of GIS-based computational techniques, the input data employed to run the model were generally directly measured in the field and afterwards imported in a specific software. In this study, a simplified application of the RUSLE model at global scale is proposed. For this purpose, the approach paved by previous GIS-based studies dealing with upscaling procedures to extent the applicability of the model as well as regional studies in Europe (*Panagos et al., 2015*), Australia (*Teng et al., 2016*) and China (*Yue et al., 2016*) was followed. The rates of soil displacement by water erosion are estimated through the GIS raster scheme. Using a GIS raster scheme applied to the USLE model means hypothesizing that each cell is independent from the others with respect to soil erosion. Soil erosion (synonym to RUSLE soil loss, Supplementary Note 2) refers to the amount of sediment that reaches the end of a specified area (cell) on a hillslope that experiences loss of soil by water erosion. The modelled area does not, in any way, include areas of the slope that experience net deposition over the long term. This is because the displaced soil amount is not routed downslope across each cell from hillslopes to the sink area or the riverine systems through a transport/deposition capacity module.

Conceptually, the global modelling presented in this study is based on the assumption that if a catchment- or a regional-scale application of RUSLE can predict meaningful soil loss estimates, the application of the model at larger-scale will provide meaningful estimates as long as the scale of the input data employed is congruent with the scale used for the estimation of the modelling factors and local applications. If the scale of the input data meets these requisites, the global scope of the model application represents a consistent repetition in space of the calculation for all cells in the modelled area.

Although the input factors show a lower spatial detail in this global-scale study compared to the recent study in Europe (*Panagos et al., 2015*), the data and the scale used are adequate to ensure a spatial description of the modelling factors and are therefore within the boundaries of the model's applicability. The characterization of land use / land cover change is based on NASA's Moderate Resolution Imaging Spectroradiometer (MODIS), which are the most consistent data currently available for soil erosion modelling at global scale (*Chappell and Webb, 2016*). With a cell size of 250 x 250m spatial resolution, the dimension of each individual land unit (equal to 6.25 ha) fairly describe the general dynamics of landscape fragmentation and can adequately represent the land unit originally targeted by the model, i.e., the US arable fields which show an average size of

about 400 acres (162 ha). For the analysis of the effect of the topography a SRTM 3 arc-seconds (ca. 90m) spatial resolution was used. This ca. 90m DEM ensured the computation of the combined topographical factor maintaining a scale congruent with the one used during the USLE's experimental measurements with plots having a length less than or equal to 122 m. Rainfall and soil characteristics were obtained using the best database available to adequately describe their pattern and dynamics during the elaboration of this study. Both have a spatial resolution of ca. 1 km.

Other methodological clarifications and improvements related to the comment (i):

Rainfall erosivity assessments. The Gaussian Process Regression (GPR) interpolation algorithm, the resulting map and the quality of the spatial predictions are described and discussed in detail in a study that we recently published (Panagos et al., 2017)⁸.

In order to avoid duplication of work, we suggest to refrain from adopting the description of the computation of the rainfall erosivity synthetic of this study. Instead, we recommend to refer to the work of Panagos et al. (2017)⁸ for comprehensive results and discussions (including uncertainty). With regard to the request to add a figure about each one of the main final modelling factors, please find below the new figure (Supplementary Fig S3) that we prepared. As suggested, similar figures were also produced for the input factors C (Supplementary Fig S10), LS (Supplementary Fig S13) and K (Supplementary Fig S14). Please find more details at the end of this letter.

⁸ Panagos, P. Borrelli, P. et al. Global rainfall erosivity assessment based on high-temporal resolution rainfall records. *Sci. Rep.* **7**, (2017).

Supplementary Fig S3 Global rainfall erosivity map. (a) Representation of the global patterns of the rainfall erosivity R-factor (spatial resolution 30 arc-seconds, ca. 1 x 1km). (b) Subset of the global rainfall erosivity map for an area of about 45,000 km² in the West Central Region of Brazil. (c) R-factor average value per continent.

Equation of the rainfall kinetic energy (KE). Yes, as stated in the Supplementary Information we used the equation proposed by Brown and Foster (1987) (reference list updated). The global rainfall erosivity factor was computed using a database derived from the collaboration of ca. 30 scientists from all continents. As a matter of fact, most of the contributing authors in the previous studies computed the rainfall kinetic energy using the equation proposed by Brown and Foster (1987):

'...>97.7% of the calculated R-factor stations are based on the original RUSLE equation.'
(Panagos et al., 2017).

We are aware that some large scale studies (e.g., Teng et al., 2016 in Australia)⁹ used the equation proposed by van Dijk et al. (2002). However, the argument for the estimation of the rainfall kinetic energy has been discussed quite controversial (Lobo and Bonilla, 2015)¹⁰. It is estimated that 14 different exponential KE-I relationships (calibrated at different sites worldwide) are currently

⁹ Teng, H., Rossel, R. A. V., Shi, Z., Behrens, T., Chappell, A., & Bui, E. (2016). Assimilating satellite imagery and visible–near infrared spectroscopy to model and map soil loss by water erosion in Australia. *Environmental Modelling & Software*, 77, 156-167.

¹⁰ Lobo, G. P., & Bonilla, C. A. (2015). Sensitivity analysis of kinetic energy-intensity relationships and maximum rainfall intensities on rainfall erosivity using a long-term precipitation dataset. *Journal of hydrology*, 527, 788-793.

available in literature. Our colleague Santiago Beguería, together with his research team, tested these 14 equation in a catchment in Europe (Angulo-Martínez et al., 2016)¹¹ and concluded:

'In our study of the inner Ebro Basin, the KE-I equation of Brown and Foster (1987) performed quite well for predicting total KE, but this result should not be generalized to other regions.'

In a study that we recently published (Nearing, M. A., Yin, S. Q., Borrelli, P., & Polyakov, V. O. (2017). Rainfall erosivity: An historical review. *Catena*, 157, 357-362) as well as in another paper that is currently in the review process (Yin, S., Nearing, M.A., Borrelli, P., Xue, X. Rainfall erosivity: a review. *Vadose Zone Journal*) we also participated in this debate to discuss the possible role of the RUSLE2 equation¹².

However, the very vast majority of local studies and scientists who contributed to our global mapping used the original RUSLE equation (97.7% of the GloREda database) proposed by Brown and Foster (1987). In many cases, this decision is supported by local observations and methodological comparisons. Accordingly, we believe that for the aim of this study a wise and commonly supported scientific basis has been used. Nevertheless, we agree with the Referee that it is an interesting topic and future studies on global rainfall erosivity should compare the rainfall kinetic energy equations to estimate the spatial resulting variability of the R-factor (Nearing et al., 2017).

The computation of the LS factor. We would like to ensure Referee #4 that a computation consistent with the state-of-the-art was conducted. We are aware that in reality the flow is not 'unlimited' as it could result from the manipulation of the DEM. In a recent study in Italy (Borrelli et al., 2016)¹³, we used remote sensing techniques to introduce the 'field boundaries/channelled flow concept' for the first time in a national scale RUSLE application.

As the Referee #4 correctly stated, to avoid possible overestimations of the LS factor with the 2D approach a max theoretically L was imposed (122m, consistently with the USLE experimentations that were carried out in plots having a length less than or equal to 122m). Please find below an example of the LS factor computed for the global modelling using the 'deterministic infinity algorithm'¹⁴ and a max L (Figure B) with one computed with a '8D flow' algorithm and without further constraints (Figure C).

Please also note that the Supplementary Information Chapter dealing with the description of the LS factor computation was modified adding the following sentence:

Modification to the Supplementary Information:

'The flow accumulation was computed using the deterministic infinity algorithm'⁷³ (Tarboton et al., 1997) and a maximum hillslope length set to 122m. The use of a 90m DEM ensured the

¹¹ Angulo-Martínez, M., Beguería, S., & Kyselý, J. (2016). Use of disdrometer data to evaluate the relationship of rainfall kinetic energy and intensity (KE-I). *Science of the Total Environment*, 568, 83-94.

¹² McGregor, K.C., R.L. Bingner, A.J. Bowie, and G.R. Foster. 1995. Erosivity index values for northern Mississippi. *Trans. ASAE*. 38(4): 1039-1047.

¹³ Borrelli, P., Paustian, K., Panagos, P., Jones, A., Schütt, B., & Lugato, E. (2016). Effect of good agricultural and environmental conditions on erosion and soil organic carbon balance: a national case study. *Land Use Policy* 50, 408-421.

¹⁴ Tarboton, D.G. A new method for the determination of flow directions and upslope areas in grid digital elevation models, *Water Resources Research*, 2, (1997).

computation of the combined LS topographical factor maintaining a scale congruent with the one used during the USLE's experimental measurements with plots having a length less than or equal to 122 m.'

Figure. Comparison between LS computations. **A**, Comparison site a hilly area of Tuscany, Italy. **B**, the LS computed using the equation employed in this study. **C**, Application of a 2D algorithm for the computation of LS using an 8D flow algorithm and unconstrained L factor. For this exercise the EU-DEM with 25m cell size was used. For a comparison between 25 and 90m DEM LS-factor we refer to Panagos, Borrelli et al. 2015 (Geosciences 5, pp. 117-126).

Comment (ii), 'what is actually calculated by the model (?)'

Also in this case Referee #4 correctly highlights an important aspect: the importance of '...stating in the main document very clearly what the end results means, what it shows and what it not shows...'.

In the previous version of the manuscripts, the definition of soil erosion was provided in the Supplementary Note 2. Here, we presented the definition that our first author provided in a recent study (Nearing, Yin, Borrelli, & Polyakov, 2017)¹⁵.

We agree that a clear definition of this modelling aspect is also essential in the main text. The following changes in the 'Soil erosion modelling Chapter' deal with these aspects:

Modification to the main text:

Definition of the 21st century human-induced assessment. What is accounted and what is not.

Soil erosion modelling. The years 2001 and 2012 form the reference periods to assess the 21st century human-induced soil erosion by water erosion at a global scale. For 21st century human-

¹⁵ Nearing, M. A., Yin, S. Q., Borrelli, P., & Polyakov, V. O. (2017). Rainfall erosivity: An historical review. Catena, 157, 357-362.

induced soil erosion we refer to the effects caused by land use / land cover changes. Permanent loss and gain of global croplands, forests and semi-natural grass vegetation are considered in the modelling scheme while the effects of grazing and the establishment of new pasturelands are implicitly reflected. Short-term effects of land use / land cover change (i.e., forest/rangeland fires and wood harvesting) and overgrazing are not modelled. Climate change and human-induced effects on climate are also not considered.

Soil erosion processes modelled.

RUSLE belongs to the so called detachment-limited model types where the soil erosion (expressed as a mass of soil lost per unit area and time, $\text{Mg ha}^{-1} \text{yr}^{-1}$) due to inter-rill and rill erosion processes is given by the multiplication of six contributing factors. Consistently with the predictive capacity of the model, soil displacement due to processes such as gullying and tillage erosion is not estimated.

Highlight the definition of soil erosion in the study:

In this study, a simplified application of the RUSLE model at global scale is proposed. For this purpose, the approach paved by previous GIS-based studies dealing with upscaling procedures to extent the applicability of the model as well as regional studies in Europe (Panagos et al., 2015), Australia (Teng et al., 2016) and China (Yue et al., 2016) was followed. The rates of soil displacement by water erosion are estimated through the GIS raster scheme. Using a GIS raster scheme applied to the USLE model means hypothesizing that each cell is independent from the others with respect to soil erosion. Soil erosion (synonym to RUSLE soil loss, Supplementary Note 2) refers to the amount of sediment that reaches the end of a specified area (cell) on a hillslope that experiences loss of soil by water erosion. The modelled area does not, in any way, include areas of the slope that experience net deposition over the long term. This is because the displaced soil amount is not routed downslope across each cell from hillslopes to the sink area or the riverine systems through a transport/deposition capacity module.

Moreover, please note that we also rephrased the sentence reported in the Introduction that could led to misunderstanding '*Unlike previous studies which dealt with soil erosion as a static process, here we shed light on the impacts of the 21st century global land use change on soil erosion.*':

Phrase rewording (Introduction):

'In this study we provide quantitative, thorough estimates of soil erosion at the global scale by means of an unprecedented high-resolution spatially distributed RUSLE-based⁵ modelling approach. The study area includes the area of the 202 countries equal to a total modelled surface of about 125 million km^2 . ~~Unlike previous studies which dealt with soil erosion as a static process, here we shed light on the impacts of the 21st century global land use change on soil erosion.~~ Insights into land cover and land use change between 2001 and 2012 are achieved by combining the extent, types and spatial distribution of global croplands and forests measured by satellite with agricultural inventory data. Here we shed light on some of the 21st century global land use change on soil erosion and the potential offsetting effects of the conservation practices. The global rainfall erosivity patterns are quantified with a thorough methodology based on rainfall intensity instead

of volume, using a time series of sub-hourly and hourly pluviographic records (3,625 stations covering 63 countries) spatialized through a Gaussian Process Regression (GPR) geo-statistical model.'

Comment (iii), 'How uncertain are the estimate spatially (?)'

In the revised version of the manuscript an estimate of the uncertainty of the spatial modelling predictions using a Markov Chain Monte Carlo (MCMC) approach is provided.

Among the thousands of studies reporting RUSLE estimates only a handful of them provides an uncertainty assessment. This is because the use of uncertainty propagation techniques to estimate the uncertainty associated with RUSLE is a challenging undertaking (Rompaey and Govers, 2002; Falk et al., 2010)^{16,17}.

RUSLE is a purely deterministic model in which the product of physical measures is used to derive the amount of soil loss. As such, unless uncertainty assessments for each original input parameter, the intermediate computational steps up to the final layers (R, K, LS, C and P) are available, a rigorous assessment of the model uncertainties is not feasible.

For similar cases, Monte Carlo analyses (Heuvelink, 1998)¹⁸ are generally used to determinate share of output error caused by modelling input factors. In the latest RUSLE-based global soil erosion estimation (Doetterl, Van Oost and Six, 2012)¹⁹ a Monte Carlo analysis was also employed. Here, we opted for a similar procedure but based on a Markov Chain Monte Carlo (MCMC) approach.

Modification to the main text (Discussions):

The uncertainty of the spatial predictions was estimated using a Markov Chain Monte Carlo (MCMC) approach (Supplementary Note 4). The map of uncertainty is presented in Supplementary Figure S8 as the standard deviation of the MCMC simulated values. The map gives an outline of the geographical distribution of the prediction variance, and it can be used to compare the potential error in different areas of the world. The error of the model estimates associated with the input data assessed with a MCMC approach is about 8 Pg yr⁻¹ for the whole world. Accounting for uncertainties in the soil erosion rates, we estimated an annual average potential soil erosion amount of 35^{+5.6}_{-2.4} and 35.9^{+5.6}_{-2.4} Pg yr⁻¹ for the 2001 and 2012 baseline scenarios, respectively.

Modification to the Supplementary Information:

Supplementary Note 4 | Uncertainty analysis

¹⁶ Rompaey, A. J. V., & Govers, G. (2002). Data quality and model complexity for regional scale soil erosion prediction. *International Journal of Geographical Information Science*, 16(7), 663-680.

¹⁷ Falk, M. G., Denham, R. J., & Mengersen, K. L. (2010). Estimating uncertainty in the revised universal soil loss equation via Bayesian melding. *Journal of agricultural, biological, and environmental statistics*, 15(1), 20-37.

¹⁸ Heuvelink, G. B. (1998). *Error propagation in environmental modelling with GIS*. CRC Press.

¹⁹ Doetterl, S., Van Oost, K., & Six, J. (2012). Towards constraining the magnitude of global agricultural sediment and soil organic carbon fluxes. *Earth Surface Processes and Landforms*, 37(6), 642-655.

The RUSLE is a purely deterministic model in which the product of physical measures is used to derive the amount of soil loss. As such, a rigorous assessment of uncertainties is not feasible, nor would it be meaningful, unless the uncertainties of the input layers and their propagation in the model scheme were quantified. Accordingly, the estimation of the uncertainty in the RUSLE model outputs remains in most case an unaddressed issue⁵¹ (Falk et al., 2010). A thorough quantification of uncertainty associated to the RUSLE model was provided only in a few local-scale studies, mainly dealing with a single model factor such as rainfall⁵³ (Wang et al., 2002), soil type⁵⁴ (Parysow et al., 2001) and topography⁵⁵ (Gertner et al., 2002).

However, in a global scale application most (if not all) of the spatial layers used to derive RUSLE variables lack information about the associated uncertainty. For instance the SRTM DEM, used to calculate the LS factor, misses the adequate spatial information about the land-cover-, latitude- and elevation-depending uncertainty of the data. For this reason, it is impossible to use uncertainty propagation techniques to estimate uncertainty.

Previous RUSLE-based global studies⁵⁶ (Doetterl et al., 2012), proposed a different approach⁵⁷ (Rompaey and Govers, 2012) based on the introduction of random errors in the input layers, with the propagation of this random noise in the final value of soil loss is taken as an estimate for uncertainty⁵⁶ (Doetterl et al., 2012). However, this approach may be problematic as the noise introduced in the layers is arbitrarily chosen (while constrained by physical parameters) in intensity and distribution. Moreover, one could argue that this kind of estimation could be performed even in absence of the input layers as one could simply calculate the product of the different noises introduced and thus derive uncertainty. Another issue is that this approach fails to account for different sources of noise; for instance noise in estimating the texture, could come from the uncertainty about the granulometric fraction within a textural class (as used by Doetterl et al.⁵⁶) or from the misprediction of the textural class itself.

Given previously proposed potential problems associated with the global uncertainty analysis, in this study a different approach was followed representing the uncertainty as a probability distribution through the use of a Bayesian modeling technique. The idea is to use the data distribution to estimate the uncertainty in the prediction. Given that the RUSLE is based on the product, for simplicity all the layers were log-transformed. Next, each of the input layers was treated as a spatial random field. A random field is a stochastic process defined in terms of expectation and covariance, once these two parameters are estimated, different simulation of the field can be created. Each of the simulation has the same parameters, but differs due to the stochasticity of the process. By combining a large number of simulations, one could, in principle, estimate how the uncertainty propagates to the model output (soil loss). As deriving spatially continuous simulations for each of the layers is impractical, a simulation approach based on Gibbs sampling and an additive model was used.

The model is expressed as:

$$z(S_0) = z(R) + z(LS) + z(K) + z(C) + e(s)$$

where the $z()$ values are realization of each of the log-transformed model input layers and $e(s)$ is the spatial component of the model.

A Markov Chain Monte Carlo (MCMC) algorithm, was used to derive realizations of $z(S_0)$ (soil loss) by simulating from the multivariate normal distribution with zero mean and covariance matrix V_b , where V_b is the Bayesian covariance matrix of the fitted model. MCMC was applied using the JAGS software⁵⁸ through R interface⁵⁹.

The map of uncertainty is presented in Supplementary Figure S8 as the standard deviation of the

MCMC simulated values. The map gives an outline of the geographical distribution of the prediction variance, and it can be used to compare the potential error in different areas of the world. As also observed by Teng et al.⁶⁰ in a large-scale analysis in Australia, a tendency of the uncertainties to be lower in areas with denser vegetation cover was found. By contrast, a tendency to higher uncertainty appears in scarcely vegetated area in arid and semiarid regions (e.g., Western Sectors of North and South America, Turkestan and central Asia) but also in areas subject to higher erosion rates such as agricultural area of US, Ethiopia, China, India and Mediterranean Europe.

The error of the model estimates associated with the input data assessed with the proposed MCMC approach is about 8 Pg yr^{-1} for the whole world. The value was calculated by calculating sample quantiles on the simulated data. It should be noted that the error interval is not symmetric around the mean, so the upper error limit (at 0.9 CI) is about 5.6 Pg yr^{-1} , while the lower is narrower at 2.4 Pg yr^{-1} . Nevertheless, the absolute value of the standard deviation has to be taken with caution as the underlying distribution is not normal, the standard deviation cannot be directly used to derive information such as confidence intervals.

Added Figure:

Supplementary Figure S8 Uncertainty analysis. The map of uncertainty presented as the standard deviation of the Markov Chain Monte Carlo (MCMC) simulated values.

Further changes related to the previous comment (ii):

A further suggestion made by Referee #4 in the comment (ii) was to include some more figures reporting the main final modelling layers into the manuscript: *'...list of equations, which are by themselves not very explanatory at this scale/resolution. I would therefore show for each of the*

factors the global map (erosivity, erodibility, LS, C factor). This would improve the credibility of the method.'

We agree with this. Four additional figures were added in the Supplementary Information, including the global map zooming in a subset region in Brazil to highlight the spatial detail of the layers and the average value per Continent.

Supplementary Fig S3 Global rainfall erosivity map. (a) Representation of the global patterns of the rainfall erosivity R-factor (spatial resolution 30 arc-seconds, ca. 1 x 1km). (b) Subset of the global rainfall erosivity map for an area of about 45,000 km² in the West Central Region of Brazil. (c) R-factor average value per continent.

Supplementary Fig S10 Land cover and management factor (C). Figure (a) and (b) share the same legend. (a) Representation of the global patterns of the C-factor (spatial resolution ca. 250 x 250m). (b) Subset of the global C-factor map for an area of about 45,000 km² in the West Central Region of Brazil. (c) C-factor average value per continent.

Supplementary Fig S13 The slope length and steepness factor (LS). Figure (a) and (b) share the same legend. (a) Representation of the global patterns of the LS-factor (original spatial resolution ca. 90 x 90m resample to ca. 250 x 250m). (b) Subset of the global LS-factor map for an area of about 45,000 km² in the West Central Region of Brazil. (c) LS-factor average value per continent.

Supplementary Fig S14 The soil erodibility (K). (a) Representation of the global patterns of the K-factor (original spatial resolution ca. 90 x 90m resample to ca. 250 x 250m). (b) Subset of the global K-factor map for an area of about 45,000 km² in the West Central Region of Brazil. (c) K-factor average value per continent.

REVIEWERS' COMMENTS:

Reviewer #4 (Remarks to the Author):

Dear authors, I have read your remarks and studied the additional material you provided. I found that all my points are well addressed. The text is well revised with careful wording, and it is clear to me what the erosion assessment shows and what it doesn't show as I requested. Supporting additional literature is provided. The examples of the individual factors make the method more comprehensible and the added uncertainty map (and underlying method) is nicely done. I see that providing global maps of the factors shows of course the main structures and climatic zones of the planet, so I appreciate the enlarged examples that are provided to improve understanding.

In summary, I have no further comments and recommend accepting the manuscript without further modifications